# EEG microstates are a candidate endophenotype for schizophrenia

Janir Ramos da Cruz ⬤ [1,2✉], Ophélie Favrod[1], Maya Roinishvili[3,4], Eka Chkonia[4,5], Andreas Brand[1], Christine Mohr[6], Patrícia Figueiredo[2,7] & Michael H. Herzog[1,7]

Electroencephalogram microstates are recurrent scalp potential configurations that remain stable for around 90 ms. The dynamics of two of the four canonical classes of microstates, commonly labeled as C and D, have been suggested as a potential endophenotype for schizophrenia. For endophenotypes, unaffected relatives of patients must show abnormalities compared to controls. Here, we examined microstate dynamics in resting-state recordings of unaffected siblings of patients with schizophrenia, patients with schizophrenia, healthy controls, and patients with first episodes of psychosis (FEP). Patients with schizophrenia and their siblings showed increased presence of microstate class C and decreased presence of microstate class D compared to controls. No difference was found between FEP and chronic patients. Our findings suggest that the dynamics of microstate classes C and D are a candidate endophenotype for schizophrenia.

[1] Laboratory of Psychophysics, Brain Mind Institute, École Polytechnique Fédérale de Lausanne (EPFL), Lausanne, Switzerland. [2] Institute for Systems and Robotics—Lisbon (LARSyS) and Department of Bioengineering, Instituto Superior Técnico, Universidade de Lisboa, Lisbon, Portugal. [3] Laboratory of Vision Physiology, Beritashvili Centre of Experimental Biomedicine, Tbilisi, Georgia. [4] Institute of Cognitive Neurosciences, Free University of Tbilisi, Tbilisi, Georgia. [5] Department of Psychiatry, Tbilisi State Medical University, Tbilisi, Georgia. [6] Faculté des Sciences Sociales et Politiques, Institut de Psychologie, Bâtiment Geopolis, Lausanne, Switzerland. [7] These authors contributed equally: Patrícia Figueiredo, Michael H. Herzog. ✉email: janir.ramos@epfl.ch

Schizophrenia is a heterogeneous disease strongly determined by genetics. However, no strong genetic correlates have been found yet[1,2]. For this reason, endophenotypes are of crucial interest. Endophenotypes are associated with the illness, state-independent, co-segregate within families, and found in unaffected relatives of individuals with the disorder at a higher prevalence than in the general population[3]. Abnormal temporal dynamics of electroencephalogram (EEG) microstates were proposed as an endophenotype for schizophrenia[4–6]. Microstates are global patterns of scalp potential topographies that remain quasi-stable for around 60–120 ms before changing to a different topography that remains quasi-stable again, suggesting semi-simultaneity of activity of large-scale brain networks[7].

EEG microstates are highly reproducible, both within and across participants[8]. This allows the use of clustering algorithms to group microstates into a finite set of classes based on their topographical similarity[9,10]. Even though there is still no general consensus on how to determine the optimal number of microstate classes[5,10–13] and the optimal number of classes of microstates may depend on the dataset analyzed[13,14], in clinical research, usually, four classes of microstates, labeled A, B, C, and D, are used[15] based on pioneering work[16–18]. These four dominant classes of microstates are consistently observed in resting-state EEG (independently of the number of electrodes, and group of participants), explaining 65–84% of the global variance of the data[5]. Here, we used these four canonical classes of microstates because this allows comparison between studies.

Several studies have attempted to identify the brain sources underlying these classes of microstates[13,16,19–22]. A direct comparison of the findings is difficult due to the differences in data acquisition and processing as well as the number of microstate classes used and the way the microstates analyses were performed. Nonetheless, these studies indicate that EEG microstates are closely related to resting-state networks (RSNs) commonly found in resting-state functional magnetic resonance imaging (fMRI) despite the different time resolutions of the two modalities (see[5] for a review). Among the above-mentioned studies, the one by Britz et al.[16] is usually referred to when discussing fMRI correlates of EEG microstates since it used an approach more closely related to the conventional EEG microstate analysis. Britz et al. found that microstate class A was associated with the auditory RSN, microstate class B with the visual RSN, microstate class C with the salience RSN, and microstate class D with the attention RSN. Similar results were found using source localization of the EEG microstate classes[13], providing further evidence for these associations. However, since both fMRI RSNs and EEG microstates are still lacking a full understanding of their significance in terms of the functional organization of brain networks, one should be cautious when interpreting these associations.

In schizophrenia research, numerous studies have reported abnormalities in the temporal dynamics of EEG microstates measured in patients with schizophrenia compared with controls[4,5]. A meta-analysis comprising seven studies from 1999 to 2015[15] revealed that microstate class C occurred more frequently and for longer durations in patients than in controls, whereas microstate class D occurred less frequently and for shorter durations. Microstate class B was shorter in patients than controls, but the effect was not significant after correction for multiple comparisons.

Similar abnormalities were also observed in adolescents with 22q11.2 deletion syndrome, a population that has a 30% risk of developing psychosis[6,23]. These results have prompted researchers to suggest that the abnormal EEG microstates dynamics may be an endophenotype for schizophrenia[6]. For an endophenotype, it is important that the siblings of the patients also show abnormal patterns. No such study exists for microstates.

Here, we analyze the microstates dynamics in unaffected siblings of patients with schizophrenia, patients with schizophrenia, and healthy controls. To preface our results, siblings show abnormalities in microstate classes C and D, similar to patients. Surprisingly, siblings also show increased presence of microstate class B compared with patients. We interpret this increased presence of microstate class B as a compensation signal, which might prevent unaffected siblings to develop the disorder even if there is a genetic predisposition.

In a second study, we investigate whether patients with a first episode of psychosis (FEP) show similar microstates dynamics as chronic patients with schizophrenia or have the compensation signal as the siblings (i.e., increased presence of microstate class B), since the disorder has not fully blown. Moreover, we test FEP three times throughout 1 year to assess whether the microstates dynamics change with the progression on the disorder.

Finally, we perform a meta-analysis over studies investigating the EEG microstates dynamics in the schizophrenia spectrum to provide an up-to-date estimate of the overall effect sizes of microstate anomalies in schizophrenia.

## Results

**Study 1**. We examined 5 min resting-state EEG data of 101 patients with schizophrenia, 43 siblings of patients with schizophrenia, and 75 healthy controls, and we estimated the dynamics of the four canonical EEG microstate classes using Cartool[24]. The four microstate classes for patients, siblings, and controls are shown in Fig. 1a. The four microstate classes across participants explained 80.33%, 82.80%, and 78.25% of the global variance in the patients, siblings, and control group, respectively. In each of the three groups, the four microstates resembled the four class model maps consistently identified in the literature[5]: two microstate classes (A and B) with diagonal axis orientations of the topographic map field, one class (C) with anterior–posterior orientation, and one class (D) with a fronto-central extreme location.

For each participant, three per class microstate parameters were computed: mean duration, time coverage, and frequency of occurrence (occurrence). Mean duration (in ms) is the average time that a given microstate was uninterruptedly present. Time coverage (in %) is the percentage of the total time spent in a given microstate. Occurrence is the mean number of times a given microstate is occurring per second. Group average statistics are depicted in Fig. 1b–d.

For patients vs. controls, three-way repeated measures (rm) ANOVAs showed nonsignificant Gender × Microstate Class × Group interaction for mean duration ($F(3,513) = 1.653$, $p = 0.176$, $\eta^2 = 0.007$, 90% CI [<0.001, 0.023]), time of coverage ($F(3,513) = 2.063$, $p = 0.104$, $\eta^2 = 0.010$, 90% CI [<0.001, 0.027]), and occurrence ($F(3,513) = 1.136$, $p = 0.334$, $\eta^2 = 0.005$, 90% CI [<0.001, 0.018]). The analyses also yielded significant Microstate Class × Group interaction effects for mean duration ($F(3,513) = 16.246$, $p = 4.219e-10$, $\eta^2 = 0.071$, 90% CI [0.048, 0.123]), time of coverage ($F(3,513) = 17.458$, $p = 8.316e-11$, $\eta^2 = 0.086$, 90% CI [0.053, 0.130]), and occurrence ($F(3,513) = 8.477$, $p = 1.664e-5$, $\eta^2 = 0.035$, 90% CI [0.019, 0.076]). These interactions indicate that group differences depend on the microstate class. Post hoc pairwise group comparisons (Table 1) showed that patients had decreased mean duration of microstate class B compared with controls. For microstate class C, patients had increased values compared with controls for all the computed microstate parameters. While for microstate class D, patients had decreased values compared with controls for all computed

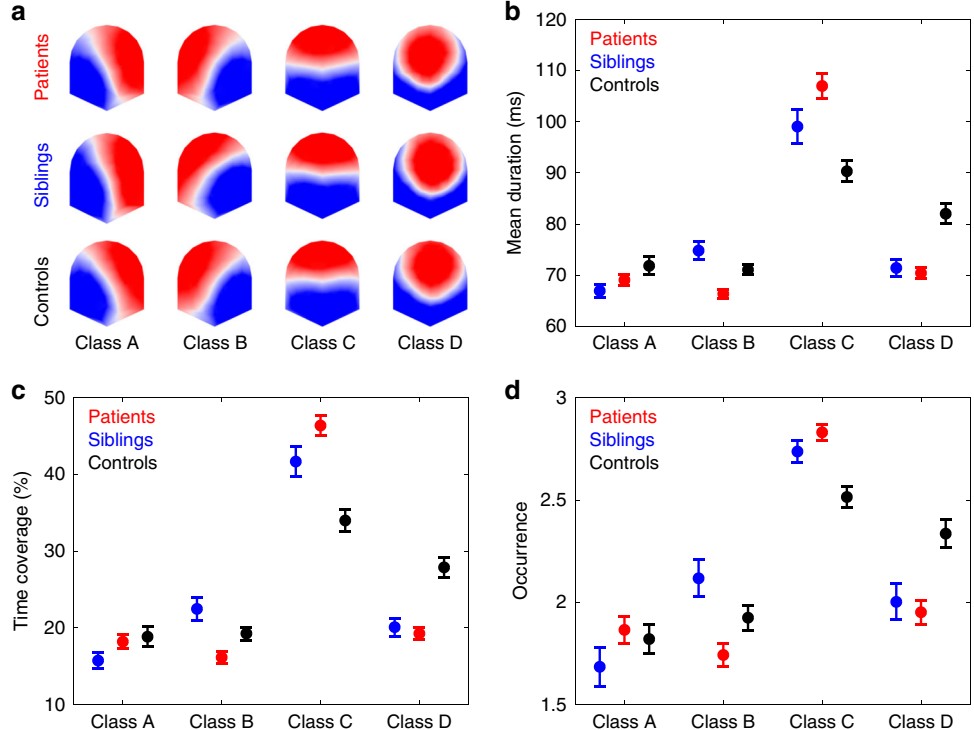

**Fig. 1 Results of the microstate analysis for study 1.** Patients data are displayed in red ($n = 101$), siblings in blue ($n = 43$), and controls in black ($n = 75$). **a** The spatial configuration of the four microstate classes (A, B, C, and D) for the three groups. Group average statistics of the temporal microstate parameters: **b** mean duration, **c** time coverage, and **d** occurrence. For patients vs. controls, since groups differed in education and gender, for each microstate parameter, we computed a repeated measured (rm)-ANOVA with Group, Microstate Class, and Gender as factors and Education as covariate. Following significant Group × Microstate Class interactions, we performed post hoc group comparisons (Table 1). For siblings vs. controls, for each parameter, we computed an rm-ANOVA with Group and Microstate Class as factors. Following significant Group × Microstate Class interactions, we performed post hoc group comparisons (Table 2). For patients vs. siblings, we performed three analyses: (1) paired a patient with his/her corresponding sibling and calculated a difference score ($n = 32$ patient–sibling pairs), (2) compared all patients against all siblings, and (3) compared patients without siblings in current study ($n = 69$) against all siblings. For the analysis (1), patient–sibling pairs difference scores were submitted to a two-sided one-sample $t$-test against 0 (Table 3). For analyses (2) and (3), since groups differed in age and gender, for each microstate parameter, we computed a rm-ANOVA with Group, Microstate Class, and Gender as factors and Age as covariate. Following significant Group × Microstate Class interactions, we performed post hoc group comparisons (Supplementary Tables 5 and 6). For each analysis (patients vs. controls, siblings vs. controls, patients vs. siblings), group comparisons for all microstate parameters and microstate classes were corrected for multiple comparisons using Bonferroni–Holm correction. Patients and siblings showed increased presence of microstate class C and decreased presence of microstate class D compared controls. Patients showed decreased mean duration of microstate class B compared with controls and an overall decreased presence of microstate class B compared with siblings. Error bars indicate s.e.m. Group average statistics are also shown in Supplementary Table 3. Source data are provided as a Source Data file.

### Table 1 Patients vs. controls for all microstate parameters and for each microstate class.

| Parameter | Microstate | $p$ | $p_{holm}$ | $d$ | 95% CI |
|---|---|---|---|---|---|
| Mean duration | Class A | 0.054 | 0.270 | −0.293 | [−0.593, 0.008] |
| | Class B | **0.003** | **0.018** | **−0.454** | **[−0.756, −0.151]** |
| | Class C | **1.315e − 4** | **0.001** | **0.590** | **[0.284, 0.894]** |
| | Class D | **3.010e − 6** | **3.311e − 5** | **−0.732** | **[−1.039, −0.423]** |
| Time coverage | Class A | 0.449 | 0.898 | −0.110 | [−0.409, 0.189] |
| | Class B | 0.074 | 0.296 | −0.271 | [−0.571, 0.029] |
| | Class C | **1.452e − 7** | **1.742e − 6** | **0.827** | **[0.515, 1.137]** |
| | Class D | **3.445e − 6** | **3.445e − 5** | **−0.725** | **[−1.032, −0.416]** |
| Occurrence | Class A | 0.882 | 0.898 | 0.023 | [−0.276, 0.322] |
| | Class B | 0.112 | 0.336 | −0.247 | [−0.547, 0.053] |
| | Class C | **1.170e − 4** | **0.001** | **0.602** | **[0.296, 0.907]** |
| | Class D | **1.620e − 4** | **0.001** | **−0.578** | **[−0.882, −0.272]** |

Post hoc pairwise group comparisons of EEG microstate dynamics of patients ($n = 101$) and controls ($n = 75$). $p$ values refer to main effects of group following Group × Gender ANCOVAs with Education as a covariate; degrees of freedom (df) of the numerators are 1 and df for the denominators are 171. $p_{holm}$ values refer to Bonferroni–Holm corrected $p$ values for 12 comparisons (3 parameters × 4 classes). $\eta^2$'s were converted to Cohen's $d$'s. Statistically significant differences are indicated in bold.

**Table 2 Siblings vs. controls for all microstate parameters and for each microstate class.**

| Parameter | Microstate | $p$ | $p_{holm}$ | $d$ | 95% CI |
|---|---|---|---|---|---|
| Mean duration | Class A | 0.055 | 0.288 | −0.371 | [−0.748, 0.008] |
| | Class B | 0.049 | 0.288 | 0.381 | [0.002, 0.758] |
| | Class C | 0.022 | 0.154 | 0.445 | [0.065, 0.823] |
| | Class D | **3.380e − 4** | **0.004** | **−0.707** | **[−1.091, −0.319]** |
| Time coverage | Class A | 0.097 | 0.288 | −0.320 | [−0.696, 0.058] |
| | Class B | 0.048 | 0.288 | 0.382 | [0.003, 0.759] |
| | Class C | **0.001** | **0.010** | **0.631** | **[0.246, 1.013]** |
| | Class D | **1.465e − 4** | **0.002** | **−0.751** | **[−1.137, −0.362]** |
| Occurrence | Class A | 0.250 | 0.288 | −0.221 | [−0.597, 0.155] |
| | Class B | 0.069 | 0.288 | 0.351 | [−0.028, 0.728] |
| | Class C | **0.006** | **0.048** | **0.533** | **[0.151, 0.913]** |
| | Class D | **0.004** | **0.036** | **−0.566** | **[−0.947, −0.183]** |

Post hoc pairwise group comparisons of EEG microstate dynamics of siblings ($n = 43$) and controls ($n = 75$). $p$ values refer to two-sided independent samples $t$-tests, with degrees of freedom (df) = 116. $p_{holm}$ values refer to Bonferroni–Holm corrected $p$ values for 12 comparisons (3 parameters × 4 classes). Statistically significant differences are indicated in bold.

**Table 3 Patients_32 vs. Siblings_32 for all microstate parameters and for each microstate class.**

| Parameter | Microstate | $p$ | $p_{holm}$ | $d$ | 95% CI |
|---|---|---|---|---|---|
| Mean duration | Class A | 0.235 | 1.000 | 0.214 | [−0.138, 0.563] |
| | Class B | **0.003** | **0.036** | **−0.576** | **[−0.947, −0.198]** |
| | Class C | 0.838 | 1.000 | −0.036 | [−0.383, 0.310] |
| | Class D | 0.270 | 1.000 | −0.199 | [−0.547, 0.153] |
| Time coverage | Class A | 0.020 | 0.200 | 0.434 | [0.068, 0.794] |
| | Class B | 0.069 | 0.621 | 0.333 | [−0.687, 0.025] |
| | Class C | 0.783 | 1.000 | 0.049 | [−0.298, 0.395] |
| | Class D | 0.491 | 1.000 | −0.123 | [−0.470, 0.225] |
| Occurrence | Class A | 0.014 | 0.154 | 0.462 | [0.094, 0.824] |
| | Class B | 0.200 | 1.000 | −0.231 | [−0.581, 0.122] |
| | Class C | 0.588 | 1.000 | 0.097 | [−0.251, 0.443] |
| | Class D | 0.772 | 1.000 | −0.052 | [−0.398, 0.295] |

Comparison of the difference scores of EEG microstate dynamics of patients and their sibling pair ($n = 32$ patient–sibling pairs). $p$ values refer to two-sided one sample $t$-tests against 0, with degrees of freedom (df) = 31. $p_{holm}$ values refer to Bonferroni–Holm corrected $p$ values for 12 comparisons (3 parameters × 4 classes). Statistically significant differences are indicated in bold.

microstates parameters. No statistically significant group differences were found for microstate class A.

For siblings vs. controls, two-way rm-ANOVAs yielded significant Microstate Class × Group interaction effects duration ($F(3,348) = 8.061$, $p = 3.310e − 5$, $\eta^2 = 0.041$, 90% CI [0.025, 0.105]), time of coverage ($F(3,348) = 9.073$, $p = 8.472e − 6$, $\eta^2 = 0.048$, 90% CI [0.030, 0.114]), and occurrence ($F(3,348) = 6.938$, $p = 1.511e − 4$, $\eta^2 = 0.031$, 90% CI [0.019, 0.094]). Post hoc pairwise group comparisons (Table 2) showed that for microstate class C, siblings had increased time coverage and occurrence compared with controls. For microstate class D, siblings had decreased values compared with controls for all computed microstates parameters.

Regarding patients vs. siblings comparisons, since 32 out of the 43 siblings were each a sibling of a patient in the current study (referred to as siblings_32 and patients_32, respectively), we paired these 32 patients to their siblings and compared their difference score (Δ) for each microstate parameter and class against 0 with one sample $t$-tests. Results revealed that siblings_32 had a longer mean duration of microstate class B than their paired patients_32 ($\Delta = −7.21 \pm 12.50$ ms; Table 3 and Supplementary Table 3). The mean duration of microstate class B and the occurrence of microstate class C of patients_32 correlated with the mean duration of microstate class B and the occurrence of microstate class C in their paired siblings_32 (Supplementary Table 4). However, the correlations were not significant after correction for multiple comparisons (mean duration of microstate

class B: $r(30) = 0.360$, $p = 0.043$, $p_{holm} = 0.473$; occurrence of microstate class C: $r(30) = 0.430$, $p = 0.014$, $p_{holm} = 0.168$). We also compared the microstates dynamics of all patients compared with all siblings (Supplementary Table 5). Results indicated increased mean duration, time coverage, and occurrence of microstate class B in siblings compared with patients. Similar results were found comparing the microstate dynamics of patients without siblings in the current study ($n = 69$) against all siblings (Supplementary Table S6).

We correlated the values of the computed microstate parameters of patients with patients' medication intake (chlorpromazine (CPZ) equivalent), Scales for the Assessment of Negative Symptoms (SANS) and Scales for the Assessment of Positive Symptoms (SAPS) scores, and illness duration (Supplementary Table S7). CPZ equivalents correlated with the occurrence of microstate class C, but the association was not significant after correcting for multiple comparisons ($r(86) = 0.236$, $p = 0.027$, $p_{holm} = 0.324$).

**Study 2.** We examined 5 min resting-state EEG data of 22 FEP and 22 chronic patients with schizophrenia (Patients_22; selected pseudo-randomly from our pool of 101 chronic patients with schizophrenia (see "Study 1") to match the 22 FEP as closely as possible, regarding to gender, age, and education). The microstates analysis was the same as in "Study 1". The four microstate classes for the FEP group are shown in Fig. 2a. The four

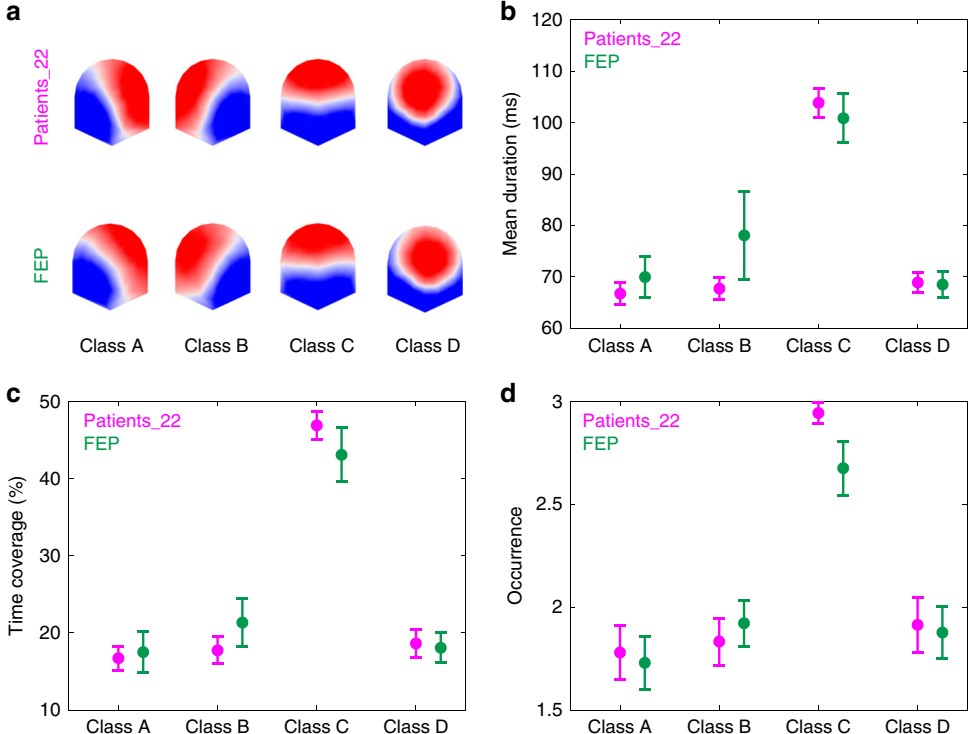

**Fig. 2 Results of the microstate analysis for study 2.** Patients with first episode of psychosis (FEP, $n = 22$) data are displayed in green and their matched patients with schizophrenia (Patients_22, $n = 22$) data in magenta. **a** The spatial configuration of the four microstate classes (A, B, C, and D) for the two groups. Group average statistics of the temporal microstate parameters: **b** mean duration, **c** time coverage, and **d** occurrence. For each microstate parameter, we computed a two-way repeated measures ANOVA with Group (FEP and Patients_22) and Microstate Class (A, B, C, and D) as factors. No statistically significant group differences and no statistically significant Group × Microstate Class interaction effects were found for any of the microstates parameters. Error bars indicate s.e.m. Group average statistics are also shown in Supplementary Table 9. Source data are provided as a Source Data file.

microstate classes explained 73.97% of the global variance across participants. The lower percentage of explained variance compared with patients, siblings, controls ("Study 1"), though in the normal range reported in the literature (65–84%[5]), might be due to the diverse diagnosis of the FEP group (Supplementary Table 12). Similarly to patients, siblings, and controls (Fig. 1a), the four microstates resembled the four class model maps consistently identified in the literature.

We found no statistically significant differences between the FEP and Patients_22 groups for any of the computed microstates parameters. Two-way rm-ANOVAs yielded nonsignificant group (FEP and Patients_22) × microstate class (A, B, C, and D) interactions for mean duration ($F(3,126) = 0.821$, $p = 0.485$, $\eta^2 = 0.011$, 90% CI [<0.001, 0.054]), time of coverage ($F(3,126) = 0.633$, $p = 0.595$, $\eta^2 = 0.007$, 90% CI [<0.001, 0.042]), and occurrence ($F(3,126) = 0.860$, $p = 0.464$, $\eta^2 = 0.009$, 90% CI [<0.001, 0.056]), as well as nonsignificant group differences for mean duration ($F(1,42) = 1.819$, $p = 0.185$, $\eta^2 = 0.042$, 90% CI [<0.001, 0.170]), time of coverage ($F(1,42) = 0.051$, $p = 0.823$, $\eta^2 = 0.001$, 90% CI [<0.001, 0.055]), and occurrence ($F(1,42) = 0.484$, $p = 0.490$, $\eta^2 = 0.011$, 90% CI [<0.001, 0.110]).

Since null results are relevant to the overall interpretation of the results, we conducted two additional analyses to evaluate the sensitivity of our study and whether there were supporting evidence for the null hypotheses. First, we conducted a sensitivity analysis with the program G*Power[25] to compute the interaction and main effect of group effect sizes that we can detect with a power of 80%, given 22 participants in each of the two groups and four microstate classes. The analysis revealed that we could detect interaction effects and main effects of group with main effect sizes with $\eta^2$ of 0.068 and

0.026, which are medium and small effect sizes according to Cohen[26].

Second, we examined the data by estimating Bayes factors using Bayesian information criteria[27], comparing the fit of the data under the main effects model and the interaction model for each of the computed microstate parameters. JZS Bayes factor ANOVAs[28] with default prior scales revealed that the main effects models were preferred to the interaction model by Bayes factors of 5.545, 6.609, and 6.236, for mean duration, time of coverage, and occurrence, respectively. In other words, the data provided positive evidence against the hypotheses that Group and Microstate Class interact in any of the computed microstates parameters. We further compared the main effects models and models without the main effect of Group. Results show that models without the main effect of Group were preferred to the main effects models by Bayes factors of 4.129, 5.444, and 3.841, for mean duration, time of coverage, and occurrence, respectively.

We correlated the values of the computed microstate parameters in FEP with FEP's medication intake (CPZ equivalent), SANS and SAPS scores, and illness duration (Supplementary Table 10). We found that SANS scores correlated negatively with the time coverage and occurrence of microstate class D, but the associations were not significant after correcting for multiple comparisons (time coverage of microstate class D: $r(20) = -0.516$, $p = 0.014$, $p_{\text{holm}} = 0.168$; occurrence of microstate class C: $r(20) = -0.429$, $p = 0.046$, $p_{\text{holm}} = 0.506$).

We tested the FEP group three times throughout 1 year to assess whether the microstates dynamics changed with the progression of the disease. Out of the 22 FEP, 16 participated 6 months later on a second session (FEP_2). Out of these 16, 11 were tested 6 months later on a third session (FEP_3).

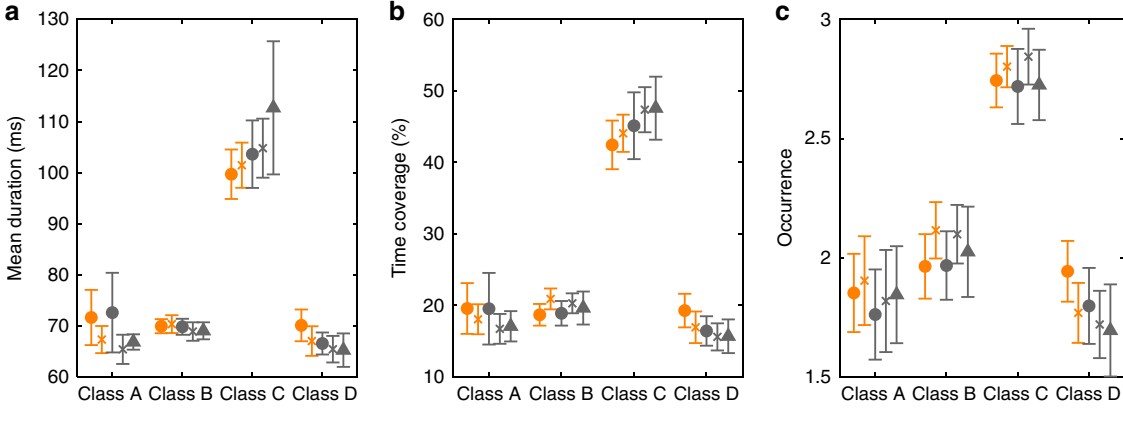

**Fig. 3 Results of the re-testing of patients with a first episode of psychosis.** Data of patients with a first episode of psychosis that participated in the first and second testing sessions (FEP_2, $n = 16$) are displayed in orange and data of patients that participated in all three testing sessions (FEP_3, $n = 11$) are in gray. Group average statistics of the temporal microstate parameters: **b** mean duration, **c** time coverage, and **d** occurrence. For each microstate parameter, we computed a two-way repeated-ANOVA with Testing Session (for FEP_2: first and second; for FEP_3: first, second, and third) and Microstate Class (A, B, C, and D) as factors. No statistically significant testing session differences and no statistically significant Testing Session × Microstate Class interaction effects were found for any of the computed microstates parameters. Circles indicate the mean values in the first testing session, crosses indicate the mean values in the second testing session, and triangles the mean values in the third testing session. Error bars indicate s.e.m. Group average statistics are also shown in Supplementary Table 11. Source data are provided as a Source Data file.

Summary statistics of the computed microstates parameters for the FEP group for the three measurements throughout 1 year (FEP_2 and FEP_3) is shown in Fig. 3.

For the FEP_2 comparison, two-way rm-ANOVAs yielded no nonsignificant Testing Session (First and Second Session) × Microstate Class (A, B, C, and D) interactions for mean duration ($F(3,45) = 0.433$, $p = 0.730$, $\eta^2 = 0.004$, 90% CI [<0.001, 0.083]), time of coverage ($F(3,45) = 0.512$, $p = 0.676$, $\eta^2 = 0.005$, 90% CI [<0.001, 0.095]), and occurrence ($F(3,45) = 1.060$, $p = 0.375$, $\eta^2 = 0.009$, 90% CI [<0.001, 0.156]), as well as nonsignificant testing session differences for mean duration ($F(1,15) = 3.416$, $p = 0.084$, $\eta^2 = 0.001$, 90% CI [<0.001, 0.426]), time of coverage ($F(1,15) = 1.000$, $p = 0.333$, $\eta^2 = 4.729e - 14$, 90% CI [<0.001, 0.293]), and occurrence ($F(1,15) = 0.171$, $p = 0.685$, $\eta^2 = 2.812e - 4$, 90% CI [<0.001, 0.187]). JZS Bayes factor ANOVAs with default prior scales revealed that the main effects models were preferred to the interaction model by Bayes factors of 4.739, 7.957, and 5.824, for mean duration, time of coverage, and occurrence, respectively. Moreover, the analyses revealed that models without the main effect of Testing Session were preferred to the main effects models by Bayes factors of 8.440, 5.464, and 5.051, for mean duration, time of coverage, and occurrence, respectively.

For the FEP_3 comparison, two-way rm-ANOVAs yielded no nonsignificant Testing Session (First, Second, and Third Session) × Microstate Class (A, B, C, and D) interactions for mean duration ($F(6,60) = 0.513$, $p = 0.796$, $\eta^2 = 0.009$, 90% CI [<0.001, 0.061]), time of coverage ($F(6,60) = 0.210$, $p = 0.972$, $\eta^2 = 0.003$, 90% CI [<0.001, 0.006]), and occurrence ($F(6,60) = 0.255$, $p = 0.955$, $\eta^2 = 0.004$, 90% CI [<0.001, 0.008]), as well as nonsignificant testing session differences for mean duration ($F(2,20) = 0.885$, $p = 0.428$, $\eta^2 = 0.002$, 90% CI [<0.001, 0.244]), time of coverage ($F(2,20) = 0.443$, $p = 0.648$, $\eta^2 = 3.214e - 14$, 90% CI [<0.001, 0.176]), and occurrence ($F(2,20) = 0.289$, $p = 0.752$, $\eta^2 = 0.001$, 90% CI [<0.001, 0.140]). JZS Bayes factor ANOVAs with default prior scales revealed that the main effects models were preferred to the interaction model by Bayes factors of 14.333, 24.000, and 22.500, for mean duration, time of coverage, and occurrence, respectively. Moreover, the analyses revealed that models without the main effect of Testing Session were preferred to the main effects models by Bayes factors of

11.628, 13.889, and 11.111, for mean duration, time of coverage, and occurrence, respectively.

**Meta-analysis.** Our literature search identified eight independent studies comparing the resting-state dynamics of the four canonical microstate classes in patients belonging to the schizophrenia spectrum against a control group[18,23,29–34]. In addition to these eight studies, we also included the current study in the meta-analysis. Forest plots of the mean effect sizes for each microstate parameter (mean duration, time coverage, and occurrence) and microstate class (A, B, C, and D) are shown in Supplementary Figs. 1–12. Similar to Rieger et al.[15], we found consistently increased time coverage ($g = 0.447$, 95% CI [0.228, 0.666], $p = 6.304e - 5$, $p_{holm} = 6.934e - 4$) and occurrence ($g = 0.688$, 95% CI [0.504, 0.872], $p = 2.430e - 13$, $p_{holm} = 2.916e - 12$) of microstate class C in patients compared with controls, as well as decreased time coverage ($g = -0.506$, 95% CI [−0.839, −0.172], $p = 0.003$, $p_{holm} = 0.027$) and mean duration ($g = -0.540$, 95% CI [−0.853, −0.227], $p = 7.170e - 4$, $p_{holm} = 0.007$) of microstate class D in patients compared with controls. We also found a decreased mean duration of microstate class B in patients compared with controls; however, the effect was not significant after correction for multiple comparisons ($g = -0.353$, 95% CI [−0.642, −0.063], $p = 0.017$, $p_{holm} = 0.136$). No consistent group differences were found for microstate class A.

## Discussion
Several studies have consistently identified abnormal temporal dynamics of EEG microstates in patients with schizophrenia[6,18,30,31,33–36]. Similar patterns were found in patients with 22q11.2 deletion syndrome[23]. Based on these findings, Tomescu et al. suggested that alterations in the temporal dynamics of EEG microstates are a candidate endophenotype for schizophrenia[6]. For an endophenotype, it is important that unaffected relatives also show abnormalities, pointing to the genetic underpinnings of the disease[3].

Here, we showed that siblings and patients show similar microstates dynamics: increased presence of microstate class C and decreased presence of microstate class D compared with

controls. These results suggest that microstate classes C and D capture some genetic component shared by the patients and their unaffected siblings. For microstate class B, patients showed decreased mean durations compared with controls. Surprisingly, microstate class B was more present in siblings compared with patients. No statistically significant group differences were found for microstate class A.

We also analyzed the EEG microstates of 22 FEP patients and a subset of 22 chronic patients with schizophrenia, selected pseudo-randomly to match the FEP patients' demographics as close as possible. We found no evidence for differences between the two groups in any of the microstate parameters of any of the microstate classes. We re-tested FEP patients two other times, separated by 6 months, and found that, in general, the microstates dynamics remained stable. However, this interpretation should be taken with care since only a subset of the initial 22 FEP (16 in the second testing and 11 in the third testing) participated in the three tests. Nevertheless, these results suggest that the microstates abnormalities are present at the beginning of the disease and remain stable until chronicity is established, which is important for fulfilling the requirements of an endophenotype[3].

Finally, we conducted a meta-analysis over nine studies investigating the EEG microstates dynamics in the schizophrenia spectrum (including the current study) to provide an up-to-date estimate of the overall effect sizes of microstate dynamics abnormalities in schizophrenia. In general, the results of our meta-analysis were similar as the ones reported in a previous meta-analysis[15]. Namely, increased presence of microstate class C and decreased presence of microstate class D in patients compared with controls, with medium effect sizes. Decreased mean duration of microstate class B in patients compared with controls, with small effect size; though this effect was not significant after correcting for multiple comparisons.

In sum, the dynamics of resting-state EEG microstates, particularly classes C and D, are a potential endophenotype for schizophrenia since it meets most of the major criteria proposed by Gottesman and Gould[3] and further practicability and explicability criteria proposed by Turetsky et al.[37], discussed one-by-one below. Association with the disease: abnormalities in the temporal metrics of microstate classes C and D have been associated with schizophrenia for almost 20 years, with medium effect sizes[15]. Relatives: here, we showed that unaffected siblings show similar abnormalities as their ill relatives. State independency: here, we showed that FEP show similar microstates dynamics as chronic patients and that, in FEP, the dynamics remain stable throughout 1 year. We did not directly compare the FEP against healthy controls or the effects of medication, but several other studies have done so and found that FEP and un-medicated chronic patients also show similar microstate classes C and D deviations[15,18,32–36]. Practicability: resting-state EEG is easily recorded in a 5 min session, and EEG montages with as low as 19 electrodes can be used for microstates analysis[38]. Moreover, even though differences in EEG preprocessing and temporal smoothing parameters might influence the results of microstate analysis, it has been shown that microstate analysis has a high test–retest reliability, independently of the clustering algorithms applied and the number of electrodes used[8]. Explicability: the abnormal microstates dynamics in schizophrenia are viewed as an imbalance between processes that load on saliency (microstate class C), which are increased, and processes that integrate contextual information (microstate class D), which are reduced[15]. This interpretation goes in line with the view of schizophrenia as a state of abnormal assignment of saliency[39] and a disorder affecting attentional processes, context update, and executive control[40]. Heritability: we currently have no information on the heritability of the patterns of microstate dynamics. However, we found positive correlations between the values of microstate parameters in patients and their siblings for occurrence of microstate class C and mean duration of microstate class B. Although the current study was not conceived to study heritability and these correlations did not survive multiple comparisons, these correlations provide some weak evidence that patterns of microstate dynamics might be heritable. Further studies, designed to address the heritability issue, might provide further evidence for this hypothesis.

We speculate that EEG microstate dynamics are not only a candidate endophenotype, but also, as our results suggest, they might reveal a potential compensation signal in unaffected siblings of patients with schizophrenia. We associate this compensation signal with the increased presence of microstate class B present in this population. More specifically, even though patients and siblings share similar traits, e.g., dynamics of microstate classes C and D, which might indicate vulnerability for schizophrenia, siblings can somehow counteract these traits by having an increased presence of microstate class B. Little is known about microstate class B. It has been related to a resting-state visual network in fMRI[13,16]. In healthy participants, it is the shortest and least frequent microstate from adolescence on[38,41]. Moreover, the visual network is expected to reach maturation much earlier than higher order cognitive networks[42]. Combined together, these observations suggest that the dynamics of microstate class B might be an early marker to discriminate people that are at risk to develop schizophrenia from those that might compensate for their vulnerability.

Regarding associations between psychopathological symptoms and microstate dynamics, in FEP, we observed negative correlations between the SANS scores and the time coverage of microstate class D as well as between the SANS scores and the occurrence of microstate class D; however, the correlations were not significant after correcting for multiple comparisons. For chronic patients with schizophrenia, we found no significant correlations between psychopathological symptoms and the microstate parameters for any of the microstate classes. Nonetheless, the coefficients of the Pearson correlation between the SANS scores of patients with schizophrenia and their microstate parameters for microstate class D were negative as in FEP (mean duration: $r(99) = -0.144$, $p = 0.150$; time coverage: $r(99) = -0.190$, $p = 0.057$; occurrence: $r(99) = -0.183$, $p = 0.067$; Supplementary Table 7). In the literature, the duration of microstate class D has been found to correlate negatively with scores of paranoid-hallucinatory symptomatology[18] and with acute hallucination experiences[43] in patients with schizophrenia. More recently, it has been reported that the time coverage of microstate class A correlated positively with avolition–apathy scores, even though there were no group differences between patients and controls[30]. Finally, in a sample of adolescents with 22q11.2 deletion syndrome, the mean durations of microstate class C were associated with increased hallucination subscores of the structured interview for prodromal syndromes[23]. These results suggest that there might be an association between the microstate dynamics and psychopathological symptoms; however, the results in the literature are too heterogeneous to make firm conclusions at this point.

There are several considerations that should be taken into account. First, there are demographics differences between patients with schizophrenia, their siblings, and controls. Tomescu et al.[41] showed evidence for age- and gender-specific effects on the microstates dynamics. Here, we tried to minimize these effects by using age as a covariate and gender as a factor in the analyses. Second, schizophrenia is a heterogeneous disease and our samples may be too small to cover the full schizophrenia spectrum. Third, we cannot exclude the potential

**Table 4 Group average statistics (±SD) of patients, their siblings, controls, Patients_32, and Siblings_32.**

|  | Patients | Siblings | Controls | Patients_32 | Siblings_32 |
|---|---|---|---|---|---|
| Gender (F/M) | 11/90 | 21/22 | 39/36 | 4/28 | 14/18 |
| Age (years) | 36.9 ± 8.8 | 31.8 ± 10.4 | 35.1 ± 7.7 | 33.6 ± 9.1 | 31.9 ± 9.6 |
| Education (years) | 13.4 ± 2.7 | 14.1 ± 3.0 | 15.1 ± 2.9 | 13.6 ± 2.7 | 14.4 ± 3.0 |
| Handedness (L/R) | 6/95 | 2/41 | 4/71 | 3/29 | 2/30 |
| Illness duration (years) | 12.8 ± 8.2 |  |  | 9.4 ± 7.4 |  |
| SANS | 10.4 ± 5.2 |  |  | 10.2 ± 5.3 |  |
| SAPS | 9.7 ± 7.7 |  |  | 8.9 ± 3.5 |  |
| CPZ equivalent[a] | 577.2 ± 400.9 |  |  | 541.4 ± 375.9 |  |

*SANS* Scale for the Assessment of Negative Symptoms, *SAPS* Scale for the Assessment of Positive Symptoms, *CPZ* chlorpromazine.
[a]Average CPZ equivalents calculated over the 88 Patients and 27 Patients_32 receiving neuroleptic medication, respectively.

effects of treatment in the microstate class B differences between siblings and patients.

Most of our patients were medicated and, in chronic patients with schizophrenia, we found a positive association with medication intake (CPZ equivalents) and the occurrence of microstate class C (although not significant after correction for multiple comparisons), providing evidence that medication interact with microstates dynamics. This potential interaction is also supported by previous work that has shown that perospirone (an antipsychotic drug) can increase the duration of microstate class D in healthy controls[44]. In addition, antipsychotic medication has been shown to normalize microstate dynamics (decrease presence of microstate class C and increase presence of microstate class D) in patients that respond well to antipsychotic treatment[33]. While most of the studies included in our meta-analysis only investigated the microstates dynamics in medication naïve patients, few studies investigated patients taking antipsychotic medication. One of these studies found that microstate class D was decreased in FEP compared with controls[31], while another found increased duration and time coverage of microstate class C in patients with schizophrenia compared with controls[30]. In addition, a recent study with FEP also identified decreased mean durations of microstate class A in FEP compared with controls, a result that does not align with the literature[29]. However, since most of the studies of EEG microstates in schizophrenia have small samples ($n < 30$), it is expected that, due to sampling error and the heterogeneity of the disorder, some effects might not be significant in some studies and even reversed in a few studies if the effect sizes are small.

In conclusion, this is the first study on the temporal dynamics of the four canonical EEG microstates in siblings of patients with schizophrenia. Results indicate that the dynamics of resting-state EEG microstates, particularly classes C and D, is a potential endophenotype for schizophrenia. Since the dynamics of microstates can be altered by neurofeedback[45] and transcranial magnetic stimulation[46], these results open avenues for the development of new treatments for the disorder.

## Methods and materials

**General information about participants.** Participants were no older than 55 years old. All participants have participated in a previous study on masking and evoked-related potentials (ERPs). Masking and ERP data of some participants have been already published, while data of other participants have not been analyzed yet. All participants signed informed consent and were informed that they could quit the experiments at any time. All procedures complied with the Declaration of Helsinki and were approved by the Ethical Committee of Institute of Postgraduate Medical Education and Continuous Professional Development (Georgia). Protocol number: 09/07. Title: "Genetic polymorphisms and early information processing in schizophrenia".

**Participants of study 1.** Three groups of participants joined study 1: chronic patients with schizophrenia ($n = 101$), unaffected siblings of patients with schizophrenia ($n = 43$), and healthy controls ($n = 75$). Resting microstate dynamics

data of 27 patients and 27 controls have already been published in previous work[6]. Masking and ERP data of 89 patients, 39 siblings, and 63 controls have already been published[47–49]. Patients with schizophrenia and their siblings were recruited from the Tbilisi Mental Health Hospital or the psycho-social rehabilitation center. Patients participated in the study when they had recovered sufficiently from an acute psychotic episode. Thirty-one were inpatients; 70 were outpatients. Patients were diagnosed using the Diagnostic and Statistical Manual of Mental Disorders Fourth Edition (DSM-IV) by means of an interview based on the Structured Clinical Interview for DSM-IV, Clinician Version, information from staff, and study of patients' records. Psychopathology of patients with schizophrenia was assessed by an experienced psychiatrist using the SANS and SAPS. Out of the 101 patients, 88 were receiving neuroleptic medication. CPZ equivalents are indicated in Table 4. We included siblings of the patients with schizophrenia only when they had no history of psychoses. Controls were recruited from the general population, aiming to match patients and siblings as closely as possible. All siblings and controls were free from psychiatric axis I disorders. Family history of psychosis was an exclusion criterion for the control group. General exclusion criteria were alcohol or drug abuse, severe neurological incidents or diagnoses (including head injury), development disorders (autims spectrum disorder or intellectual disability), or other somatic mind-altering illnesses, assessed through interview by certified psychiatrists. Group characteristics are presented in Table 4. Since patients and controls differed in terms of gender ($X^2(1) = 35.762$, $p = 2.229e − 9$) and education ($t(174) = −3.915$, $p = 1.297e − 4$), but not in terms of age ($t(174) = 1.399$, $p = 0.164$) nor handedness ($X^2(1) = 0.030$, $p = 0.863$), gender was used as a factor while education was used as a covariate in subsequent analyses. Siblings and controls had similar characteristics: gender ($X^2(1) = 0.109$, $p = 0.741$), age ($t(116) = −1.976$, $p = 0.051$), education ($t(174) = −1.653$, $p = 0.101$), and handedness ($X^2(1) = 0.026$, $p = 0.871$).

Out of the 43 siblings, 32 were each a sibling of a single patient in the current study (hereinafter referred to as siblings_32 and patients_32). The remaining 11 siblings were siblings of patients that performed a battery of tests but did not participate in the current EEG experiment. Group characteristics of patients_32 and siblings_32 are presented in Table 4. In subsequent analyses, for each of the computed microstate parameters, the score of siblings_32 was subtracted from their patients_32 pair, resulting in a difference score (∆), which was submitted for statistical analysis.

**Participants of study 2.** Twenty-two FEP participated in the study. Masking and ERP data of 21 of them have been published in previous work[50]. FEP were recruited from the Tbilisi Mental Health Hospital or the Acute Psychiatric Departments of Multiprofile Clinics. FEP selection, exclusion criteria, and psychopathological assessment were the same as for chronic patients with schizophrenia, see "Participants of Study 1". Out of the 22 FEP, 20 were receiving neuroleptic medication: 4 were inpatients; 18 were outpatients. From our pool of 101 chronic patients with schizophrenia (see "Participants of Study 1"), we pseudo-randomly selected 22 patients (Patients_22), to match the 22 FEP as closely as possible, regarding gender ($X^2(1) = 0.026$, $p = 0.871$), age ($t(42) = −0.563$, $p = 0.576$), and education ($t(42) = −0.780$, $p = 0.440$). Group characteristics are shown in Table 5. FEP and Patients_22 groups differed only in terms of illness duration ($t(42) = −5.838$, $p = 6.786e − 7$), SANS ($t(42) = −2.271$, $p = 0.028$) and SAPS ($t(42) = −2.433$, $p = 0.019$) scores. The two groups did not significantly differ in terms of CPZ equivalent ($t(36) = −0.305$, $p = 0.762$) nor handedness ($X^2(1) = 0.000$, $p = 1.000$).

We tested the FEP group three times throughout 1 year to assess whether the microstates dynamics changed with the progression of the disease. Out of the 22 patients, 16 participated 6 months later on a second session (FEP_2). Out of these 16, 11 were tested 6 months later on a third session (FEP_3). All the 22 FEP were invited to participate in all three session, but six of them dropped out after the first session and the other five patients dropped out after the second session. At the second testing, 10 out of the 16 FEP_2 were receiving neuroleptic medication, and they were all outpatients. At the third testing, 6 out of the 11 FEP_3 were receiving neuroleptic medication, and they were all outpatients. Group characteristics of the

FEP_2 and FEP_3 patients are shown in Table 6. Subtypes of FEP diagnosis according to the DSM-IV for all three testing sessions are shown in Supplementary Table 12.

**EEG recording and data processing**. Participants were sitting in a dim lit room. They were instructed to keep their eyes closed and to relax for 5 min. Resting-state EEG was recorded before participants participated in a masking experiment and using a BioSemi Active 2 system (Biosemi) with 64 Ag-AgCl sintered active electrodes, referenced to the common mode sense electrode. The recording sampling rate was 2048 Hz. Offline data were downsampled to 128 Hz and preprocessed using an automatic pipeline (APP)[51]: filtering with a bandpass of 1–40 Hz; removal of powerline noise; re-referencing to the biweight estimate of the mean of all channels; removal and 3D spline interpolation of bad channels; removal of bad EEG periods; independent component analysis to remove eye movement-, muscular- and bad channel-related artifacts; re-referencing to common average reference. The proportion of interpolated electrodes was <5% for each participant. The amount of removed EEG periods was 5.56% ± 3.78 for patients, 4.65% ± 3.42 for siblings and 4.96% ± 3.76 for controls. A one-way ANOVA revealed nonsignificant effect of group on the amount of removed EEG periods ($F(2,216) = 0.810$, $p = 0.446$). For Patients_22 and FEP, the amount of removed EEG periods was 5.03% ± 4.53 and 3.46% ± 2.08, for each group, respectively. An independent samples t-test showed that the amount of removed EEG periods was not significantly different between Patients_22 and FEP ($t(42) = 0.145$, $p = 0.145$).

The global field power (GFP) of the preprocessed EEG data was determined for each participant. GFP is an instantaneous reference-independent measure of neuronal activity throughout the brain, and it is calculated as the standard deviation of the electrical potential across all electrodes at each time point[52]. Since EEG map topographies remain stable around the GFP peaks and these are the best representatives of the topographic maps regarding signal-to-noise ratio[38], only EEG topographies at the GFP peaks were submitted to further analysis. The GFP-reduced data were submitted to k-means clustering[12,24] to identify the most dominant topographies as classes of microstates present in the recordings. The clustering analysis was first done at the individual level and then across participants in each group. To have equal contributions of microstates per participant, each participant contributed to the group k-means clustering with his/hers four most dominant microstates. To compare our results with previous studies, we selected

four microstates for each group, and labeled them A–D according to their similarities to the previously reported microstate classes[5]. To ensure that the four selected microstates were similar across groups, we computed spatial correlation[53] for each of the four microstate classes for each pair of groups. High spatial correlation coefficients indicated that the microstate classes were similar between groups (see Supplementary Tables 1 and 8).

Subsequently, for each group, we compared the four microstates with the instantaneous scalp potential maps in each participant's artifact-correct EEG using a competitive fitting procedure. For each time point of the individual EEG, the scalp topography was compared with each microstate class using spatial correlation. The time point was labeled according to the microstate that exhibited the greatest correlation. Temporal smoothing (window (half) size = 5, strength (Besag) = 10, rejection of microstates with durations of one time frame) was applied to ensure that noise during low GFP periods did not interrupt segments of quasi-stable topographies[10]. For each subject, three per class microstate parameters were computed: mean duration, time coverage, and frequency of occurrence (occurrence). Mean duration (in ms) is the average time that a given microstate was uninterruptedly present. Time coverage (in %) is the percentage of the total analysis time spent in a given microstate. Occurrence is the mean number of times a given microstate is occurring per second. Microstates analysis was performed using Cartool[24] (version 3.70).

The temporal smoothing in the current study was different from the one performed by Tomescu et al.[6], a study with data from 27 patients and 27 controls included in our sample. As mentioned above, in the current study, we rejected microstates with durations of one time frame, while Tomescu et al. did not. This resulted in the microstate mean durations in the current study to be longer than the ones reported by Tomescu et al. However, in a subsequent work from the same group of researchers[41], rejection of microstates with durations of one time frame was applied, which led to microstate mean durations similar to the ones in current study.

We did not remove potentially truncated microstates before evaluating the microstate parameters. Since the groups did not significantly differ in the amount of EEG periods removed, this does not pose a problem in the overall group comparisons.

**Meta-analysis**. A literature search was conducted for papers published before 29 November 2019 via PubMed, to identify studies investigating EEG microstate dynamics in schizophrenia. The keywords were "schizophreni*" in conjunction with "microstate*", in order to get schizophrenia, schizophrenic, and schizophrenics as well as microstate and microstates. Furthermore, a prior meta-analysis and two reviews on EEG microstates were inspected for potentially missed studies[4,5,15]. We identified 28 relevant studies. For our meta-analysis, we selected studies according to the following criteria:

- Criterion 1. The study reported original data from a group of patients belonging to the psychosis spectrum as well as a healthy control group.
- Criterion 2. The reported sample sizes, summary statistics, or t-, F-, or p values had to be sufficiently detailed in order to compute effect sizes estimates and their variances. If the relevant information was not provided, we contacted the corresponding authors of the studies and asked for additional information. This was the case for two studies, Koenig et al.[18] and Murphy et al.[29]. For these, Thomas Koenig and Michael Murphy, authors of[18] and[29], respectively, provided the summary statistics, via e-mail.
- Criterion 3. The EEG montage employed the standard 10–20 system.
- Criterion 4. Four microstate classes (A, B, C, and D) were considered, since this is the number of microstate classes most frequently used in the literature[5].
- Criterion 5. The study was a resting-state state study, i.e., participants were not engaged in any particular task.

---

**Table 5 Group average statistics (±SD) of the FEP and Patients_22 groups.**

|  | FEP | Patients_22 |
|---|---|---|
| Gender (F/M) | 12/10 | 8/14 |
| Age (years) | 29.6 ± 9.1 | 31.3 ± 10.1 |
| Education (years) | 12.9 ± 2.5 | 13.5 ± 2.7 |
| Handedness (L/R) | 1/21 | 1/21 |
| Illness duration (years) | 0.7 ± 0.4 | 7.2 ± 5.2 |
| SANS | 7.6 ± 4.8 | 10.7 ± 4.2 |
| SAPS | 6.7 ± 3.1 | 9.0 ± 3.0 |
| CPZ equivalent[a] | 465.3 ± 312.6 | 495.0 ± 285.5 |

SANS Scale for the Assessment of Negative Symptoms, SAPS Scale for the Assessment of Positive Symptoms, CPZ chlorpromazine.
[a]Average CPZ equivalents calculated over the 20 FEP and 27 Patients_32 receiving neuroleptic medication, respectively.

---

**Table 6 Group average statistics (±SD) of the FEP_2 and FEP_3.**

|  | FEP_2 (n = 16) | FEP_3 (n = 11) |
|---|---|---|
| Gender (F/M)[a] | 10/6 | 6/5 |
| Age (years)[a] | 29.1 ± 9.1 | 29.4 ± 10.9 |
| Education (years)[a] | 12.6 ± 2.4 | 12.7 ± 2.3 |
| Handedness (L/R)[a] | 1/15 | 0/11 |
| Illness duration (months)[a] | 0.7 ± 0.4 | 0.8 ± 0.4 |
| SANS | 7.8 ± 5.3[a]; 9.2 ± 6.4[b] | 9.2 ± 5.6[a]; 9.5 ± 6.7[b]; 8.9 ± 6.8[c] |
| SAPS | 6.9 ± 2.9[a]; 6.2 ± 3.3[b] | 7.4 ± 3.4[a]; 6.2 ± 3.2[b]; 6.7 ± 2.5[c] |
| CPZ equivalent[d] | 452.5 ± 338.5[a]; 257.8 ± 358.6[b] | 452.0 ± 376.4[a]; 220.5 ± 262.2[b]; 130.8 ± 184.6[c] |

Patients with a first episode of psychosis that participated in the first and second testing sessions (FEP_2) and all the three testing sessions (FEP_3).
SANS Scale for the Assessment of Negative Symptoms, SAPS Scale for the Assessment of Positive Symptoms, CPZ chlorpromazine.
[a]Data collected during the first testing session.
[b]Data collected during the second testing session.
[c]Data collected during the third testing session.
[d]Average CPZ equivalents calculated over 16 and 11 patients in FEP_2 and FEP_3, respectively, by assigning a value of 0 to participants off neuroleptic medication.

- Criterion 6. The study reported at least one of the following three microstate parameters: mean duration, time coverage, and occurrence.

Only nine of the initial identified 28 studies met these six criteria. In addition, we included the current study (da Cruz et al.) and removed one study[6] because it consisted of a subset of participants of the current study. Hence, we included a total of nine studies in our meta-analysis. A comprehensive list of all identified studies, with a short explanation for exclusion (if applicable), is presented in Supplementary Table 13.

Apart from three studies, all the other studies reported the three relevant microstate parameters. The study by Nishida et al.[32] did not report the time coverage, the study by Giordano et al.[30] did not report the occurrence, while the study by Murphy et al.[29] only reported the mean duration (however, the time coverage and occurrence were obtained through personal correspondence). For each study, we calculated Cohen's d as the mean difference between patients and controls divided by the within group standard deviation, for each available microstate parameter and for each microstate class. For the current study (da Cruz et al.), we used the Cohen's d values reported in Table 1, which are corrected for gender and education differences. Hedges' g was calculated using Cohen's d multiplied by the coefficient J, which is a correction for small samples[54].

**Statistical analysis**. In study 1, for patients vs. controls, for each of the computed microstate parameters (mean duration, time coverage, and occurrence), we performed a three-way rm-ANOVA, with Group (patients and controls), Microstate Class (A, B, C, and D), Gender as factors and Education as a covariate. For siblings vs. controls, for each of the computed microstate parameters, we performed a two-way rm-ANOVA with Group (siblings and controls) and Microstate Class as factors. For patients_32 vs. siblings_32 pairs, their difference scores (Δ) for each microstate parameter and microstate class were submitted to a two-sided one-sample t-test against 0. For each of the three analyses (patients vs. controls, siblings vs. controls, and patients_32 vs. siblings_32), pairwise group comparisons for all microstate parameters and for each microstate class were corrected for multiple comparisons with Bonferroni–Holm for 12 comparisons (3 parameters × 4 classes).

In study 2, for FEP vs. Patients_22, for each computed microstate parameter, we performed a two-way rm-ANOVA with group and microstate class as factors. To investigate whether the computed microstates changed throughout 1 year for the FEPs, we divided the analysis in two parts. First, we analyzed the microstate parameters in the FEP_2 (FEP that completed the first and second testing session). Then, we analyzed the microstate parameters in the FEP_3 (FEP that completed all the three testing sessions). In both cases, for each computed microstate parameter, we computed a two-way rm-ANOVA with testing session (for FEP_2: first and second; for FEP_3: first, second, and third) and microstate class as factors.

For the meta-analysis, Hedges' g values were introduced as a generic effect size in the OpenMeta Analyst software (http://www.cebm.brown.edu/openmeta/) with the corresponding standard error. We used the continuous random-effect analysis with the restricted maximum likelihood method. The meta-analysis software computed the effect sizes, with 95% confidence intervals (CI) and the pooled effect size g*. Z-tests were conducted to test the significance of the of the pooled effect size g*. p values were corrected for 12 comparisons (3 microstate parameters × 4 microstate classes) using Bonferroni–Holm correction.

Where applicable, statistical tests were always two-sided. We considered a statistical test to be significant when the p value was below 0.05 after correction for multiple comparisons using Bonferroni–Holm correction ($p_{holm}$). As estimates of effect size, we report Cohen's d with 95% CI and $\eta^2$ with 90% CI.

Statistical tests were performed with JASP[55] software (version 0.12.1) and R[56] (version 3.6.1).

**Reporting summary**. Further information on research design is available in the Nature Research Reporting Summary linked to this article.

## Data availability

The data that support the findings of this study are available upon reasonable request. The source data underlying Figs. 1a–d, 2a–d, and 3a–c and Supplementary Figs. 1–12 are provided as a Source Data File. Source data are provided with this paper.

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

## Acknowledgements

This work was partially funded by the Fundação para a Ciência e a Tecnologia under grant FCT PD/BD/105785/2014 and the National Centre of Competence in Research (NCCR) Synapsy financed by the Swiss National Science Foundation under grant 51NF40-158776. We would like to thank Professor André Berchtold for assistance with statistical analysis as well as Professor Thomas Koenig and Dr. Michael Murphy for providing additional information of their manuscripts for the meta-analysis.

## Author contributions

M.H.H, C.M., A.B., and M.R. designed the research; M.R. and E.C. collected the data; J.R. C and O.F. analyzed data; J.R.C., O.F. A.B., C.M., P.F., and M.H.H. wrote the paper.

## Competing interests

The authors declare no competing interests.
