## [Peer Review File · Nature Communications]

Reviewers' comments:

Reviewer #1 (Remarks to the Author):

The submitted paper reports on a multipart study that sets out to conclusively resolve whether particular EEG microstate abnormalities that have repeatedly been in patients with schizophrenia can be considered as endophenotypes of schizophrenia. In its core findings, the obtained results replicate findings of a whole series of earlier studies on schizophrenia patients, and identify similar abnormalities in unaffected siblings of patients with schizophrenia. In addition, the paper reports that the finding remains stable, at least in a relatively small group of first-episode patients that could be followed up for 6 months. Finally they report a finding that looks like a compensatory mechanism that seems to be specific for the unaffected siblings.

The study is important, in my opinion, both from the perspective of the quality of the presented work that is both technically state of the art, and that left out no extra effort that helped making the intended point, and second, in the internal (i.e. between the sub-projects presented in the present study) and external (i.e. in relation to the existing literature) consistency of the obtained empirical results which clearly indicates that the authors are into something that is "real", relevant and new.

In terms of the EEG microstate quantification and the presented statistics, there is very little to say about, the employed procedure replicates the currently accepted, and seemingly quite successful standards established in previous research. The only issue where I was having doubts is the inclusion and exclusion of studies for the meta-analysis. While the Irisawa paper used only three microstate classes, one of these classes truly resembles the class B, so that result from the study may be included. But the Strelets paper had only 10 electrodes, and as a consequence, the map topographies are, in my honest opinion, so different from the typical microstate class maps that I don't see how these results can reasonably be integrated with the majority of the existing studies. Which of the 4 classes reported in this paper was assigned to which of the "canonical" 4 microstate classes, and on what basis was that assignment done?

Reviewer #2 (Remarks to the Author):

The authors investigated the potential value of EEG resting-state microstates as a potential endophenotype for schizophrenia, by combining results from 3 different studies (investigation of siblings of patients; comparison of first-episode with chronic patients; longitudinal investigation of first-episode patients during a 1-year period) and 1 meta-analysis.

The topic is very interesting, as the proposed endophenotype can be assessed simply and inexpensively by means of a short resting-state EEG recording, and the combination of different complementary approaches is certainly laudable. However, the broad scope of the paper comes at the cost of methodological rigor (or at least, methods are not sufficiently explained, with the result that a number of points remain unclear).

General remarks:

- The introduction and the discussion do not do justice to the complexity of the topic. The concept of microstates is only briefly presented, previous findings (e.g. networks represented by microstates) are reported in a rather simplified manner, and current debates are overlooked (e.g. the question of the optimal number of microstates).
- The issue of reliability (important for an endophenotype) is not discussed at all. For example,

microstate results can be affected by basic pre-processing steps such as filtering (see Michel & Koenig, 2018).

- The method of the three patient studies is not described in sufficient detail: e.g. how were patients, siblings etc. recruited? Were neurological disorders or craniocerebral trauma an exclusion criterion, and how was this assessed? How were diagnoses (or the lack of a diagnosis in siblings and healthy controls) confirmed? What were diagnoses in FEP (a diagnosis of schizophrenia requires a certain follow-up observation time)? Regarding microstate calculation, were potentially truncated microstates (at the beginning and the end of each epoch) removed before evaluating microstate parameters?
- There is a lot of repetition throughout the manuscript, in particular a lot of overlap between methods and results sections.

Study 1:

- The sibling sample size is much smaller than patient sample size. Apparently, some patients were tested along with (at least) one sibling, while others were not. That makes statistical comparisons challenging.
- On a related note: observations in patient-sibling pairs are not completely independent from one another, which does not appear to have been taken into account in analyses.
- The majority of patients were medicated. This makes interpretation of results challenging, especially regarding microstate B (the authors interpret microstate B increase in siblings as indicating some compensation mechanism, but it could theoretically just as well be the case that 'normal' microstate B in patients represents a medication effect). Moreover, the authors do not discuss their results compared to previous studies, most of which have investigated unmedicated, acutely ill patients. Only two studies assessed medicated patients. Of those, a study by Tomescu et al. in a (probably) overlapping chronic patient sample with the current study found similar microstate profiles in patients, but another by Andreou et al. in a FEP sample reported partially different results.
- Compared to the study by Tomescu et al., average microstate duration is longer in the present study. This is in itself not problematic, unless there is indeed overlap with the current patient sample, in which case the difference is difficult to explain without assuming that the authors made changes in their pre- or postprocessing analysis pipeline. Please clarify.
- The authors should discuss symptom severity in their sample compared to previous studies – most previous studies investigated acutely ill patients.
- Minor remark, as it will not affect significance levels in most cases: It is not clear how Bonferroni-Holm was applied; were results corrected for 4 microstates or 4 ms x 3 parameters each?

Studies 2 and 3:

- Negative results are rather difficult to assess given the small sample sizes; it might be useful to present power calculations.
- Explained variance in FEP seems low compared to study 1, would the authors like to comment on that?
- 50% attrition in study 3 is problematic even for linear mixed models (and the final sample size of 11 is quite small)
- Again, symptom severity and medication status are not commented on.

Meta-analysis:

- The authors do not state anywhere how they conducted the search, criteria of study inclusion etc. It is not clear, for example, why they included two older studies that had been excluded from the meta-analysis by Rieger et al.
- Related to the above: The term 'update' and the lack of methodological details makes me think that the authors did not conduct an independent search, but rather used the search results reported in the paper by Rieger et al. It might be advisable to avoid giving that impression, since none of the authors of the original meta-analysis were involved in the present study.

General Comments:

Dear anonymous reviewers, thank you very much for your comments and suggestions that improved the quality of the paper. While the paper was under review, we collected data from 12 schizophrenia patients, 5 unaffected siblings, and 6 healthy controls. We added these data to the revised manuscript. The added data did neither change the results nor the conclusions.

Reviewers' comments:

Reviewer #1 (Remarks to the Author):

The submitted paper reports on a multipart study that sets out to conclusively resolve whether particular EEG microstate abnormalities that have repeatedly been in patients with schizophrenia can be considered as endophenotypes of schizophrenia. In its core findings, the obtained results replicate findings of a whole series of earlier studies on schizophrenia patients, and identify similar abnormalities in unaffected siblings of patients with schizophrenia. In addition, the paper reports that the finding remains stable, at least in a relatively small group of first-episode patients that could be followed up for 6 months. Finally they report a finding that looks like a compensatory mechanism that seems to be specific for the unaffected siblings.

The study is important, in my opinion, both from the perspective of the quality of the presented work that is both technically state of the art, and that left out no extra effort that helped making the intended point, and second, in the internal (i.e. between the sub-projects presented in the present study) and external (i.e. in relation to the existing literature) consistency of the obtained empirical results which clearly indicates that the authors are into something that is "real", relevant and new.

In terms of the EEG microstate quantification and the presented statistics, there is very little to say about, the employed procedure replicates the currently accepted, and seemingly quite successful standards established in previous research. The only issue where I was having doubts is the inclusion and exclusion of studies for the meta-analysis. While the Irisawa paper used only three microstate classes, one of these classes truly resembles the class B, so that result from the study may be included. But the Strelets paper had only 10 electrodes, and as a consequence, the map topographies are, in my honest opinion, so different from the typical microstate class maps that I don't see how these results can reasonably be integrated with the majority of the existing studies. Which of the 4 classes reported in this paper was assigned to which of the "canonical" 4 microstate classes, and on what basis was that assignment done?

Response:

We thank the reviewer for the very positive comments on the manuscript.

We also thank the reviewer for the comment regarding the meta-analysis. We agree with the reviewer and we have now limited our meta-analysis to papers reporting the 4 microstate classes and the 10-20 system. We have also added to the manuscript the information about how we

conducted the literature research; our criteria for inclusion and exclusion of a study are as follows:

4. Methods and Materials

4.3. Meta-analysis:

A literature search was conducted for papers published before 29th November 2019 via PubMed, to identify studies investigating EEG microstates dynamics in schizophrenia. The key words were ‘schizophreni*’ in conjunction with ‘microstate*’, in order to get schizophrenia, schizophrenic, and schizophrenics as well as microstate and microstates. Furthermore, a prior meta-analysis and two reviews on EEG microstates were inspected for potentially missed studies (4,5,15). We identified 28 relevant studies. For our meta-analysis, we selected studies according to the following criteria:

- Criterion 1. The study reported original data from a group of patients belonging to the psychosis spectrum as well as a healthy control group.
- Criterion 2. The reported sample sizes, summary statistics, or *t*-, *F*-, or *p*-values had to be sufficiently detailed in order to compute effect sizes estimates and their variances. If the relevant information was not provided, we contacted the corresponding authors of the studies and asked for additional information. This was the case for 2 studies, (18) and (29): for these, Thomas Koenig and Michael Murphy, authors of (18) and (29), respectively, provided the summary statistics, via e-mail.
- Criterion 3. The EEG montage employed the standard 10-20 system.
- Criterion 4. Four microstate classes (A, B, C, and D) were considered, since this is the number of microstate classes most frequently used in the literature (5).
- Criterion 5. The study was a resting-state state study, i.e., participants were not engaged in any particular task.
- Criterion 6. The study reported at least one of the following 3 microstate parameters: mean duration, time coverage, and occurrence.

Only 9 of the initial identified 28 studies met these 6 criteria. In addition, we included the current study (da Cruz et al.) and removed one study (6) because it consisted of a subset of participants of the current study. Hence, we included a total of 9 studies in our meta-analysis. A comprehensive list of all identified studies, with a short explanation for exclusion (if applicable), is presented in **Table S12** in **Supplementary Information**.

Apart from 3 studies, all the other studies reported the 3 relevant microstate parameters. The study by Nishida et al. (32) did not report the time coverage, the study by Giordano and colleagues (30) did not report the occurrence, while the study by Murphy et al. (29) only reported the mean duration (however, the time coverage and occurrence were obtained through personal correspondence). For each study, we calculated Cohen’s *d* as the mean difference between patients and controls divided by the within group standard deviation, for each available microstate parameter and for each microstate class. For the current study (da Cruz et al.), we used the Cohen’s *d* values reported in **Table 1**, which are corrected for gender and education differences. Hedges’ *g* was calculated using Cohen’s *d* multiplied by the coefficient *J*, which is a correction for small samples (52).

Hedges’ *g* values were introduced as a generic effect size in the OpenMeta Analyst software (<http://www.cebm.brown.edu/openmeta/>) with the corresponding standard error (SE). We used the continuous random-effect analysis with the Restricted Maximum Likelihood (REML)

method. The meta-analysis software computed the effect sizes, with 95% confidence intervals (C.I.) and the pooled effect size g^* . P -values were corrected for 12 comparisons (3 microstate parameters \times 4 microstate classes) using Bonferroni-Holm correction.

Table 1 - Pairwise group comparisons for all microstate parameters (mean duration, time coverage, and occurrence) and for each microstate class (A, B, C, and D). Statistically significant differences, after Bonferroni-Holm correction, are indicated in bold.

Comparison	Microstate Class	Mean Duration	Time Coverage	Occurrence
Patients vs. Controls	A	$p=0.054, p_{holm}=0.270, d=-0.293$	$p=0.449, p_{holm}=0.898, d=-0.110$	$p=0.882, p_{holm}=0.898, d=0.023$
	B	$p=0.003, p_{holm}=0.018, d=-0.454$	$p=0.074, p_{holm}=0.296, d=-0.271$	$p=0.112, p_{holm}=0.336, d=-0.247$
	C	$p=1.315e-4, p_{holm}=0.001, d=0.590$	$p=1.452e-7, p_{holm}=1.742e-7, d=0.827$	$p=1.170e-4, p_{holm}=0.001, d=0.602$
	D	$p=3.010e-6, p_{holm}=3.311e-5, d=-0.732$	$p=3.445e-6, p_{holm}=3.445e-5, d=-0.725$	$p=1.620e-4, p_{holm}=0.001, d=-0.578$
Siblings vs. Controls	A	$p=0.055, p_{holm}=0.288, d=-0.371$	$p=0.097, p_{holm}=0.288, d=-0.320$	$p=0.250, p_{holm}=0.288, d=0.221$
	B	$p=0.049, p_{holm}=0.288, d=0.381$	$p=0.048, p_{holm}=0.288, d=0.382$	$p=0.069, p_{holm}=0.288, d=0.351$
	C	$p=0.022, p_{holm}=0.154, d=0.445$	$p=0.001, p_{holm}=0.010, d=0.631$	$p=0.006, p_{holm}=0.048, d=0.533$
	D	$p=3.380e-4, p_{holm}=0.004, d=-0.707$	$p=1.465e-4, p_{holm}=0.002, d=-0.751$	$p=0.004, p_{holm}=0.036, d=-0.566$
Patients_32 vs. Siblings_32	A	$p=0.235, p_{holm}=1.000, d=0.214$	$p=0.020, p_{holm}=0.200, d=0.434$	$p=0.014, p_{holm}=0.154, d=0.462$
	B	$p=0.003, p_{holm}=0.036, d=-0.576$	$p=0.069, p_{holm}=0.621, d=-0.333$	$p=0.200, p_{holm}=1.000, d=-0.231$
	C	$p=0.838, p_{holm}=1.000, d=-0.036$	$p=0.783, p_{holm}=1.000, d=0.049$	$p=0.588, p_{holm}=1.000, d=0.097$
	D	$p=0.270, p_{holm}=1.000, d=-0.199$	$p=0.491, p_{holm}=1.000, d=-0.123$	$p=0.772, p_{holm}=1.000, d=-0.052$

Table S12 - List of studies identified during the literature search and information whether it was included or excluded from the meta-analysis.

N	Study ID	Excluded	Exclusion Reason	Population
1	Koenig et al., 1999	no		Schizophrenia
2	Lehmann et al., 2005	no		Schizophrenia
3	Kikuchi et al., 2007	no		Schizophrenia
4	Nishida et al., 2013	no		Schizophrenia
5	Andreou et al., 2014	no		FEP
6	Tomescu et al., 2015	yes	same participants as in da Cruz et al., current	Schizophrenia
7	Tomescu et al., 2014	no		22q11
8	Irisawa et al., 2006	yes	3 classes	Schizophrenia
9	Strelets et al., 2003	yes	not 10-20 system, low n of electrodes	Schizophrenia
10	Giordano et al., 2018	no		Schizophrenia
11	Murphy et al., 2019	no		FEP
12	Soni et al., 2019	yes	task-related	Schizophrenia
13	Soni et al., 2018	yes	5 classes	Schizophrenia
14	Sverak et al., 2018	yes	5 classes and TMS	Schizophrenia
15	Rieger et al., 2016	yes	meta-analysis, no original data	
16	Diaz Hernandez et al., 2016	yes	no schizophrenia patients + neurofeedback	Healthy
17	Khanna et al., 2015	yes	review, no original data	
18	Schlegel et al., 2012	yes	personality traits, skeptical versus believer	Healthy
19	Kindler et al., 2011	yes	no control group, hallucination	Schizophrenia
20	Mucci et al., 2005	yes	schizotypy	Healthy
21	Stevens et al., 1997	yes	task-related and only one microstate	Schizophrenia
22	Kleinlogel et al., 2007	yes	task-related	Schizophrenia
23	Katayama et al., 2007	yes	no schizophrenia patients, hypnosis	Healthy
24	Yoshimura et al., 2007	yes	healthy participants and drugs	Healthy
25	Kochi et al., 1996	yes	Evoked-related potentials	Schizophrenia
26	Begré et al., 2008	yes	task (working memory)	Schizophrenia
27	Stirk et al., 1995	yes	no schizophrenia patients	Depressive
28	Michel and Koenig, 2018	yes	review, no original data	
29	da Cruz et al., current	no		Schizophrenia

2. Results

2.3. Meta-analysis:

Our literature search identified 8 independent studies comparing the resting-state dynamics of the 4 *canonical* microstate classes in patients belonging to the schizophrenia spectrum against a control group (18,23,29–34). In addition to these 8 studies, we also included the current study in the meta-analysis. Forest plots of the mean effect sizes for each microstate parameter (mean duration, time coverage, and occurrence) and microstate class (A, B, C, and D) are shown in the Supplementary Figure S1 - Figure S12. Similar to Rieger and colleagues (15), we found consistently increased time coverage ($g=0.447$, $p=6.304e-5$, $p_{holm}=6.934e-4$) and occurrence ($g=0.688$, $p=2.430e-13$, $p_{holm}=2.916e-12$) of microstate class C in patients compared to controls, as well as decreased time coverage ($g=-0.506$, $p=0.003$, $p_{holm}=0.027$) and mean duration ($g=-0.540$, $p=7.170e-4$, $p_{holm}=0.007$) of microstate class D in patients compared to controls. We also found a decreased mean duration of microstate class B in patients compared to controls;

however, the effect was not significant after correction for multiple comparisons ($g=-0.353$, $p=0.017$, $p_{holm}=0.136$). No consistent group differences were found for microstate class A.

Figure S1 - Forest plot of studies considering the mean duration of microstate class A ($N=685$, $k=9$, $g=-0.232$, $p=0.140$, $p_{holm}=0.889$). Results show no consistent group differences between patients and controls. (Heterogeneity $I^2=68\%$, $p=0.002$).

Figure S2 - Forest plot of studies considering the time coverage of microstate class A ($N=647$, $k=8$, $g=0.004$, $p=0.980$, $p_{holm}=1.000$). Results show no consistent group differences between patients and controls. (Heterogeneity $I^2=65\%$, $p=0.006$).

Figure S3 - Forest plot of studies considering the occurrence of microstate class A (N=479, k=8, $g=0.318$, $p=0.127$, $p_{holm}=0.889$). Results show no consistent group differences between patients and controls. (Heterogeneity $I^2=74\%$, $p=0.0003$).

Figure S4 - Forest plot of studies considering the mean duration of microstate class B (N=685, k=9, $g=-0.353$, $p=0.017$, $p_{holm}=0.136$). We found that patients have shorter microstate class B mean durations than controls; however, the result was not significant after correction for multiple comparisons. (Heterogeneity $I^2=68\%$, $p=0.002$).

Figure S5 - Forest plot of studies considering the time coverage of microstate class B (N=647, k=8, $g=-0.048$, $p=0.754$, $p_{holm}=1.000$). Results show no consistent group differences between patients and controls. (Heterogeneity $I^2=62\%$, $p=0.010$).

Figure S6 - Forest plot of studies considering the occurrence of microstate class B (N=479, k=8, $g=0.148$, $p=0.378$, $p_{holm}=1.000$). Results show no consistent group differences between patients and controls. (Heterogeneity $I^2=66\%$, $p=0.005$).

Figure S7 - Forest plot of studies considering the duration of microstate class C (N=685, k=9, $g=0.078$, $p=0.611$, $p_{holm}=1.000$). Results show no consistent group differences between patients and controls. (Heterogeneity $I^2=69\%$, $p=0.001$).

Figure S8 - Forest plot of studies considering the time coverage of microstate class C (N=647, k=8, $g=0.447$, $p=6.304e-5$, $p_{holm}=6.934e-4$). We found that patients have significantly longer microstate class C time coverage than controls. (Heterogeneity $I^2=35\%$, $p=0.148$).

Figure S9 - Forest plot of studies considering the occurrence of microstate class C (N=479, k=8, $g=0.688$, $p=2.430e-13$, $p_{holm}=2.916e-12$). We found that microstate class C occurs significantly more in patients than controls. (Heterogeneity $I^2=8\%$, $p=0.367$).

Figure S10 - Forest plot of studies considering the mean duration of microstate class D (N=685, k=9, $g=-0.540$, $p=7.170e-4$, $p_{holm}=0.007$). We found that patients have significantly shorter microstate class B mean durations than controls. (Heterogeneity $I^2=73\%$, $p=0.0003$).

Figure S11 - Forest plot of studies considering the time coverage of microstate class D (N=647, k=8, $g=-0.506$, $p=0.003$, $p_{holm}=0.027$). We found that patients have significantly shorter microstate class D time coverage than controls. (Heterogeneity $I^2=68\%$, $p=0.002$).

Figure S12 - Forest plot of studies considering the occurrence of microstate class D (N=479, k=8, $g=-0.201$, $p=0.228$, $p_{holm}=1.000$). Results show no consistent group differences between patients and controls. (Heterogeneity $I^2=65\%$, $p=0.005$).

References:

4. Khanna A, Pascual-Leone A, Michel CM, Farzan F (2015): Microstates in resting-state EEG: Current status and future directions. *Neuroscience & Biobehavioral Reviews* 49: 105–113.
5. Michel CM, Koenig T (2018): EEG microstates as a tool for studying the temporal dynamics of whole-brain neuronal networks: A review. *NeuroImage* 180: 577–593.

6. Tomescu MI, Rihs TA, Roinishvili M, Karahanoglu FI, Schneider M, Menghetti S, *et al.* (2015): Schizophrenia patients and 22q11.2 deletion syndrome adolescents at risk express the same deviant patterns of resting state EEG microstates: A candidate endophenotype of schizophrenia. *Schizophrenia Research: Cognition* 2: 159–165.
15. Rieger K, Diaz Hernandez L, Baenninger A, Koenig T (2016): 15 Years of Microstate Research in Schizophrenia – Where Are We? A Meta-Analysis. *Front Psychiatry* 7. <https://doi.org/10.3389/fpsy.2016.00022>
18. Koenig T, Lehmann D, Merlo MCG, Kochi K, Hell D, Koukkou M (1999): A deviant EEG brain microstate in acute, neuroleptic-naïve schizophrenics at rest. *European Archives of Psychiatry and Clinical Neurosciences* 249: 205–211.
23. Tomescu MI, Rihs TA, Becker R, Britz J, Custo A, Grouiller F, *et al.* (2014): Deviant dynamics of EEG resting state pattern in 22q11.2 deletion syndrome adolescents: A vulnerability marker of schizophrenia? *Schizophrenia Research* 157: 175–181.
29. Murphy M, Stickgold R, Öngür D (2019): Electroencephalogram Microstate Abnormalities in Early-Course Psychosis. *Biological Psychiatry: Cognitive Neuroscience and Neuroimaging*. <https://doi.org/10.1016/j.bpsc.2019.07.006>
30. Giordano GM, Koenig T, Mucci A, Vignapiano A, Amodio A, Di Lorenzo G, *et al.* (2018): Neurophysiological correlates of Avolition-apathy in schizophrenia: A resting-EEG microstates study. *NeuroImage: Clinical* 20: 627–636.
31. Andreou C, Faber PL, Leicht G, Schoettle D, Polomac N, Hanganu-Opatz IL, *et al.* (2014): Resting-state connectivity in the prodromal phase of schizophrenia: Insights from EEG microstates. *Schizophrenia Research* 152: 513–520.
32. Nishida K, Morishima Y, Yoshimura M, Isotani T, Irisawa S, Jann K, *et al.* (2013): EEG microstates associated with salience and frontoparietal networks in frontotemporal dementia, schizophrenia and Alzheimer’s disease. *Clinical Neurophysiology* 124: 1106–1114.
33. Kikuchi M, Koenig T, Wada Y, Higashima M, Koshino Y, Strik W, Dierks T (2007): Native EEG and treatment effects in neuroleptic-naïve schizophrenic patients: Time and frequency domain approaches. *Schizophrenia Research* 97: 163–172.
34. Lehmann D, Faber PL, Galderisi S, Herrmann WM, Kinoshita T, Koukkou M, *et al.* (2005): EEG microstate duration and syntax in acute, medication-naïve, first-episode schizophrenia: a multi-center study. *Psychiatry Research: Neuroimaging* 138: 141–156.
52. Francis G (2017): Equivalent statistics and data interpretation. *Behav Res* 49: 1524–1538.

Reviewer #2 (Remarks to the Author):

The authors investigated the potential value of EEG resting-state microstates as a potential endophenotype for schizophrenia, by combining results from 3 different studies (investigation of siblings of patients; comparison of first-episode with chronic patients; longitudinal investigation of first-episode patients during a 1-year period) and 1 meta-analysis.

The topic is very interesting, as the proposed endophenotype can be assessed simply and inexpensively by means of a short resting-state EEG recording, and the combination of different complementary approaches is certainly laudable. However, the broad scope of the paper comes at the cost of methodological rigor (or at least, methods are not sufficiently explained, with the result that a number of points remain unclear).

Response:

Thank you for your positive comments on the manuscript.

General remarks:

- The introduction and the discussion do not do justice to the complexity of the topic. The concept of microstates is only briefly presented, previous findings (e.g. networks represented by microstates) are reported in a rather simplified manner, and current debates are overlooked (e.g. the question of the optimal number of microstates).

Response:

We thank the reviewer for the comment. We have reformulated the introduction and the discussion in order to mention some unresolved issues with the microstates topic as follows:

Introduction:

EEG microstates are highly reproducible, both within and across participants (8). This allows the use of clustering algorithms to group microstates into a finite set of classes based on their topographical similarity (9,10). Even though there is still no general consensus on how to determine the optimal number of microstate classes (5,10–13) and the optimal number of classes of microstates may depend on the dataset analyzed (13,14), in clinical research, usually, four classes of microstates, labeled A, B, C, and D, are used (15) based on pioneering work (16–18). These four dominant classes of microstates are consistently observed in resting-state EEG (independently of the number of electrodes, and group of participants), explaining 65–84% of the global variance of the data (5). Here, we used these four *canonical* classes of microstates because this allows comparison between studies.

Several studies have attempted to identify the brain sources underlying these classes of microstates (13,16,19–22). A direct comparison of the findings is difficult due to the differences in data acquisition and processing as well as the number of microstate classes used and the way the microstates analyses were performed. Nonetheless, these studies indicate that EEG microstates are closely related to resting-state networks (RSNs) commonly found in resting-state functional magnetic resonance imaging (fMRI) despite the different time resolutions of the two modalities (see (5) for a review). Among the above-mentioned studies, the one by Britz *et al.*

(16) is usually referred to when discussing fMRI correlates of EEG microstates since it used an approach more closely related to the conventional EEG microstate analysis. Britz and colleagues found that microstate class A was associated with the auditory RSN, microstate class B with the visual RSN, microstate class C with the salience RSN, and microstate class D with the attention RSN. Similar results were found using source localization of the EEG microstate classes (13), providing further evidence for these associations. However, since both fMRI RSNs and EEG microstates are still lacking a full understanding of their significance in terms of the functional organization of brain networks, one should be cautious when interpreting these associations.

Discussion:

Practicability: resting-state EEG is easily recorded in a 5 minutes session, and EEG montages with as low as 19 electrodes can be used for microstates analysis (38). Moreover, even though differences in EEG pre-processing and temporal smoothing parameters might influence the results of microstate analysis, it has been shown that microstate analysis has a high test-retest reliability, independently of the clustering algorithms applied and the number of electrodes used (8).

References:

5. Michel CM, Koenig T (2018): EEG microstates as a tool for studying the temporal dynamics of whole-brain neuronal networks: A review. *NeuroImage* 180: 577–593.
8. Khanna A, Pascual-Leone A, Farzan F (2014): Reliability of Resting-State Microstate Features in Electroencephalography ((T. Koenig, editor)). *PLoS ONE* 9: e114163.
9. Wackermann J, Lehmann D, Michel CM, Strik WK (1993): Adaptive segmentation of spontaneous EEG map series into spatially defined microstates. *International Journal of Psychophysiology* 14: 269–283.
10. Pascual-Marqui RD, Michel CM, Lehmann D (1995): Segmentation of brain electrical activity into microstates: model estimation and validation. *IEEE Transactions on Biomedical Engineering* 42: 658–665.
11. Milligan GW, Cooper MC (1985): An examination of procedures for determining the number of clusters in a data set. *Psychometrika* 50: 159–179.
12. Murray MM, Brunet D, Michel CM (2008): Topographic ERP Analyses: A Step-by-Step Tutorial Review. *Brain Topogr* 20: 249–264.
13. Custo A, Van De Ville D, Wells WM, Tomescu MI, Brunet D, Michel CM (2017): Electroencephalographic Resting-State Networks: Source Localization of Microstates. *Brain Connectivity* 7: 671–682.
14. Seitzman BA, Abell M, Bartley SC, Erickson MA, Bolbecker AR, Hetrick WP (2017): Cognitive manipulation of brain electric microstates. *NeuroImage* 146: 533–543.

15. Rieger K, Diaz Hernandez L, Baenninger A, Koenig T (2016): 15 Years of Microstate Research in Schizophrenia – Where Are We? A Meta-Analysis. *Front Psychiatry* 7. <https://doi.org/10.3389/fpsy.2016.00022>
16. Britz J, Van De Ville D, Michel CM (2010): BOLD correlates of EEG topography reveal rapid resting-state network dynamics. *NeuroImage* 52: 1162–1170.
17. Brodbeck V, Kuhn A, von Wegner F, Morzelewski A, Tagliazucchi E, Borisov S, *et al.* (2012): EEG microstates of wakefulness and NREM sleep. *NeuroImage* 62: 2129–2139.
18. Koenig T, Lehmann D, Merlo MCG, Kochi K, Hell D, Koukkou M (1999): A deviant EEG brain microstate in acute, neuroleptic-naive schizophrenics at rest. *European Archives of Psychiatry and Clinical Neurosciences* 249: 205–211.
19. Musso F, Brinkmeyer J, Mobascher A, Warbrick T, Winterer G (2010): Spontaneous brain activity and EEG microstates. A novel EEG/fMRI analysis approach to explore resting-state networks. *NeuroImage* 52: 1149–1161.
20. Yuan H, Zotev V, Phillips R, Drevets WC, Bodurka J (2012): Spatiotemporal dynamics of the brain at rest — Exploring EEG microstates as electrophysiological signatures of BOLD resting state networks. *NeuroImage* 60: 2062–2072.
21. Pascual-Marqui RD, Lehmann D, Faber P, Milz P, Kochi K, Yoshimura M, *et al.* (2014): The resting microstate networks (RMN): cortical distributions, dynamics, and frequency specific information flow. *arXiv:14111949 [q-bio]*. Retrieved November 16, 2018, from <http://arxiv.org/abs/1411.1949>
22. Milz P, Pascual-Marqui RD, Achermann P, Kochi K, Faber PL (2017): The EEG microstate topography is predominantly determined by intracortical sources in the alpha band. *NeuroImage* 162: 353–361.
38. Koenig T, Prichep L, Lehmann D, Sosa PV, Braeker E, Kleinlogel H, *et al.* (2002): Millisecond by Millisecond, Year by Year: Normative EEG Microstates and Developmental Stages. *NeuroImage* 16: 41–48.

- The issue of reliability (important for an endophenotype) is not discussed at all. For example, microstate results can be affected by basic pre-processing steps such as filtering (see Michel & Koenig, 2018).

Response:

Thank you for the suggestion. We now mention the reliability of microstates analysis in the Discussion as follows:

Practicability: resting-state EEG is easily recorded in a 5 minutes session, and EEG montages with as low as 19 electrodes can be used for microstates analysis (38). Moreover, even though differences in EEG pre-processing and temporal smoothing parameters might influence the

results of microstate analysis, it has been shown that microstate analysis has a high test-retest reliability, independently of the clustering algorithms applied and the number of electrodes used (8).

References:

8. Khanna A, Pascual-Leone A, Farzan F (2014): Reliability of Resting-State Microstate Features in Electroencephalography. *PLoS ONE* 9: e114163.

38. Koenig T, Prichep L, Lehmann D, Sosa PV, Braeker E, Kleinlogel H, *et al.* (2002): Millisecond by Millisecond, Year by Year: Normative EEG Microstates and Developmental Stages. *NeuroImage* 16: 41–48.

- The method of the three patient studies is not described in sufficient detail: e.g. how were patients, siblings etc. recruited? Were neurological disorders or craniocerebral trauma an exclusion criterion, and how was this assessed? How were diagnoses (or the lack of a diagnosis in siblings and healthy controls) confirmed? What were diagnoses in FEP (a diagnosis of schizophrenia requires a certain follow-up observation time)? Regarding microstate calculation, were potentially truncated microstates (at the beginning and the end of each epoch) removed before evaluating microstate parameters?

Response:

We thank the reviewer for the pertinent questions regarding the participants. We have now rewritten the information about the participants in the “Methods and Materials” as well as in the “Supplementary Information” to provide a more complete description of the participants’ recruitment, assessment and characteristics.

4. Methods and Materials

4.1. Participants

Participants were no older than 55 years old. All participants have participated in a previous study on masking and evoked-related potentials (ERPs). Masking and ERP data of some participants have been already published, while data of other participants have not been analyzed yet. All participants signed informed consent and were informed that they could quit the experiments at any time. All procedures complied with the Declaration of Helsinki and were approved by the local ethics committee.

4.1.1. Study 1: Patients, Siblings, and Controls

Three groups of participants joined study 1: chronic schizophrenia patients (n=101), unaffected siblings of schizophrenia patients (n=43), and healthy controls (n=75). Resting microstate dynamics data of 27 patients and 27 controls have already been published in previous work (6). Masking and ERP data of 89 patients, 39 siblings, and 63 controls have already been published (47). Schizophrenia patients and their siblings were recruited from the Tbilisi Mental Health Hospital or the psycho-social rehabilitation center.

Patients participated in the study when they had recovered sufficiently from an acute psychotic episode. 31 were inpatients; 70 were outpatients. Patients were diagnosed using the *Diagnostic and Statistical Manual of Mental Disorders Fourth Edition* (DSM-IV) by means of an interview based on the SCID-CV (Structured Clinical Interview for DSM-IV, Clinician Version), information from staff, and study of patients' records. Psychopathology of schizophrenia patients was assessed by an experienced psychiatrist using the Scales for the Assessment of Negative Symptoms (SANS) and Scales for the Assessment of Positive Symptoms (SAPS). 88 out of the 101 patients were receiving neuroleptic medication. Chlorpromazine equivalents are indicated in **Table 2**. We included siblings of the schizophrenia patients only when they had no history of psychoses. Controls were recruited from the general population, aiming to match patients and siblings as closely as possible. All siblings and controls were free from psychiatric axis I disorders. Family history of psychosis was an exclusion criterion for the control group. General exclusion criteria were alcohol or drug abuse, severe neurological incidents or diagnoses (including head injury), development disorders (autism spectrum disorder or intellectual disability) or other somatic mind-altering illnesses, assessed through interview by certified psychiatrists. Group characteristics are presented in **Table 2**. Since patients and controls differed in terms of gender and education, gender was used as a factor while education was used as a covariate in subsequent analyses.

32 out of the 43 siblings were each a sibling of a single patient in the current study (hereinafter referred to as siblings_32 and patients_32). The remaining 11 siblings were siblings of patients that performed a battery of tests but did not participate in the current EEG experiment. Group characteristics of patients_32 and siblings_32 are presented in **Table 2**. In subsequent analyses, for each of the computed microstate parameters, the score of siblings_32 was subtracted from their patients_32 pair, resulting in a difference score (Δ), which was submitted for statistical analysis.

Table 2 - Group average statistics (\pm SD) of Schizophrenia Patients, their Unaffected Siblings, Healthy Controls, Patients_32, and Siblings_32

	Patients	Siblings	Controls	Patients_32	Siblings_32	Patients vs. Controls	Siblings vs. Controls	Patients_32 vs. Siblings_32
Gender (F/M)	11 / 90	21 / 22	39 / 36	4 / 28	14 / 18	$p=2.229e-9$	$p=0.741$	$p=0.005$
Age (years)	36.9 ± 8.8	31.8 ± 10.4	35.1 ± 7.7	33.6 ± 9.1	31.9 ± 9.6	$p=0.164$	$p=0.101$	$p=0.471$
Education (years)	13.4 ± 2.7	14.1 ± 3.0	15.1 ± 2.9	13.6 ± 2.7	14.4 ± 3.0	$p=1.297e-4$	$p=0.051$	$p=0.259$
Handedness (L/R)	6 / 95	2 / 41	4 / 71	3 / 29	2 / 30	$p=0.863$	$p=0.871$	$p=0.641$
Illness duration (years)	12.8 ± 8.2			9.4 ± 7.4				
SANS	10.4 ± 5.2			10.2 ± 5.3				
SAPS	9.7 ± 7.7			8.9 ± 3.5				
CPZ ¹	577.2 ± 400.9			541.4 ± 375.9				

Abbreviations: SANS, Scale for the Assessment of Negative Symptoms; SAPS, Scale for the Assessment of Positive Symptoms; CPZ, Chlorpromazine equivalent

¹Average CPZ equivalents calculated over the 88 Patients and 27 Patients_32 receiving neuroleptic medication, respectively.

4.1.2. Study 2: FEP

22 FEP participated in the study. Masking and ERP data of 21 of them have been published in previous work (48). FEP were recruited from the Tbilisi Mental Health Hospital or the Acute Psychiatric Departments of Multiprofile Clinics. FEP selection, exclusion criteria, and psychopathological assessment were the same as for chronic schizophrenia patients, see sub-section 4.1.1. 20 out of the 22 FEP were receiving neuroleptic medication. 4 were inpatients; 18 were outpatients. From our pool of 101 chronic schizophrenia patients (see sub-section 4.1.1.), we pseudo-randomly selected 22 patients (Patients_22), to match the 22 FEP as closely as possible, regarding gender, age, and education. Group characteristics are shown in **Table 3**. FEP and Patients_22 groups differed only in terms of illness duration, SANS and SAPS scores.

Table 3 – Group average statistics (\pm SD) of the FEP and Patients_22 groups

	FEP	Patients_22	Statistics
Gender (F/M)	12/10	8/14	$p=0.226$
Age (years)	29.6 ± 9.1	31.3 ± 10.1	$p=0.576$
Education (years)	12.9 ± 2.5	13.5 ± 2.7	$p=0.440$
Handedness (L/R)	1/21	1/21	$p=1.000$
Illness duration (years)	0.7 ± 0.4	7.2 ± 5.2	$p=6.786e-7$
SANS	7.6 ± 4.8	10.7 ± 4.2	$p=0.028$
SAPS	6.7 ± 3.1	9.0 ± 3.0	$p=0.019$
CPZ ¹	465.3 ± 312.6	495.0 ± 285.5	$p=0.762$

Abbreviations: SANS, Scale for the Assessment of Negative Symptoms; SAPS, Scale for the Assessment of Positive Symptoms; CPZ, Chlorpromazine equivalent

¹Average CPZ equivalents calculated over the 20 FEP and 27 Patients_32 receiving neuroleptic medication, respectively.

We tested the FEP group three times throughout one year to assess whether the microstates dynamics changed with the progression of the disease. Out of the 22 patients, 16 participated six months later on a second session (FEP_2). Out of these 16, 11 were tested six months later on a third session (FEP_3). All the 22 FEP were invited to participate in all three session but six of them dropped out after the first session and the other five patients dropped out after the second session. At the second testing, 10 out of the 16 FEP_2 were receiving neuroleptic medication, and they were all outpatients. At the third testing, 6 out of the 11 FEP_3 were receiving neuroleptic medication, and they were all outpatients. Group characteristics of the FEP_2 and FEP_3 patients are shown in **Table 4**. Subtypes of FEP diagnosis according to the DSM-IV for all three testing sessions are shown in **Supplementary Table S11**.

Table 4 - Group average statistics (\pm SD) of the patients with a first episode of psychosis that participated in the first and second testing sessions (FEP_2), and all the three testing sessions (FEP_3).

	FEP_2 (n = 16)		FEP_3 (n = 11)				
Gender (F/M) ¹	10/6		6/5				
Age (years) ¹	29.1 \pm 9.1		29.4 \pm 10.9				
Education (years) ¹	12.6 \pm 2.4		12.7 \pm 2.3				
Handedness (L/R) ¹	1/15		0/11				
Illness duration (months) ¹	0.7 \pm 0.4		0.8 \pm 0.4				
	Testing Session		Statistics	Testing Session			Statistics
	First	Second		First	Second	Third	
SANS	7.8 \pm 5.3	9.2 \pm 6.4	p =0.205	9.2 \pm 5.6	9.5 \pm 6.7	8.9 \pm 6.8	p =0.880
SAPS	6.9 \pm 2.9	6.2 \pm 3.3	p =0.462	7.4 \pm 3.4	6.2 \pm 3.2	6.7 \pm 2.5	p =0.592
CPZ ²	452.5 \pm 338.5	257.8 \pm 358.6	p =0.106	452.0 \pm 376.4	220.5 \pm 262.2	130.8 \pm 184.6	p =0.036

Abbreviations: SANS, Scale for the Assessment of Negative Symptoms; SAPS, Scale for the Assessment of Positive Symptoms; CPZ, Chlorpromazine equivalent

¹Data collected during the first session

²Average CPZ equivalents calculated over 16 and 11 patients in FEP_2 and FEP_3, respectively, by assigning a value of 0 to participants off neuroleptic medication

Table S11 - Subtypes of FEP diagnosis of FEP according to the DSM-IV for all three testing sessions

Testing Session	Number of patients	Diagnosis (DSM-IV)
First (total n = 22)		
	2	Schizophrenia, Disorganized Type (295.1)
	16	Schizophrenia, Paranoid Type (295.3)
	3	Schizophrenia, Undifferentiated Type (295.9)
	1	Bipolar I Disorder, Most Recent Episode Depressed, In Partial Remission (296.55)
Second (total n = 16)		
	1	Schizophrenia, Disorganized Type (295.1)
	8	Schizophrenia, Paranoid Type (295.3)
	2	Schizoaffective Disorder (295.7)
	3	Schizophrenia, Undifferentiated Type (295.9)
	1	Bipolar I Disorder, Most Recent Episode Depressed, Mild (296.51)
	1	Bipolar I Disorder, Most Recent Episode Depressed, In Partial Remission (296.55)
Third (total n = 11)		
	8	Schizophrenia, Paranoid Type (295.3)
	2	Schizophrenia, Undifferentiated Type (295.9)
	1	Bipolar I Disorder, Most Recent Episode Depressed, In Partial Remission (296.55)

References:

6. Tomescu MI, Rihs TA, Roinishvili M, Karahanoglu FI, Schneider M, Menghetti S, *et al.* (2015): Schizophrenia patients and 22q11.2 deletion syndrome adolescents at risk express the same deviant patterns of resting state EEG microstates: A candidate endophenotype of schizophrenia. *Schizophrenia Research: Cognition* 2: 159–165.
47. da Cruz JR, Shaqiri A, Roinishvili M, Favrod O, Chkonia E, Brand A, *et al.* (2020): Neural Compensation Mechanisms of Siblings of Schizophrenia Patients as Revealed by High-Density EEG. *Schizophrenia Bulletin*. <https://doi.org/10.1093/schbul/sbz133>
48. Favrod O, Roinishvili M, da Cruz JR, Brand A, Okruashvili M, Gamkrelidze T, *et al.* (2018): Electrophysiological correlates of visual backward masking in patients with first episode

psychosis. *Psychiatry Research: Neuroimaging*.
<https://doi.org/10.1016/j.psychresns.2018.10.008>

Thank you for your question regarding potentially truncated microstates. Potentially truncated microstates were not removed before evaluating the microstate parameters. We believe that this does not pose a problem in our group comparisons because the amount of removed periods was similar across groups. We have now added the information about the amount of removed epochs to the manuscript, as well as the information that potentially truncated microstates were not removed before evaluating the microstate parameters.

4. Methods and Materials

4.2. EEG Recording and Data Processing:

The amount of removed EEG periods was $5.56\% \pm 3.78$ for patients, $4.65\% \pm 3.42$ for siblings and $4.96\% \pm 3.76$ for controls. A one-way ANOVA revealed non-significant effect of group on the amount of removed EEG periods ($F(2,216)=0.810, P=0.446$). For Patients_22 and FEP, the amount of removed EEG periods was $5.03\% \pm 4.53$ and $3.46\% \pm 2.08$, for each group respectively. An independent samples *t*-test showed that the amount of removed EEG periods was not significantly different between Patients_22 and FEP ($t(42)=0.145, p=0.145$).

...

We did not remove potentially truncated microstates before evaluating the microstate parameters. Since the groups did not significantly differ in the amount of EEG periods removed, this does not pose a problem in the overall group comparisons.

- There is a lot of repetition throughout the manuscript, in particular a lot of overlap between methods and results sections.

Response:

Thank you for the comment. Since the journal manuscripts follow a Introduction - Results - Discussion - Methods structure, we opted to briefly introduce the methods before reporting the results to make it easier for the reader to understand the results.

Study 1:

- The sibling sample size is much smaller than patient sample size. Apparently, some patients were tested along with (at least) one sibling, while others were not. That makes statistical comparisons challenging.

Response:

Thank you for your comment. 32 patients were tested along with one sibling only. We have now computed the comparison between siblings and patients in three different ways and the results were similar:

- 1- We paired each of the 32 patients with their sibling and calculated a difference score for each microstate parameter and class for each patient-sibling pair.
- 2- We compared all the patients against all the siblings.
- 3- We compared all the patients without siblings (n=69) against all siblings.

4. Methods and Materials

4.1.1. Study 1: Patients, Siblings, and Controls:

32 out of the 43 siblings were each a sibling of a single patient in the current study (hereinafter referred to as siblings_32 and patients_32). The remaining 11 siblings were siblings of patients that performed a battery of tests but did not participate in the current EEG experiment. Group characteristics of patients_32 and siblings_32 are presented in **Table 2**. In subsequent analyses, for each of the computed microstate parameters, the score of siblings_32 was subtracted from their patients_32 pair, resulting in a difference score (Δ), which was submitted for statistical analysis.

Table 2 - Group average statistics (\pm SD) of Schizophrenia Patients, their Unaffected Siblings, Healthy Controls, Patients_32, and Siblings_32

	Patients	Siblings	Controls	Patients_32	Siblings_32	Patients vs. Controls	Siblings vs. Controls	Patients_32 vs. Siblings_32
Gender (F/M)	11 / 90	21 / 22	39 / 36	4 / 28	14 / 18	$p=2.229e-9$	$p=0.741$	$p=0.005$
Age (years)	36.9 ± 8.8	31.8 ± 10.4	35.1 ± 7.7	33.6 ± 9.1	31.9 ± 9.6	$p=0.164$	$p=0.101$	$p=0.471$
Education (years)	13.4 ± 2.7	14.1 ± 3.0	15.1 ± 2.9	13.6 ± 2.7	14.4 ± 3.0	$p=1.297e-4$	$p=0.051$	$p=0.259$
Handedness (L/R)	6 / 95	2 / 41	4 / 71	3 / 29	2 / 30	$p=0.863$	$p=0.871$	$p=0.641$
Illness duration (years)	12.8 ± 8.2			9.4 ± 7.4				
SANS	10.4 ± 5.2			10.2 ± 5.3				
SAPS	9.7 ± 7.7			8.9 ± 3.5				
CPZ ¹	577.2 ± 400.9			541.4 ± 375.9				

Abbreviations: SANS, Scale for the Assessment of Negative Symptoms; SAPS, Scale for the Assessment of Positive Symptoms; CPZ, Chlorpromazine equivalent

¹Average CPZ equivalents calculated over the 88 Patients and 27 Patients_32 receiving neuroleptic medication, respectively.

4.2. EEG Recording and Data Processing:

For patients_32 vs. siblings_32 pairs, their difference scores (Δ) for each microstate parameter and microstate class were submitted to a one-sample *t*-test against 0.

2.1. Study 1: Patients, siblings, and controls

Regarding patients vs. siblings comparisons, since 32 out of the 43 siblings were each a sibling of a patient in the current study (referred to as siblings_32 and patients_32, respectively), we paired these 32 patients to their siblings and compared their difference score (Δ) for each microstate parameter and class against 0 with one sample *t*-tests. Results revealed that siblings_32 had a longer mean duration of microstate class B than their paired patients_32 ($\Delta = -7.21 \pm 12.50$ ms; **Table 1** and **Supplementary Table S3**). The mean duration of microstate class B and the occurrence of microstate class C of patients_32 correlated with the mean duration of microstate class B and the occurrence of microstate class C in their paired siblings_32 (**Supplementary Table S4**). However, the correlations were not significant after correction for multiple comparisons (mean duration of microstate class B: $r(30)=0.360$, $p=0.043$, $p_{holm}=0.473$; occurrence of microstate class C: $r(30)=0.430$, $p=0.014$, $p_{holm}=0.168$). We also compared the microstates dynamics of all patients compared to all siblings (**Supplementary Information 1.1., Table S5**). Results indicated increased mean duration, time coverage, and occurrence of microstate class B in siblings compared to patients. Similar results were found comparing the microstate dynamics of patients without siblings in the current study ($n=69$) against all siblings (**Supplementary Information 1.1, Table S6**).

Table 1 - Pairwise group comparisons for all microstate parameters (mean duration, time coverage, and occurrence) and for each microstate class (A, B, C, and D). Statistically significant differences, after Bonferroni-Holm correction, are indicated in bold.

Comparison	Microstate Class	Mean Duration	Time Coverage	Occurrence
Patients vs. Controls	A	$p=0.054, p_{holm}=0.270, d=-0.293$	$p=0.449, p_{holm}=0.898, d=-0.110$	$p=0.882, p_{holm}=0.898, d=0.023$
	B	$p=0.003, p_{holm}=0.018, d=-0.454$	$p=0.074, p_{holm}=0.296, d=-0.271$	$p=0.112, p_{holm}=0.336, d=-0.247$
	C	$p=1.315e-4, p_{holm}=0.001, d=0.590$	$p=1.452e-7, p_{holm}=1.742e-7, d=0.827$	$p=1.170e-4, p_{holm}=0.001, d=0.602$
	D	$p=3.010e-6, p_{holm}=3.311e-5, d=-0.732$	$p=3.445e-6, p_{holm}=3.445e-5, d=-0.725$	$p=1.620e-4, p_{holm}=0.001, d=-0.578$
Siblings vs. Controls	A	$p=0.055, p_{holm}=0.288, d=-0.371$	$p=0.097, p_{holm}=0.288, d=-0.320$	$p=0.250, p_{holm}=0.288, d=0.221$
	B	$p=0.049, p_{holm}=0.288, d=0.381$	$p=0.048, p_{holm}=0.288, d=0.382$	$p=0.069, p_{holm}=0.288, d=0.351$
	C	$p=0.022, p_{holm}=0.154, d=0.445$	$p=0.001, p_{holm}=0.010, d=0.631$	$p=0.006, p_{holm}=0.048, d=0.533$
	D	$p=3.380e-4, p_{holm}=0.004, d=-0.707$	$p=1.465e-4, p_{holm}=0.002, d=-0.751$	$p=0.004, p_{holm}=0.036, d=-0.566$
Patients_32 vs. Siblings_32	A	$p=0.235, p_{holm}=1.000, d=0.214$	$p=0.020, p_{holm}=0.200, d=0.434$	$p=0.014, p_{holm}=0.154, d=0.462$
	B	$p=0.003, p_{holm}=0.036, d=-0.576$	$p=0.069, p_{holm}=0.621, d=-0.333$	$p=0.200, p_{holm}=1.000, d=-0.231$
	C	$p=0.838, p_{holm}=1.000, d=-0.036$	$p=0.783, p_{holm}=1.000, d=0.049$	$p=0.588, p_{holm}=1.000, d=0.097$
	D	$p=0.270, p_{holm}=1.000, d=-0.199$	$p=0.491, p_{holm}=1.000, d=-0.123$	$p=0.772, p_{holm}=1.000, d=-0.052$

Supplementary Information

1. Supplementary Results

1.1. Study 1: Patients, siblings, and controls:

Table S3 – Group average statistics (\pm SD) of the Schizophrenia Patients, their Unaffected Siblings and Healthy Controls as well as the average difference score of Patients_32 and their paired Siblings_32 for all the computed microstates parameters and classes.

Microstate Parameter	Microstate Class	Group			Patients_32 - Siblings_32 (Δ)
		Patients	Siblings	Controls	
Mean Duration (ms)	A	69.05 \pm 10.06	66.93 \pm 8.61	71.83 \pm 15.20	2.45 \pm 11.45
	B	66.34 \pm 8.97	74.81 \pm 11.87	71.05 \pm 8.52	-7.21 \pm 12.50
	C	106.99 \pm 27.75	99.05 \pm 22.16	90.31 \pm 18.05	-0.93 \pm 25.40
	D	70.42 \pm 11.16	71.41 \pm 11.02	82.01 \pm 16.84	-3.17 \pm 15.96
Time Coverage (%)	A	18.21 \pm 8.98	15.75 \pm 6.79	18.85 \pm 11.03	4.32 \pm 9.94
	B	16.15 \pm 7.71	22.48 \pm 9.99	19.27 \pm 7.35	-3.69 \pm 11.06
	C	46.38 \pm 12.98	41.68 \pm 12.70	34.00 \pm 11.87	0.82 \pm 16.71
	D	19.27 \pm 8.38	20.10 \pm 7.73	27.89 \pm 11.61	-1.45 \pm 11.79
Occurrence	A	1.87 \pm 0.66	1.69 \pm 0.62	1.82 \pm 0.61	0.38 \pm 0.81
	B	1.74 \pm 0.58	2.12 \pm 0.59	1.93 \pm 0.53	-0.18 \pm 0.79
	C	2.83 \pm 0.38	2.74 \pm 0.36	2.52 \pm 0.45	0.04 \pm 0.43
	D	1.95 \pm 0.59	2.00 \pm 0.58	2.34 \pm 0.59	-0.04 \pm 0.80

Table S4 - Pearson correlation results of the Patients_32 - Siblings_32 pairs for all the computer microstates parameters and classes. Statistically significant results (without Bonferroni-Holm correction) are indicated in bold. No statistically significant results were found after correction for multiple comparisons.

Microstate Parameter	Microstate Class	$r(30)$ - value	p - value	p_{holm} - value
Mean Duration	A	0.016	0.930	1.000
	B	0.360	0.043	0.473
	C	0.043	0.817	1.000
	D	0.090	0.623	1.000
Time Coverage	A	0.032	0.861	1.000
	B	0.314	0.080	0.800
	C	0.109	0.552	1.000
	D	-0.023	0.898	1.000
Occurrence	A	0.056	0.759	1.000
	B	0.120	0.514	1.000
	C	0.430	0.014	0.168
	D	0.044	0.811	1.000

We also compared the microstates dynamics of all patients against all siblings. Since the groups differed in terms of age ($t(142)=3.007, p=0.003$) and gender ($\chi^2(1)=25.126, p=5.371e-7$) but not in terms of education ($t(142)=1.430, p=0.155$) or handedness ($\chi^2(1)=0.096, p=0.757$), gender was used as a factor and age as a covariate in subsequent analyses. For each of the computed microstates parameters (mean duration, time coverage, and occurrence), we performed a three-way repeated measures (rm)-ANOVA, with three factors: gender (male and female), group (patients and siblings), and microstate class (A, B, C, and D). The analyses showed non-significant Gender \times Group \times Microstate Class interaction for mean duration ($F(3,417)=1.066, P=0.363, \eta^2=0.006$), time of coverage ($F(3,417)=1.077, P=0.358, \eta^2=0.007$), and occurrence ($F(3,417)=0.770, P=0.511, \eta^2=0.004$). The analyses also yielded significant Microstate Class \times Group interaction for mean duration ($F(3,417)=4.845, P=0.003, \eta^2=0.027$), time of coverage ($F(3,417)=4.883, P=0.002, \eta^2=0.030$), and occurrence ($F(3,417)=3.449, P=0.017, \eta^2=0.016$). The interactions indicate that group differences depend on the microstate class. Post-hoc pairwise group comparisons (**Table S5**), using Bonferroni-Holm correction for the 12 comparisons, showed that siblings had increased mean duration and time coverage of microstate class B compared to patients. Since, after Bonferroni-Holm correction for multiple comparisons, occurrence of microstate class B in siblings was marginally significantly higher than in patients, we conducted a JZS Bayes ANCOVA with default priors (3-5) to evaluate if there was more

evidence in favor of the null hypothesis (no difference between groups) as compared to the alternative (i.e., that there was a difference between groups). The results indicated that the model with the Group effect was preferred to the models with Group + Gender, Group + Gender + Group × Gender, Group + Gender + Age, Group + Gender + Age + Group × Gender, and the Null model by Bayes factors of 4.255, 12.821, 20.408, 35.714, and 50.000, respectively. No statistically significant group differences were found for microstate classes A, C, and D.

Table S5 - Post-hoc group comparisons (all patients vs. all siblings) of all microstate parameters (mean duration, time coverage, and occurrence), using Bonferroni-Holm correction, for each microstate class (A, B, C, and D). Statistically significant differences are indicated in bold.

Microstate Parameter	Microstate Class			
	A	B	C	D
Mean Duration	$p=0.863$, $p_{holm}=1.000$, $d=0.029$	$p=2.102e-4$, $p_{holm}=0.003$, $d=-$ 0.637	$p=0.058$, $p_{holm}=0.464$, $d=0.320$	$p=0.147$, $p_{holm}=0.959$, $d=-$ 0.247
Time Coverage	$p=0.435$, $p_{holm}=1.000$, $d=0.127$	$p=0.002$, $p_{holm}=0.022$, $d=-$ 0.523	$p=0.017$, $p_{holm}=0.153$, $d=0.408$	$p=0.154$, $p_{holm}=0.959$, $d=-$ 0.238
Occurrence	$p=0.587$, $p_{holm}=1.000$, $d=0.090$	$p=0.006$, $p_{holm}=0.060$, $d=-$ 0.473	$p=0.240$, $p_{holm}=0.960$, $d=0.201$	$p=0.137$, $p_{holm}=0.959$, $d=-$ 0.247

We also compared the microstates dynamics of all patients without siblings in the current study (Patients_no_Sibs; $n = 69$) against all siblings. Since the groups differed in terms of age ($t(110)=3.724$, $p=3.114e-4$) and gender ($\chi^2(1)=21.152$, $p=4.243e-6$) but not in terms of education ($t(110)=1.451$, $p=0.150$) or handedness ($\chi^2(1)=0.006$, $p=0.940$), gender was used as a factor and age as a covariate in subsequent analyses. For each of the computed microstates parameters (mean duration, time coverage, and occurrence), we performed a three-way repeated measures (rm)-ANOVA, with three factors: gender (male and female), group (Patients_no_Sibs and siblings), and microstate class (A, B, C, and D). The analyses showed non-significant Gender × Group × Microstate Class interaction for mean duration ($F(3,321)=0.357$, $P=0.784$, $\eta^2=0.003$), time of coverage ($F(3,321)=0.320$, $P=0.811$, $\eta^2=0.002$), and occurrence ($F(3,321)=0.172$, $P=0.915$, $\eta^2=9.971e-4$). The analyses also yielded significant Microstate Class × Group interaction for mean duration ($F(3,321)=5.522$, $P=0.001$, $\eta^2=0.039$), time of coverage ($F(3,321)=5.702$, $P=8.174e-4$, $\eta^2=0.044$), and occurrence ($F(3,321)=4.303$, $P=0.005$, $\eta^2=0.025$). These interactions indicate that group differences depend on the microstate class. Summary statistics for Patients_no_Sibs and post-hoc pairwise group comparisons are shown in **Table S6**. Results, using Bonferroni-Holm correction for the 12 comparisons, showed that siblings had increased mean duration, time coverage, and occurrence of microstate class B compared to Patients_no_Sibs. No statistically significant group differences were found for microstate classes A, C, and D.

Table S1 - Group average statistics (\pm SD) of the schizophrenia patients without siblings in the current study (Patients_no_Sibs) for all the computed microstates parameters (mean duration, time coverage, and occurrence) and classes (A, B, C, and D). As well as post-hoc group comparisons (Patients_no_Sibs vs. Siblings) for all microstate parameters, using Bonferroni-Holm correction, for each microstate class. Statistically significant differences are indicated in bold.

Microstate Parameter	Microstate Class	Patients_no_Sibs	Patients_no_Sibs vs. Siblings
	A	69.01 \pm 10.28	$p=0.626, p_{holm}=1.000, d=0.090$
Mean Duration	B	65.89 \pm 8.32	$p=2.331e-4, p_{holm}=0.003, d=0.725$
(ms)	C	111.09 \pm 26.43	$p=0.029, p_{holm}=0.232, d=0.424$
	D	70.78 \pm 11.09	$p=0.346, p_{holm}=1.000, d=0.180$
	A	17.61 \pm 9.20	$p=0.501, p_{holm}=1.000, d=0.127$
Time Coverage	B	15.11 \pm 6.51	$p=3.014e-4, p_{holm}=0.003, d=0.717$
(%)	C	48.07 \pm 12.46	$p=0.011, p_{holm}=0.099, d=0.496$
	D	19.21 \pm 8.48	$p=0.322, p_{holm}=1.000, d=0.191$
	A	1.79 \pm 0.66	$p=0.772, p_{holm}=1.000, d=0.056$
Occurrence	B	1.66 \pm 0.52	$p=7.857e-4, p_{holm}=0.008, d=0.667$
	C	2.84 \pm 0.34	$p=0.142, p_{holm}=0.994, d=0.286$
	D	1.94 \pm 0.60	$p=0.322, p_{holm}=1.000, d=0.191$

References:

3. Wagenmakers E-J (2007): A practical solution to the pervasive problems of p values. *Psychonomic Bulletin & Review* 14: 779–804.
4. Rouder JN, Morey RD, Speckman PL, Province JM (2012): Default Bayes factors for ANOVA designs. *Journal of Mathematical Psychology* 56: 356–374.
5. Ly A, Verhagen J, Wagenmakers E-J (2016): Harold Jeffreys’s default Bayes factor hypothesis tests: Explanation, extension, and application in psychology. *Journal of Mathematical Psychology* 72: 19–32.

- On a related note: observations in patient-sibling pairs are not completely independent from one another, which does not appear to have been taken into account in analyses.

Response:

Thank you for your comment. As mentioned above, we have now computed the comparison between siblings and patients in three different ways:

- 1- We paired each of the 32 patients with their sibling and calculated a difference score for each microstate parameter and class for each patient-sibling pair.
- 2- We compared all the patients against all the siblings.
- 3- We compared all the patients without siblings (n=69) against all siblings.

- The majority of patients were medicated. This makes interpretation of results challenging, especially regarding microstate B (the authors interpret microstate B increase in siblings as indicating some compensation mechanism, but it could theoretically just as well be the case that 'normal' microstate B in patients represents a medication effect). Moreover, the authors do not discuss their results compared to previous studies, most of which have investigated unmedicated, acutely ill patients. Only two studies assessed medicated patients. Of those, a study by Tomescu et al. in a (probably) overlapping chronic patient sample with the current study found similar microstate profiles in patients, but another by Andreou et al. in a FEP sample reported partially different results.

Response:

Thank you for your comment. Regarding the microstate class B, that is our interpretation. We did not find any correlation between medication and microstate class B. However, since absence of proof is not proof of absence, in the discussion we mention that "Third, we cannot exclude the potential effects of treatment in the microstate class B differences between siblings and patients."

Regarding comparison with previous studies, we have now discussed our results in the light of previous findings as follows:

Discussion:

Regarding associations between psychopathological symptoms and microstate dynamics, in FEP, we observed negative correlations between the SANS scores and the time coverage of microstate class D as well as between the SANS scores and the occurrence of microstate class D; however, the correlations were not significant after correcting for multiple comparisons. For chronic schizophrenia patients, we found no significant correlations between psychopathological symptoms and the microstate parameters for any of the microstate classes. Nonetheless, the coefficients of the Pearson correlation between the SANS scores of schizophrenia patients and their microstate parameters for microstate class D were negative as in FEP (mean duration: $r(99)=-0.144, p=0.150$; time coverage: $r(99)=-0.190, p=0.057$; occurrence: $r(99)=-0.183, p=0.067$; **Supplementary Table S7**). In the literature, the duration of microstate class D has

been found to correlate negatively with scores of paranoid-hallucinatory symptomatology (18) and with acute hallucination experiences (43) in patients with schizophrenia. More recently, it has been reported that the time coverage of microstate class A correlated positively with avolition-apathy scores, even though there were no group differences between patients and controls (30). Finally, in a sample of adolescents with 22q11.2 deletion syndrome, the mean durations of microstate class C were associated with increased hallucination subscores of the Structured Interview for Prodromal Syndromes (23). These results suggest that there might be an association between the microstate dynamics and psychopathological symptoms, however, the results in the literature are too heterogeneous to make firm conclusions at this point.

There are several considerations that should be taken into account. First, there are demographics differences between schizophrenia patients, their siblings, and controls. Tomescu and colleagues (41) showed evidence for age and gender-specific effects on the microstates dynamics. Here, we tried to minimize these effects by using age as a co-variate and gender as a factor in the analyses. Second, schizophrenia is a heterogeneous disease and our samples may be too small to cover the full schizophrenia spectrum. Third, we cannot exclude the potential effects of treatment in the microstate class B differences between siblings and patients.

Most of our patients were medicated and, in chronic schizophrenia patients, we found a positive association with medication intake (CPZ equivalents) and the occurrence of microstate class C (although not significant after correction for multiple comparisons), providing evidence that medication interact with microstates dynamics. This potential interaction is also supported by previous work that has shown that perospirone (an antipsychotic drug) can increase the duration of microstate class D in healthy controls (44). In addition, antipsychotic medication has been shown to normalize microstate dynamics (decrease presence of microstate class C and increase presence of microstate class D), in patients that respond well to antipsychotic treatment (33). While most of the studies included in our meta-analysis only investigated the microstates dynamics in medication naïve patients, few studies investigated patients taking anti-psychotic medication. One of these studies found that microstate class D was decreased in FEP compared to controls (31), while another found increased duration and time coverage of microstate class C in schizophrenia patients compared to controls (30). Additionally, a recent study with FEP also identified decreased mean durations of microstate class A in FEP compared to controls, a result that does not align with the literature (29). However, since most of the studies of EEG microstates in schizophrenia have small samples ($n < 30$), it is expected that, due to sampling error and the heterogeneity of the disorder, some effects might not be significant in some studies and even reversed in a few studies if the effect sizes are small.

Supplementary Information

1. Supplementary Results

1.1. Study 1: Patients, siblings, and controls:

In chronic schizophrenia patients, the CPZ equivalents were found to correlate with the occurrence of microstate class C (**Table S7**). However, the correlation was not significant after correction for 12 comparisons (4 microstate classes \times 3 microstate parameters) with Bonferroni-

Holm ($r(86)=0.236$, $p=0.027$, $p_{holm}=0.324$). No other significant associations were found between the computed microstate parameters and either CPZ equivalent, SANS, SAPS, or illness duration.

Table S7 - Pearson correlation between all computed microstate parameters and Chlorpromazine equivalent (CPZ), Scale for Assessment of Negative (SANS) and Positive (SAPS) Symptoms, and Illness duration, for chronic schizophrenia patients. (df - degrees of freedom). Statistically significant correlations (without correction for multiple comparisons) are indicated in bold.

Microstate Parameter	Microstate Class	CPZ (df = 86)	SANS (df = 99)	SAPS (df = 99)	Illness Duration (df = 99)
Mean	A	$r=-0.015$, $p=0.890$	$r=0.018$, $p=0.862$	$r=0.007$, $p=0.948$	$r=0.069$, $p=0.490$
	B	$r=-0.057$, $p=0.596$	$r=0.056$, $p=0.576$	$r=0.041$, $p=0.685$	$r=-0.062$, $p=0.539$
Duration	C	$r=0.012$, $p=0.910$	$r=-0.017$, $p=0.866$	$r=-0.066$, $p=0.514$	$r=-0.053$, $p=0.598$
	D	$r=0.092$, $p=0.391$	$r=-0.144$, $p=0.150$	$r=0.004$, $p=0.972$	$r=0.042$, $p=0.674$
Time	A	$r=-0.041$, $p=0.702$	$r=0.028$, $p=0.778$	$r=0.026$, $p=0.794$	$r=0.089$, $p=0.374$
	B	$r=-0.098$, $p=0.366$	$r=0.119$, $p=0.237$	$r=0.039$, $p=0.702$	$r=-0.065$, $p=0.516$
Coverage	C	$r=0.038$, $p=0.724$	$r=0.032$, $p=0.749$	$r=-0.048$, $p=0.635$	$r=-0.060$, $p=0.552$
	D	$r=0.079$, $p=0.465$	$r=-0.190$, $p=0.057$	$r=0.010$, $p=0.918$	$r=0.057$, $p=0.571$
Occurrence	A	$r=-0.011$, $p=0.9211$	$r=0.034$, $p=0.734$	$r=0.045$, $p=0.653$	$r=0.080$, $p=0.426$
	B	$r=-0.062$, $p=0.567$	$r=0.121$, $p=0.228$	$r=0.037$, $p=0.717$	$r=-0.073$, $p=0.469$
	C	$r=0.236$, $p=0.027$	$r=0.010$, $p=0.919$	$r=0.033$, $p=0.743$	$r=-0.085$, $p=0.397$
	D	$r=0.081$, $p=0.453$	$r=-0.183$, $p=0.067$	$r=0.017$, $p=0.863$	$r=0.036$, $p=0.720$

Supplementary Information

1. Supplementary Results

1.2. Study 2: FEP:

In FEP, the SANS scores correlated negatively with the time coverage and occurrence of microstate class D (**Table S9**). However, the correlations were not significant after correction for 12 comparisons (4 microstate classes \times 3 microstate parameters) with Bonferroni-Holm (time coverage of microstate class D: $r(20)=-0.516$, $p=0.014$, $p_{holm}=0.168$; occurrence of microstate class C: $r(20)=-0.429$, $p=0.046$, $p_{holm}=0.506$). No other significant associations were found between the computed microstate parameters and either CPZ equivalent, SANS, SAPS, or illness duration.

Table S9 - Pearson correlation between all computed microstate parameters and Chlorpromazine equivalent (CPZ), Scale for Assessment of Negative (SANS) and Positive (SAPS) Symptoms, and Illness duration, for patients with a first episode of psychosis (FEP). (df - degrees of freedom). Statistically significant correlations (without correction for multiple comparisons) are indicated in bold.

Microstate Parameter	Microstate Class	CPZ (df = 18)	SANS (df = 20)	SAPS (df = 20)	Illness Duration (df = 20)
Mean Duration	A	$r=-0.154$, $p=0.517$	$r=0.197$, $p=0.379$	$r=0.179$, $p=0.426$	$r=0.117$, $p=0.605$
	B	$r=-0.125$, $p=0.600$	$r=-0.054$, $p=0.812$	$r=-0.126$, $p=0.576$	$r=-0.110$, $p=0.627$
	C	$r=0.242$, $p=0.304$	$r=0.106$, $p=0.638$	$r=0.170$, $p=0.449$	$r=-0.193$, $p=0.390$
	D	$r=-0.178$, $p=0.452$	$r=-0.352$, $p=0.109$	$r=0.142$, $p=0.529$	$r=-0.230$, $p=0.303$
Time Coverage	A	$r=-0.156$, $p=0.511$	$r=0.244$, $p=0.273$	$r=0.150$, $p=0.504$	$r=0.205$, $p=0.359$
	B	$r=-0.127$, $p=0.595$	$r=0.055$, $p=0.808$	$r=-0.150$, $p=0.506$	$r=0.033$, $p=0.884$
	C	$r=0.240$, $p=0.309$	$r=0.049$, $p=0.828$	$r=0.069$, $p=0.759$	$r=-0.121$, $p=0.590$
	D	$r=-0.030$, $p=0.901$	$r=-0.516$, $p=0.014$	$r=-0.091$, $p=0.688$	$r=-0.118$, $p=0.600$
Occurrence	A	$r=-0.103$, $p=0.666$	$r=0.281$, $p=0.205$	$r=0.048$, $p=0.832$	$r=0.212$, $p=0.344$
	B	$r=-0.061$, $p=0.799$	$r=0.390$, $p=0.073$	$r=0.047$, $p=0.834$	$r=0.410$, $p=0.058$
	C	$r=0.134$, $p=0.573$	$r=0.075$, $p=0.741$	$r=0.079$, $p=0.728$	$r=0.084$, $p=0.709$
	D	$r=0.033$, $p=0.890$	$r=-0.429$, $p=0.046$	$r=-0.149$, $p=0.507$	$r=0.045$, $p=0.844$

References:

18. Koenig T, Lehmann D, Merlo MCG, Kochi K, Hell D, Koukkou M (1999): A deviant EEG brain microstate in acute, neuroleptic-naïve schizophrenics at rest. *European Archives of Psychiatry and Clinical Neuroscience* 249: 205–211.
23. Tomescu MI, Rihs TA, Becker R, Britz J, Custo A, Grouiller F, *et al.* (2014): Deviant dynamics of EEG resting state pattern in 22q11.2 deletion syndrome adolescents: A vulnerability marker of schizophrenia? *Schizophrenia Research* 157: 175–181.
29. Murphy M, Stickgold R, Öngür D (2019): Electroencephalogram Microstate Abnormalities in Early-Course Psychosis. *Biological Psychiatry: Cognitive Neuroscience and Neuroimaging*. <https://doi.org/10.1016/j.bpsc.2019.07.006>

30. Giordano GM, Koenig T, Mucci A, Vignapiano A, Amodio A, Di Lorenzo G, *et al.* (2018): Neurophysiological correlates of Avolition-apathy in schizophrenia: A resting-EEG microstates study. *NeuroImage: Clinical* 20: 627–636.
31. Andreou C, Faber PL, Leicht G, Schoettle D, Polomac N, Hanganu-Opatz IL, *et al.* (2014): Resting-state connectivity in the prodromal phase of schizophrenia: Insights from EEG microstates. *Schizophrenia Research* 152: 513–520.
33. Kikuchi M, Koenig T, Wada Y, Higashima M, Koshino Y, Strik W, Dierks T (2007): Native EEG and treatment effects in neuroleptic-naïve schizophrenic patients: Time and frequency domain approaches. *Schizophrenia Research* 97: 163–172.
41. Tomescu MI, Rihs TA, Rochas V, Hardmeier M, Britz J, Allali G, *et al.* (2018): From swing to cane: Sex differences of EEG resting-state temporal patterns during maturation and aging. *Developmental Cognitive Neuroscience* 31: 58–66.
43. Kindler J, Hubl D, Strik WK, Dierks T, Koenig T (2011): Resting-state EEG in schizophrenia: Auditory verbal hallucinations are related to shortening of specific microstates. *Clinical Neurophysiology* 122: 1179–1182.
44. Yoshimura M, Koenig T, Irisawa S, Isotani T, Yamada K, Kikuchi M, *et al.* (2007): A pharmaco-EEG study on antipsychotic drugs in healthy volunteers. *Psychopharmacology* 191: 995–1004.

- Compared to the study by Tomescu et al., average microstate duration is longer in the present study. This is in itself not problematic, unless there is indeed overlap with the current patient sample, in which case the difference is difficult to explain without assuming that the authors made changes in their pre- or postprocessing analysis pipeline. Please clarify.

Response:

We thank the reviewer for the comment. There is indeed an overlap of 27 patients and 27 controls between the current study and the study by Tomescu et al., 2015. We have now mentioned this in the manuscript and explained that the differences are due to the temporal smoothing; in fact, in the current study we rejected microstates with durations of 1 time frame, which was not the case in Tomescu et al., 2015.

4. Methods and Materials

4.2. EEG Recording and Data Processing:

The temporal smoothing in the current study was different from the one performed by Tomescu and colleagues (6), a study with data from 27 patients and 27 controls included in our sample. As mentioned above, in the current study, we rejected microstates with durations of 1 time frame, while Tomescu and colleagues did not. This resulted in the microstate mean durations in the current study to be longer than the ones reported by Tomescu et al. However, in a subsequent

work from the same group of researchers (41), rejection of microstates with durations of 1 time frame was applied, which led to microstate mean durations similar to the ones in current study.

References:

6. Tomescu MI, Rihs TA, Roinishvili M, Karahanoglu FI, Schneider M, Menghetti S, *et al.* (2015): Schizophrenia patients and 22q11.2 deletion syndrome adolescents at risk express the same deviant patterns of resting state EEG microstates: A candidate endophenotype of schizophrenia. *Schizophrenia Research: Cognition* 2: 159–165.

41. Tomescu MI, Rihs TA, Rochas V, Hardmeier M, Britz J, Allali G, *et al.* (2018): From swing to cane: Sex differences of EEG resting-state temporal patterns during maturation and aging. *Developmental Cognitive Neuroscience* 31: 58–66.

- The authors should discuss symptom severity in their sample compared to previous studies – most previous studies investigated acutely ill patients.

Response:

Thank you for the suggestion. We have now discussed the symptom severity of our sample and previous studies in the discussion as follows:

Discussion:

Regarding associations between psychopathological symptoms and microstate dynamics, in FEP, we observed negative correlations between the SANS scores and the time coverage of microstate class D as well as between the SANS scores and the occurrence of microstate class D; however, the correlations were not significant after correcting for multiple comparisons. For chronic schizophrenia patients, we found no significant correlations between psychopathological symptoms and the microstate parameters for any of the microstate classes. Nonetheless, the coefficients of the Pearson correlation between the SANS scores of schizophrenia patients and their microstate parameters for microstate class D were negative as in FEP (mean duration: $r(99)=-0.144$, $p=0.150$; time coverage: $r(99)=-0.190$, $p=0.057$; occurrence: $r(99)=-0.183$, $p=0.067$; **Supplementary Table S7**). In the literature, the duration of microstate class D has been found to correlate negatively with scores of paranoid-hallucinatory symptomatology (18) and with acute hallucination experiences (43) in patients with schizophrenia. More recently, it has been reported that the time coverage of microstate class A correlated positively with avolition-apathy scores, even though there were no group differences between patients and controls (30). Finally, in a sample of adolescents with 22q11.2 deletion syndrome, the mean durations of microstate class C were associated with increased hallucination subscores of the Structured Interview for Prodromal Syndromes (23). These results suggest that there might be an association between the microstate dynamics and psychopathological symptoms, however, the results in the literature are too heterogeneous to make firm conclusions at this point.

Supplementary Information

1. Supplementary Results

1.1. Study 1: Patients, siblings, and controls:

In chronic schizophrenia patients, the CPZ equivalents were found to correlate with the occurrence of microstate class C (**Table S7**). However, the correlation was not significant after correction for 12 comparisons (4 microstate classes × 3 microstate parameters) with Bonferroni-Holm ($r(86)=0.236$, $p=0.027$, $p_{holm}=0.324$). No other significant associations were found between the computed microstate parameters and either CPZ equivalent, SANS, SAPS, or illness duration.

Table S7 - Pearson correlation between all computed microstate parameters and Chlorpromazine equivalent (CPZ), Scale for Assessment of Negative (SANS) and Positive (SAPS) Symptoms, and Illness duration, for chronic schizophrenia patients. (df - degrees of freedom). Statistically significant correlations (without correction for multiple comparisons) are indicated in bold.

Microstate Parameter	Microstate Class	CPZ (df = 86)	SANS (df = 99)	SAPS (df = 99)	Illness Duration (df = 99)
Mean	A	$r=-0.015$, $p=0.890$	$r=0.018$, $p=0.862$	$r=0.007$, $p=0.948$	$r=0.069$, $p=0.490$
	B	$r=-0.057$, $p=0.596$	$r=0.056$, $p=0.576$	$r=0.041$, $p=0.685$	$r=-0.062$, $p=0.539$
	C	$r=0.012$, $p=0.910$	$r=-0.017$, $p=0.866$	$r=-0.066$, $p=0.514$	$r=-0.053$, $p=0.598$
	D	$r=0.092$, $p=0.391$	$r=-0.144$, $p=0.150$	$r=0.004$, $p=0.972$	$r=0.042$, $p=0.674$
Time	A	$r=-0.041$, $p=0.702$	$r=0.028$, $p=0.778$	$r=0.026$, $p=0.794$	$r=0.089$, $p=0.374$
	B	$r=-0.098$, $p=0.366$	$r=0.119$, $p=0.237$	$r=0.039$, $p=0.702$	$r=-0.065$, $p=0.516$
	C	$r=0.038$, $p=0.724$	$r=0.032$, $p=0.749$	$r=-0.048$, $p=0.635$	$r=-0.060$, $p=0.552$
	D	$r=0.079$, $p=0.465$	$r=-0.190$, $p=0.057$	$r=0.010$, $p=0.918$	$r=0.057$, $p=0.571$
Coverage	A	$r=-0.011$, $p=0.9211$	$r=0.034$, $p=0.734$	$r=0.045$, $p=0.653$	$r=0.080$, $p=0.426$
	B	$r=-0.062$, $p=0.567$	$r=0.121$, $p=0.228$	$r=0.037$, $p=0.717$	$r=-0.073$, $p=0.469$
	C	$r=0.236$, $p=0.027$	$r=0.010$, $p=0.919$	$r=0.033$, $p=0.743$	$r=-0.085$, $p=0.397$
	D	$r=0.081$, $p=0.453$	$r=-0.183$, $p=0.067$	$r=0.017$, $p=0.863$	$r=0.036$, $p=0.720$

Supplementary Information

1. Supplementary Results

1.2. Study 2: FEP:

In FEP, the SANS scores correlated negatively with the time coverage and occurrence of microstate class D (**Table S9**). However, the correlations were not significant after correction for 12 comparisons (4 microstate classes \times 3 microstate parameters) with Bonferroni-Holm (time coverage of microstate class D: $r(20)=-0.516$, $p=0.014$, $p_{holm}=0.168$; occurrence of microstate class C: $r(20)=-0.429$, $p=0.046$, $p_{holm}=0.506$). No other significant associations were found between the computed microstate parameters and either CPZ equivalent, SANS, SAPS, or illness duration.

Table S9 - Pearson correlation between all computed microstate parameters and Chlorpromazine equivalent (CPZ), Scale for Assessment of Negative (SANS) and Positive (SAPS) Symptoms, and Illness duration, for patients with a first episode of psychosis (FEP). (df - degrees of freedom). Statistically significant correlations (without correction for multiple comparisons) are indicated in bold.

Microstate Parameter	Microstate Class	CPZ (df = 18)	SANS (df = 20)	SAPS (df = 20)	Illness Duration (df = 20)
Mean Duration	A	$r=-0.154$, $p=0.517$	$r=0.197$, $p=0.379$	$r=0.179$, $p=0.426$	$r=0.117$, $p=0.605$
	B	$r=-0.125$, $p=0.600$	$r=-0.054$, $p=0.812$	$r=-0.126$, $p=0.576$	$r=-0.110$, $p=0.627$
	C	$r=0.242$, $p=0.304$	$r=0.106$, $p=0.638$	$r=0.170$, $p=0.449$	$r=-0.193$, $p=0.390$
	D	$r=-0.178$, $p=0.452$	$r=-0.352$, $p=0.109$	$r=0.142$, $p=0.529$	$r=-0.230$, $p=0.303$
Time Coverage	A	$r=-0.156$, $p=0.511$	$r=0.244$, $p=0.273$	$r=0.150$, $p=0.504$	$r=0.205$, $p=0.359$
	B	$r=-0.127$, $p=0.595$	$r=0.055$, $p=0.808$	$r=-0.150$, $p=0.506$	$r=0.033$, $p=0.884$
	C	$r=0.240$, $p=0.309$	$r=0.049$, $p=0.828$	$r=0.069$, $p=0.759$	$r=-0.121$, $p=0.590$
	D	$r=-0.030$, $p=0.901$	$r=-0.516$, $p=0.014$	$r=-0.091$, $p=0.688$	$r=-0.118$, $p=0.600$
Occurrence	A	$r=-0.103$, $p=0.666$	$r=0.281$, $p=0.205$	$r=0.048$, $p=0.832$	$r=0.212$, $p=0.344$
	B	$r=-0.061$, $p=0.799$	$r=0.390$, $p=0.073$	$r=0.047$, $p=0.834$	$r=0.410$, $p=0.058$
	C	$r=0.134$, $p=0.573$	$r=0.075$, $p=0.741$	$r=0.079$, $p=0.728$	$r=0.084$, $p=0.709$
	D	$r=0.033$, $p=0.890$	$r=-0.429$, $p=0.046$	$r=-0.149$, $p=0.507$	$r=0.045$, $p=0.844$

References:

18. Koenig T, Lehmann D, Merlo MCG, Kochi K, Hell D, Koukkou M (1999): A deviant EEG brain microstate in acute, neuroleptic-naive schizophrenics at rest. *European Archives of Psychiatry and Clinical Neurosciences* 249: 205–211.
23. Tomescu MI, Rihs TA, Becker R, Britz J, Custo A, Grouiller F, *et al.* (2014): Deviant dynamics of EEG resting state pattern in 22q11.2 deletion syndrome adolescents: A vulnerability marker of schizophrenia? *Schizophrenia Research* 157: 175–181.
30. Giordano GM, Koenig T, Mucci A, Vignapiano A, Amodio A, Di Lorenzo G, *et al.* (2018): Neurophysiological correlates of Avolition-apathy in schizophrenia: A resting-EEG microstates study. *NeuroImage: Clinical* 20: 627–636.
43. Kindler J, Hubl D, Strik WK, Dierks T, Koenig T (2011): Resting-state EEG in schizophrenia: Auditory verbal hallucinations are related to shortening of specific microstates. *Clinical Neurophysiology* 122: 1179–1182.

- Minor remark, as it will not affect significance levels in most cases: It is not clear how Bonferroni-Holm was applied; were results corrected for 4 microstates or 4 ms x 3 parameters each?

Response:

Thank you for the question. We have now clarified throughout the paper how the p-values were corrected (in general 4 microstates x 3 parameters) and we provided the exact p-values and the p-values after Bonferroni-Holm correction.

Studies 2 and 3:

- Negative results are rather difficult to assess given the small sample sizes; it might be useful to present power calculations.

Thank you for the comment. We did not perform a power analysis because we cannot determine what is the minimum effect size that one is interested in for the difference between patients with a first episodes of psychosis and chronic schizophrenia patients. However, we conducted a sensitivity analysis to determine what is the minimum effect size that we can detect with our design. The results showed that we could detect medium to small effect sizes for the interaction effects and the main effect of group, respectively. We also conducted Bayesian ANOVAs to evaluate whether there was more evidence for the null (no difference between groups) or the alternative hypothesis (that there was a difference between groups). The results showed that there was positively more evidence for no group differences or group x microstate class interaction.

We have added the analyses as follows:

2. Results:

2.2. Study 2: FEP:

Since the null results are relevant to the overall interpretation of the results, we conducted two additional analyses to evaluate the sensitivity of our study and whether there were supporting evidence for the null hypotheses. First, we conducted a sensitivity analysis with the program G*Power (25) to compute the interaction and main effect of group effect sizes that we can detect with a power of 80%, given 22 participants in each of the 2 groups and 4 conditions. The analysis revealed that we could detect interaction effects and main effects of group with main effect sizes with η^2 of 0.068 and 0.026, which are medium and small effect sizes according to Cohen (26).

Second, we examined the data by estimating Bayes factors using Bayesian Information Criteria (27), comparing the fit of the data under the main effects model and the interaction model for each of the computed microstates parameters. JZS Bayes factor ANOVAs (28) with default prior scales revealed that the main effects models were preferred to the interaction model by Bayes factors of 5.545, 6.609, and 6.236, for mean duration, time of coverage, and occurrence, respectively. In other words, the data provided positive evidence against the hypotheses that group and microstate class interact in any of the computed microstates parameters. We further compared the main effects models and models without the main effect of group. Results show that models without the main effect of group were preferred to the main effects models by Bayes factors of 4.129, 5.444, and 3.841, for mean duration, time of coverage, and occurrence, respectively.

We also computed Bayesian ANOVAs for the longitudinal study of the patients with a first episode of psychosis. Results show that there was positively more evidence for no testing session differences or testing session x microstate class interaction.

2. Results:

2.2. Study 2: FEP:

For the FEP_2 comparison, two-way rm-ANOVAs yielded no non-significant testing session (First and Session) \times microstate class (A, B, C, and D) interactions for mean duration ($F(3,45)=0.433$, $P=0.730$, $\eta^2=0.004$), time of coverage ($F(3,45)=0.512$, $P=0.676$, $\eta^2=0.005$), and occurrence ($F(3,45)=1.060$, $P=0.375$, $\eta^2=0.009$), as well as non-significant testing session differences for mean duration ($F(1,15)=3.416$, $P=0.084$, $\eta^2=0.001$), time of coverage ($F(1,15)=1.000$, $P=0.333$, $\eta^2=4.729e-14$), and occurrence ($F(1,15)=0.171$, $P=0.685$, $\eta^2=2.812e-4$). JZS Bayes factor ANOVAs with default prior scales revealed that the main effects models were preferred to the interaction model by Bayes factors of 4.739, 7.957, and 5.824, for mean duration, time of coverage, and occurrence, respectively. Moreover, the analyses revealed that models without the main effect of testing session were preferred to the main effects models by Bayes factors of 8.440, 5.464, and 5.051, for mean duration, time of coverage, and occurrence, respectively.

For the FEP_3 comparison, two-way rm-ANOVAs yielded no non-significant testing session (First, Second, and Third Session) × microstate class (A, B, C, and D) interactions for mean duration ($F(6,60)=0.513$, $P=0.796$, $\eta^2=0.009$), time of coverage ($F(6,60)=0.210$, $P=0.972$, $\eta^2=0.003$), and occurrence ($F(6,60)=0.255$, $P=0.955$, $\eta^2=0.004$), as well as non-significant testing session differences for mean duration ($F(2,20)=0.885$, $P=0.428$, $\eta^2=0.002$), time of coverage ($F(2,20)=0.443$, $P=0.648$, $\eta^2=3.214e-14$), and occurrence ($F(2,20)=0.289$, $P=0.752$, $\eta^2=0.001$). JZS Bayes factor ANOVAs with default prior scales revealed that the main effects models were preferred to the interaction model by Bayes factors of 14.333, 24.000, and 22.500, for mean duration, time of coverage, and occurrence, respectively. Moreover, the analyses revealed that models without the main effect of testing session were preferred to the main effects models by Bayes factors of 11.628, 13.889, and 11.111, for mean duration, time of coverage, and occurrence, respectively.

References:

25. Erdfelder E, Faul F, Buchner A (1996): GPOWER: A general power analysis program. *Behavior Research Methods, Instruments, & Computers* 28: 1–11.

26. Cohen J (1988): *Statistical Power Analysis for the Behavioral Sciences*, 2nd ed. Routledge. <https://doi.org/10.4324/9780203771587>

27. Wagenmakers E-J (2007): A practical solution to the pervasive problems of p values. *Psychonomic Bulletin & Review* 14: 779–804.

28. Rouder JN, Morey RD, Speckman PL, Province JM (2012): Default Bayes factors for ANOVA designs. *Journal of Mathematical Psychology* 56: 356–374.

- Explained variance in FEP seems low compared to study 1, would the authors like to comment on that?

Response:

Thank you for the question. We speculate that the low explained variance in the FEP group might be due to the heterogenous nature of the group. We have now mentioned this point in the manuscript and, to support it, we have added in the “Supplementary Information” a table with the different subtypes of FEP in our study.

2. Results

2.2. Study 2: FEP :

The four microstate classes explained 73.97% of the global variance, across participants. The lower percentage of explained variance compared to patients, siblings, controls (sub-section 2.1), though in the normal range reported in the literature (65–84% (5)), might be due to the diverse diagnosis of the FEP group (**Supplementary Table S11**).

Supplementary Information
 2. Supplementary Methods and Materials
 2.1. Study 2: FEP :

Table S11 - Subtypes of FEP diagnosis of FEP according to the DSM-IV for all three testing sessions

Testing Session	Number of patients	Diagnosis (DSM-IV)
First (total n = 22)		
	2	Schizophrenia, Disorganized Type (295.1)
	16	Schizophrenia, Paranoid Type (295.3)
	3	Schizophrenia, Undifferentiated Type (295.9)
	1	Bipolar I Disorder, Most Recent Episode Depressed, In Partial Remission (296.55)
Second (total n = 16)		
	1	Schizophrenia, Disorganized Type (295.1)
	8	Schizophrenia, Paranoid Type (295.3)
	2	Schizoaffective Disorder (295.7)
	3	Schizophrenia, Undifferentiated Type (295.9)
	1	Bipolar I Disorder, Most Recent Episode Depressed, Mild (296.51)
	1	Bipolar I Disorder, Most Recent Episode Depressed, In Partial Remission (296.55)
Third (total n = 11)		
	8	Schizophrenia, Paranoid Type (295.3)
	2	Schizophrenia, Undifferentiated Type (295.9)
	1	Bipolar I Disorder, Most Recent Episode Depressed, In Partial Remission (296.55)

Reference:

5. Michel CM, Koenig T (2018): EEG microstates as a tool for studying the temporal dynamics of whole-brain neuronal networks: A review. *NeuroImage* 180: 577–593.

- 50% attrition in study 3 is problematic even for linear mixed models (and the final sample size of 11 is quite small)

Response:

Thank you for the comment. We agree that an attrition of 50% is problematic for linear mixed effect models even though the model converges. We have modified the analysis to repeated measures ANOVAs by dividing the FEP groups into two subsets (FEP_2 (completed first and second session, n = 16) and FEP_3 (completed all the three sessions, n = 11)), and analyzing the subgroups separately. Again, we found no significant main effects of testing session nor significant testing session x microstate class interaction effects, for any of the computed microstate parameters. Repeated measures Bayesian ANOVAs showed that there was positively more evidence for no testing session differences or testing session x microstate class interaction.

2. Results:

2.2. Study 2: FEP:

For the FEP_2 comparison, two-way rm-ANOVAs yielded no non-significant testing session (First and Session) \times microstate class (A, B, C, and D) interactions for mean duration ($F(3,45)=0.433, P=0.730, \eta^2=0.004$), time of coverage ($F(3,45)=0.512, P=0.676, \eta^2=0.005$), and occurrence ($F(3,45)=1.060, P=0.375, \eta^2=0.009$), as well as non-significant testing session differences for mean duration ($F(1,15)=3.416, P=0.084, \eta^2=0.001$), time of coverage ($F(1,15)=1.000, P=0.333, \eta^2=4.729e-14$), and occurrence ($F(1,15)=0.171, P=0.685, \eta^2=2.812e-4$). JZS Bayes factor ANOVAs with default prior scales revealed that the main effects models were preferred to the interaction model by Bayes factors of 4.739, 7.957, and 5.824, for mean duration, time of coverage, and occurrence, respectively. Moreover, the analyses revealed that models without the main effect of testing session were preferred to the main effects models by Bayes factors of 8.440, 5.464, and 5.051, for mean duration, time of coverage, and occurrence, respectively.

For the FEP_3 comparison, two-way rm-ANOVAs yielded no non-significant testing session (First, Second, and Third Session) \times microstate class (A, B, C, and D) interactions for mean duration ($F(6,60)=0.513, P=0.796, \eta^2=0.009$), time of coverage ($F(6,60)=0.210, P=0.972, \eta^2=0.003$), and occurrence ($F(6,60)=0.255, P=0.955, \eta^2=0.004$), as well as non-significant testing session differences for mean duration ($F(2,20)=0.885, P=0.428, \eta^2=0.002$), time of coverage ($F(2,20)=0.443, P=0.648, \eta^2=3.214e-14$), and occurrence ($F(2,20)=0.289, P=0.752, \eta^2=0.001$). JZS Bayes factor ANOVAs with default prior scales revealed that the main effects models were preferred to the interaction model by Bayes factors of 14.333, 24.000, and 22.500, for mean duration, time of coverage, and occurrence, respectively. Moreover, the analyses revealed that models without the main effect of testing session were preferred to the main effects models by Bayes factors of 11.628, 13.889, and 11.111, for mean duration, time of coverage, and occurrence, respectively.

4. Methods and Materials

4.2. EEG Recording and Data Processing:

To investigate whether the computed microstates changed throughout one year for the FEPs, we divided the analysis in two parts. First, we analyzed the microstate parameters in the FEP_2 (FEP that completed the first and second testing session). Then, we analyzed the microstate parameters in the FEP_3 (FEP that completed all the three testing sessions). In both cases, for each computed microstate parameter, we computed a two-way rm-ANOVA with testing session (for FEP_2: first and second; for FEP_3: first, second, and third) and microstate class as factors.

Regarding the final sample size ($n=11$), we agree that it is small and we mention in the discussion that the sample is small and that interpretation should be taken with care.

Discussion:

We re-tested FEP patients two other times, separated by six months, and found that the microstates dynamics remained stable. However, this interpretation should be taken with care since only a subset of the initial 22 FEP (16 in the second testing and 11 in the third testing) participated in the three tests.

- Again, symptom severity and medication status are not commented on.

Response:

We thank the reviewer for the comment. We have now correlated the negative, positive symptoms, medications status with the microstate parameters, and discussed them in the discussion as follows:

Discussion:

Regarding associations between psychopathological symptoms and microstate dynamics, in FEP, we observed negative correlations between the SANS scores and the time coverage of microstate class D as well as between the SANS scores and the occurrence of microstate class D; however, the correlations were not significant after correcting for multiple comparisons. For chronic schizophrenia patients, we found no significant correlations between psychopathological symptoms and the microstate parameters for any of the microstate classes. Nonetheless, the coefficients of the Pearson correlation between the SANS scores of schizophrenia patients and their microstate parameters for microstate class D were negative as in FEP (mean duration: $r(99)=-0.144$, $p=0.150$; time coverage: $r(99)=-0.190$, $p=0.057$; occurrence: $r(99)=-0.183$, $p=0.067$; **Supplementary Table S7**). In the literature, the duration of microstate class D has been found to correlate negatively with scores of paranoid-hallucinatory symptomatology (18) and with acute hallucination experiences (43) in patients with schizophrenia. More recently, it has been reported that the time coverage of microstate class A correlated positively with avolition-apathy scores, even though there were no group differences between patients and controls (30). Finally, in a sample of adolescents with 22q11.2 deletion syndrome, the mean durations of microstate class C were associated with increased hallucination subscores of the Structured Interview for Prodromal Syndromes (23). These results suggest that there might be an association between the microstate dynamics and psychopathological symptoms, however, the results in the literature are too heterogeneous to make firm conclusions at this point.

There are several considerations that should be taken into account. First, there are demographics differences between schizophrenia patients, their siblings, and controls. Tomescu and colleagues (41) showed evidence for age and gender-specific effects on the microstates dynamics. Here, we tried to minimize these effects by using age as a co-variate and gender as a factor in the analyses. Second, schizophrenia is a heterogeneous disease and our samples may be too small to cover the

full schizophrenia spectrum. Third, we cannot exclude the potential effects of treatment in the microstate class B differences between siblings and patients.

Most of our patients were medicated and, in chronic schizophrenia patients, we found a positive association with medication intake (CPZ equivalents) and the occurrence of microstate class C (although not significant after correction for multiple comparisons), providing evidence that medication interact with microstates dynamics. This potential interaction is also supported by previous work that has shown that perospirone (an antipsychotic drug) can increase the duration of microstate class D in healthy controls (44). In addition, antipsychotic medication has been shown to normalize microstate dynamics (decrease presence of microstate class C and increase presence of microstate class D), in patients that respond well to antipsychotic treatment (33). While most of the studies included in our meta-analysis only investigated the microstates dynamics in medication naïve patients, few studies investigated patients taking anti-psychotic medication. One of these studies found that microstate class D was decreased in FEP compared to controls (31), while another found increased duration and time coverage of microstate class C in schizophrenia patients compared to controls (30). Additionally, a recent study with FEP also identified decreased mean durations of microstate class A in FEP compared to controls, a result that does not align with the literature (29). However, since most of the studies of EEG microstates in schizophrenia have small samples ($n < 30$), it is expected that, due to sampling error and the heterogeneity of the disorder, some effects might not be significant in some studies and even reversed in a few studies if the effect sizes are small.

Supplementary Information

1. Supplementary Results

1.1. Study 1: Patients, siblings, and controls:

In chronic schizophrenia patients, the CPZ equivalents were found to correlate with the occurrence of microstate class C (**Table S7**). However, the correlation was not significant after correction for 12 comparisons (4 microstate classes \times 3 microstate parameters) with Bonferroni-Holm ($r(86)=0.236$, $p=0.027$, $p_{holm}=0.324$). No other significant associations were found between the computed microstate parameters and either CPZ equivalent, SANS, SAPS, or illness duration.

Table S7 - Pearson correlation between all computed microstate parameters and Chlorpromazine equivalent (CPZ), Scale for Assessment of Negative (SANS) and Positive (SAPS) Symptoms, and Illness duration, for chronic schizophrenia patients. (df - degrees of freedom). Statistically significant correlations (without correction for multiple comparisons) are indicated in bold.

Microstate Parameter	Microstate Class	CPZ (df = 86)	SANS (df = 99)	SAPS (df = 99)	Illness Duration (df = 99)
Mean Duration	A	$r=-0.015$, $p=0.890$	$r=0.018$, $p=0.862$	$r=0.007$, $p=0.948$	$r=0.069$, $p=0.490$
	B	$r=-0.057$, $p=0.596$	$r=0.056$, $p=0.576$	$r=0.041$, $p=0.685$	$r=-0.062$, $p=0.539$
	C	$r=0.012$, $p=0.910$	$r=-0.017$, $p=0.866$	$r=-0.066$, $p=0.514$	$r=-0.053$, $p=0.598$
	D	$r=0.092$, $p=0.391$	$r=-0.144$, $p=0.150$	$r=0.004$, $p=0.972$	$r=0.042$, $p=0.674$
Time Coverage	A	$r=-0.041$, $p=0.702$	$r=0.028$, $p=0.778$	$r=0.026$, $p=0.794$	$r=0.089$, $p=0.374$
	B	$r=-0.098$, $p=0.366$	$r=0.119$, $p=0.237$	$r=0.039$, $p=0.702$	$r=-0.065$, $p=0.516$
	C	$r=0.038$, $p=0.724$	$r=0.032$, $p=0.749$	$r=-0.048$, $p=0.635$	$r=-0.060$, $p=0.552$
	D	$r=0.079$, $p=0.465$	$r=-0.190$, $p=0.057$	$r=0.010$, $p=0.918$	$r=0.057$, $p=0.571$
Occurrence	A	$r=-0.011$, $p=0.9211$	$r=0.034$, $p=0.734$	$r=0.045$, $p=0.653$	$r=0.080$, $p=0.426$
	B	$r=-0.062$, $p=0.567$	$r=0.121$, $p=0.228$	$r=0.037$, $p=0.717$	$r=-0.073$, $p=0.469$
	C	$r=0.236$, $p=0.027$	$r=0.010$, $p=0.919$	$r=0.033$, $p=0.743$	$r=-0.085$, $p=0.397$
	D	$r=0.081$, $p=0.453$	$r=-0.183$, $p=0.067$	$r=0.017$, $p=0.863$	$r=0.036$, $p=0.720$

Supplementary Information

1. Supplementary Results

1.2. Study 2: FEP:

In FEP, the SANS scores correlated negatively with the time coverage and occurrence of microstate class D (**Table S9**). However, the correlations were not significant after correction for 12 comparisons (4 microstate classes \times 3 microstate parameters) with Bonferroni-Holm (time coverage of microstate class D: $r(20)=-0.516$, $p=0.014$, $p_{holm}=0.168$; occurrence of microstate class C: $r(20)=-0.429$, $p=0.046$, $p_{holm}=0.506$). No other significant associations were found between the computed microstate parameters and either CPZ equivalent, SANS, SAPS, or illness duration.

Table S9 - Pearson correlation between all computed microstate parameters and Chlorpromazine equivalent (CPZ), Scale for Assessment of Negative (SANS) and Positive (SAPS) Symptoms, and Illness duration, for patients with a first episode of psychosis (FEP). (df - degrees of freedom). Statistically significant correlations (without correction for multiple comparisons) are indicated in bold.

Microstate Parameter	Microstate Class	CPZ (df = 18)	SANS (df = 20)	SAPS (df = 20)	Illness Duration (df = 20)
Mean Duration	A	$r=-0.154$, $p=0.517$	$r=0.197$, $p=0.379$	$r=0.179$, $p=0.426$	$r=0.117$, $p=0.605$
	B	$r=-0.125$, $p=0.600$	$r=-0.054$, $p=0.812$	$r=-0.126$, $p=0.576$	$r=-0.110$, $p=0.627$
	C	$r=0.242$, $p=0.304$	$r=0.106$, $p=0.638$	$r=0.170$, $p=0.449$	$r=-0.193$, $p=0.390$
	D	$r=-0.178$, $p=0.452$	$r=-0.352$, $p=0.109$	$r=0.142$, $p=0.529$	$r=-0.230$, $p=0.303$
Time Coverage	A	$r=-0.156$, $p=0.511$	$r=0.244$, $p=0.273$	$r=0.150$, $p=0.504$	$r=0.205$, $p=0.359$
	B	$r=-0.127$, $p=0.595$	$r=0.055$, $p=0.808$	$r=-0.150$, $p=0.506$	$r=0.033$, $p=0.884$
	C	$r=0.240$, $p=0.309$	$r=0.049$, $p=0.828$	$r=0.069$, $p=0.759$	$r=-0.121$, $p=0.590$
	D	$r=-0.030$, $p=0.901$	$r=-0.516$, $p=0.014$	$r=-0.091$, $p=0.688$	$r=-0.118$, $p=0.600$
Occurrence	A	$r=-0.103$, $p=0.666$	$r=0.281$, $p=0.205$	$r=0.048$, $p=0.832$	$r=0.212$, $p=0.344$
	B	$r=-0.061$, $p=0.799$	$r=0.390$, $p=0.073$	$r=0.047$, $p=0.834$	$r=0.410$, $p=0.058$
	C	$r=0.134$, $p=0.573$	$r=0.075$, $p=0.741$	$r=0.079$, $p=0.728$	$r=0.084$, $p=0.709$
	D	$r=0.033$, $p=0.890$	$r=-0.429$, $p=0.046$	$r=-0.149$, $p=0.507$	$r=0.045$, $p=0.844$

References:

18. Koenig T, Lehmann D, Merlo MCG, Kochi K, Hell D, Koukkou M (1999): A deviant EEG brain microstate in acute, neuroleptic-naïve schizophrenics at rest. *European Archives of Psychiatry and Clinical Neuroscience* 249: 205–211.
23. Tomescu MI, Rihs TA, Becker R, Britz J, Custo A, Grouiller F, *et al.* (2014): Deviant dynamics of EEG resting state pattern in 22q11.2 deletion syndrome adolescents: A vulnerability marker of schizophrenia? *Schizophrenia Research* 157: 175–181.
29. Murphy M, Stickgold R, Öngür D (2019): Electroencephalogram Microstate Abnormalities in Early-Course Psychosis. *Biological Psychiatry: Cognitive Neuroscience and Neuroimaging*. <https://doi.org/10.1016/j.bpsc.2019.07.006>

30. Giordano GM, Koenig T, Mucci A, Vignapiano A, Amodio A, Di Lorenzo G, *et al.* (2018): Neurophysiological correlates of Avolition-apathy in schizophrenia: A resting-EEG microstates study. *NeuroImage: Clinical* 20: 627–636.
31. Andreou C, Faber PL, Leicht G, Schoettle D, Polomac N, Hanganu-Opatz IL, *et al.* (2014): Resting-state connectivity in the prodromal phase of schizophrenia: Insights from EEG microstates. *Schizophrenia Research* 152: 513–520.
33. Kikuchi M, Koenig T, Wada Y, Higashima M, Koshino Y, Strik W, Dierks T (2007): Native EEG and treatment effects in neuroleptic-naïve schizophrenic patients: Time and frequency domain approaches. *Schizophrenia Research* 97: 163–172.
41. Tomescu MI, Rihs TA, Rochas V, Hardmeier M, Britz J, Allali G, *et al.* (2018): From swing to cane: Sex differences of EEG resting-state temporal patterns during maturation and aging. *Developmental Cognitive Neuroscience* 31: 58–66.
43. Kindler J, Hubl D, Strik WK, Dierks T, Koenig T (2011): Resting-state EEG in schizophrenia: Auditory verbal hallucinations are related to shortening of specific microstates. *Clinical Neurophysiology* 122: 1179–1182.
44. Yoshimura M, Koenig T, Irisawa S, Isotani T, Yamada K, Kikuchi M, *et al.* (2007): A pharmaco-EEG study on antipsychotic drugs in healthy volunteers. *Psychopharmacology* 191: 995–1004.

Meta-analysis:

- The authors do not state anywhere how they conducted the search, criteria of study inclusion etc. It is not clear, for example, why they included two older studies that had been excluded from the meta-analysis by Rieger et al.

Response:

We thank the reviewer for the comment. We have added to the manuscript the information about how we conducted the literature research; our criteria for inclusion and exclusion of a study are as follows:

4. Methods and Materials

4.3. Meta-analysis:

A literature search was conducted for papers published before 29th November 2019 via PubMed, to identify studies investigating EEG microstates dynamics in schizophrenia. The key words were ‘schizophreni*’ in conjunction with ‘microstate*’, in order to get schizophrenia, schizophrenic, and schizophrenics as well as microstate and microstates. Furthermore, a prior meta-analysis and two reviews on EEG microstates were inspected for potentially missed studies (4,5,15). We identified 28 relevant studies. For our meta-analysis, we selected studies according to the following criteria:

- Criterion 1. The study reported original data from a group of patients belonging to the psychosis spectrum as well as a healthy control group.
- Criterion 2. The reported sample sizes, summary statistics, or *t*-, *F*-, or *p*-values had to be sufficiently detailed in order to compute effect sizes estimates and their variances. If the relevant information was not provided, we contacted the corresponding authors of the studies and asked for additional information. This was the case for 2 studies, (18) and (29): for these, Thomas Koenig and Michael Murphy, authors of (18) and (29), respectively, provided the summary statistics, via e-mail.
- Criterion 3. The EEG montage employed the standard 10-20 system.
- Criterion 4. Four microstate classes (A, B, C, and D) were considered, since this is the number of microstate classes most frequently used in the literature (5).
- Criterion 5. The study was a resting-state study, i.e., participants were not engaged in any particular task.
- Criterion 6. The study reported at least one of the following 3 microstate parameters: mean duration, time coverage, and occurrence.

Only 9 of the initial identified 28 studies met these 6 criteria. In addition, we included the current study (da Cruz et al.) and removed one study (6) because it consisted of a subset of participants of the current study. Hence, we included a total of 9 studies in our meta-analysis. A comprehensive list of all identified studies, with a short explanation for exclusion (if applicable), is presented in **Table S12** in **Supplementary Information**.

Apart from 3 studies, all the other studies reported the 3 relevant microstate parameters. The study by Nishida et al. (32) did not report the time coverage, the study by Giordano and colleagues (30) did not report the occurrence, while the study by Murphy et al. (29) only reported the mean duration (however, the time coverage and occurrence were obtained through personal correspondence). For each study, we calculated Cohen's *d* as the mean difference between patients and controls divided by the within group standard deviation, for each available microstate parameter and for each microstate class. For the current study (da Cruz et al.), we used the Cohen's *d* values reported in **Table 1**, which are corrected for gender and education differences. Hedges' *g* was calculated using Cohen's *d* multiplied by the coefficient *J*, which is a correction for small samples (52).

Hedges' *g* values were introduced as a generic effect size in the OpenMeta Analyst software (<http://www.cebm.brown.edu/openmeta/>) with the corresponding standard error (SE). We used the continuous random-effect analysis with the Restricted Maximum Likelihood (REML) method. The meta-analysis software computed the effect sizes, with 95% confidence intervals (C.I.) and the pooled effect size *g**. *P*-values were corrected for 12 comparisons (3 microstate parameters × 4 microstate classes) using Bonferroni-Holm correction.

Table 1 - Pairwise group comparisons for all microstate parameters (mean duration, time coverage, and occurrence) and for each microstate class (A, B, C, and D). Statistically significant differences, after Bonferroni-Holm correction, are indicated in bold.

Comparison	Microstate Class	Mean Duration	Time Coverage	Occurrence
Patients vs. Controls	A	$p=0.054, p_{holm}=0.270, d=-0.293$	$p=0.449, p_{holm}=0.898, d=-0.110$	$p=0.882, p_{holm}=0.898, d=0.023$
	B	$p=0.003, p_{holm}=0.018, d=-0.454$	$p=0.074, p_{holm}=0.296, d=-0.271$	$p=0.112, p_{holm}=0.336, d=-0.247$
	C	$p=1.315e-4, p_{holm}=0.001, d=0.590$	$p=1.452e-7, p_{holm}=1.742e-7, d=0.827$	$p=1.170e-4, p_{holm}=0.001, d=0.602$
	D	$p=3.010e-6, p_{holm}=3.311e-5, d=-0.732$	$p=3.445e-6, p_{holm}=3.445e-5, d=-0.725$	$p=1.620e-4, p_{holm}=0.001, d=-0.578$
Siblings vs. Controls	A	$p=0.055, p_{holm}=0.288, d=-0.371$	$p=0.097, p_{holm}=0.288, d=-0.320$	$p=0.250, p_{holm}=0.288, d=0.221$
	B	$p=0.049, p_{holm}=0.288, d=0.381$	$p=0.048, p_{holm}=0.288, d=0.382$	$p=0.069, p_{holm}=0.288, d=0.351$
	C	$p=0.022, p_{holm}=0.154, d=0.445$	$p=0.001, p_{holm}=0.010, d=0.631$	$p=0.006, p_{holm}=0.048, d=0.533$
	D	$p=3.380e-4, p_{holm}=0.004, d=-0.707$	$p=1.465e-4, p_{holm}=0.002, d=-0.751$	$p=0.004, p_{holm}=0.036, d=-0.566$
Patients_32 vs. Siblings_32	A	$p=0.235, p_{holm}=1.000, d=0.214$	$p=0.020, p_{holm}=0.200, d=0.434$	$p=0.014, p_{holm}=0.154, d=0.462$
	B	$p=0.003, p_{holm}=0.036, d=-0.576$	$p=0.069, p_{holm}=0.621, d=-0.333$	$p=0.200, p_{holm}=1.000, d=-0.231$
	C	$p=0.838, p_{holm}=1.000, d=-0.036$	$p=0.783, p_{holm}=1.000, d=0.049$	$p=0.588, p_{holm}=1.000, d=0.097$
	D	$p=0.270, p_{holm}=1.000, d=-0.199$	$p=0.491, p_{holm}=1.000, d=-0.123$	$p=0.772, p_{holm}=1.000, d=-0.052$

Table S12 - List of studies identified during the literature search and information whether it was included or excluded from the meta-analysis.

N	Study ID	Excluded	Exclusion Reason	Population
1	Koenig et al., 1999	no		Schizophrenia
2	Lehmann et al., 2005	no		Schizophrenia
3	Kikuchi et al., 2007	no		Schizophrenia
4	Nishida et al., 2013	no		Schizophrenia
5	Andreou et al., 2014	no		FEP
6	Tomescu et al., 2015	yes	same participants as in da Cruz et al., current	Schizophrenia
7	Tomescu et al., 2014	no		22q11
8	Irisawa et al., 2006	yes	3 classes	Schizophrenia
9	Strelets et al., 2003	yes	not 10-20 system, low n of electrodes	Schizophrenia
10	Giordano et al., 2018	no		Schizophrenia
11	Murphy et al., 2019	no		FEP
12	Soni et al., 2019	yes	task-related	Schizophrenia
13	Soni et al., 2018	yes	5 classes	Schizophrenia
14	Sverak et al., 2018	yes	5 classes and TMS	Schizophrenia
15	Rieger et al., 2016	yes	meta-analysis, no original data	
16	Diaz Hernandez et al., 2016	yes	no schizophrenia patients + neurofeedback	Healthy
17	Khanna et al., 2015	yes	review, no original data	
18	Schlegel et al., 2012	yes	personality traits, skeptical versus believer	Healthy
19	Kindler et al., 2011	yes	no control group, hallucination	Schizophrenia
20	Mucci et al., 2005	yes	schizotypy	Healthy
21	Stevens et al., 1997	yes	task-related and only one microstate	Schizophrenia
22	Kleinlogel et al., 2007	yes	task-related	Schizophrenia
23	Katayama et al., 2007	yes	no schizophrenia patients, hypnosis	Healthy
24	Yoshimura et al., 2007	yes	healthy participants and drugs	Healthy
25	Kochi et al., 1996	yes	Evoked-related potentials	Schizophrenia
26	Begré et al., 2008	yes	task (working memory)	Schizophrenia
27	Stirk et al., 1995	yes	no schizophrenia patients	Depressive
28	Michel and Koenig, 2018	yes	review, no original data	
29	da Cruz et al., current	no		Schizophrenia

References:

4. Khanna A, Pascual-Leone A, Michel CM, Farzan F (2015): Microstates in resting-state EEG: Current status and future directions. *Neuroscience & Biobehavioral Reviews* 49: 105–113.
5. Michel CM, Koenig T (2018): EEG microstates as a tool for studying the temporal dynamics of whole-brain neuronal networks: A review. *NeuroImage* 180: 577–593.
6. Tomescu MI, Rihs TA, Roinishvili M, Karahanoglu FI, Schneider M, Menghetti S, *et al.* (2015): Schizophrenia patients and 22q11.2 deletion syndrome adolescents at risk express the same deviant patterns of resting state EEG microstates: A candidate endophenotype of schizophrenia. *Schizophrenia Research: Cognition* 2: 159–165.

15. Rieger K, Diaz Hernandez L, Baenninger A, Koenig T (2016): 15 Years of Microstate Research in Schizophrenia – Where Are We? A Meta-Analysis. *Front Psychiatry* 7. <https://doi.org/10.3389/fpsy.2016.00022>
18. Koenig T, Lehmann D, Merlo MCG, Kochi K, Hell D, Koukkou M (1999): A deviant EEG brain microstate in acute, neuroleptic-naïve schizophrenics at rest. *European Archives of Psychiatry and Clinical Neurosciences* 249: 205–211.
23. Tomescu MI, Rihs TA, Becker R, Britz J, Custo A, Grouiller F, *et al.* (2014): Deviant dynamics of EEG resting state pattern in 22q11.2 deletion syndrome adolescents: A vulnerability marker of schizophrenia? *Schizophrenia Research* 157: 175–181.
29. Murphy M, Stickgold R, Öngür D (2019): Electroencephalogram Microstate Abnormalities in Early-Course Psychosis. *Biological Psychiatry: Cognitive Neuroscience and Neuroimaging*. <https://doi.org/10.1016/j.bpsc.2019.07.006>
30. Giordano GM, Koenig T, Mucci A, Vignapiano A, Amodio A, Di Lorenzo G, *et al.* (2018): Neurophysiological correlates of Avolition-apathy in schizophrenia: A resting-EEG microstates study. *NeuroImage: Clinical* 20: 627–636.
31. Andreou C, Faber PL, Leicht G, Schoettle D, Polomac N, Hanganu-Opatz IL, *et al.* (2014): Resting-state connectivity in the prodromal phase of schizophrenia: Insights from EEG microstates. *Schizophrenia Research* 152: 513–520.
32. Nishida K, Morishima Y, Yoshimura M, Isotani T, Irisawa S, Jann K, *et al.* (2013): EEG microstates associated with salience and frontoparietal networks in frontotemporal dementia, schizophrenia and Alzheimer’s disease. *Clinical Neurophysiology* 124: 1106–1114.
33. Kikuchi M, Koenig T, Wada Y, Higashima M, Koshino Y, Strik W, Dierks T (2007): Native EEG and treatment effects in neuroleptic-naïve schizophrenic patients: Time and frequency domain approaches. *Schizophrenia Research* 97: 163–172.
34. Lehmann D, Faber PL, Galderisi S, Herrmann WM, Kinoshita T, Koukkou M, *et al.* (2005): EEG microstate duration and syntax in acute, medication-naïve, first-episode schizophrenia: a multi-center study. *Psychiatry Research: Neuroimaging* 138: 141–156.
52. Francis G (2017): Equivalent statistics and data interpretation. *Behav Res* 49: 1524–1538.
- Related to the above: The term ‘update’ and the lack of methodological details makes me think that the authors did not conduct an independent search, but rather used the search results reported in the paper by Rieger *et al.* It might be advisable to avoid giving that impression, since none of the authors of the original meta-analysis were involved in the present study.

Response:

We thank the reviewer for the comment. We agree with the reviewer and we have now added to the manuscript the information about how we conducted the literature research. We also added our criteria for inclusion and exclusion of a study as well as more details regarding the results, as follows:

4. Methods and Materials

4.3. Meta-analysis:

A literature search was conducted for papers published before 29th November 2019 via PubMed, to identify studies investigating EEG microstates dynamics in schizophrenia. The key words were ‘schizophreni*’ in conjunction with ‘microstate*’, in order to get schizophrenia, schizophrenic, and schizophrenics as well as microstate and microstates. Furthermore, a prior meta-analysis and two reviews on EEG microstates were inspected for potentially missed studies (4,5,15). We identified 28 relevant studies. For our meta-analysis, we selected studies according to the following criteria:

- Criterion 1. The study reported original data from a group of patients belonging to the psychosis spectrum as well as a healthy control group.
- Criterion 2. The reported sample sizes, summary statistics, or *t*-, *F*-, or *p*-values had to be sufficiently detailed in order to compute effect sizes estimates and their variances. If the relevant information was not provided, we contacted the corresponding authors of the studies and asked for additional information. This was the case for 2 studies, (18) and (29): for these, Thomas Koenig and Michael Murphy, authors of (18) and (29), respectively, provided the summary statistics, via e-mail.
- Criterion 3. The EEG montage employed the standard 10-20 system.
- Criterion 4. Four microstate classes (A, B, C, and D) were considered, since this is the number of microstate classes most frequently used in the literature (5).
- Criterion 5. The study was a resting-state study, i.e., participants were not engaged in any particular task.
- Criterion 6. The study reported at least one of the following 3 microstate parameters: mean duration, time coverage, and occurrence.

Only 9 of the initial identified 28 studies met these 6 criteria. In addition, we included the current study (da Cruz et al.) and removed one study (6) because it consisted of a subset of participants of the current study. Hence, we included a total of 9 studies in our meta-analysis. A comprehensive list of all identified studies, with a short explanation for exclusion (if applicable), is presented in **Table S12 in Supplementary Information**.

Apart from 3 studies, all the other studies reported the 3 relevant microstate parameters. The study by Nishida et al. (32) did not report the time coverage, the study by Giordano and colleagues (30) did not report the occurrence, while the study by Murphy et al. (29) only reported the mean duration (however, the time coverage and occurrence were obtained through personal correspondence). For each study, we calculated Cohen’s *d* as the mean difference between patients and controls divided by the within group standard deviation, for each available microstate parameter and for each microstate class. For the current study (da Cruz et al.), we used the Cohen’s *d* values reported in **Table 1**, which are corrected for gender and education

differences. Hedges' *g* was calculated using Cohen's *d* multiplied by the coefficient *J*, which is a correction for small samples (52).

Hedges' *g* values were introduced as a generic effect size in the OpenMeta Analyst software (<http://www.cebm.brown.edu/openmeta/>) with the corresponding standard error (SE). We used the continuous random-effect analysis with the Restricted Maximum Likelihood (REML) method. The meta-analysis software computed the effect sizes, with 95% confidence intervals (C.I.) and the pooled effect size *g**. *P*-values were corrected for 12 comparisons (3 microstate parameters × 4 microstate classes) using Bonferroni-Holm correction.

Table 1 - Pairwise group comparisons for all microstate parameters (mean duration, time coverage, and occurrence) and for each microstate class (A, B, C, and D). Statistically significant differences, after Bonferroni-Holm correction, are indicated in bold.

Comparison	Microstate Class	Mean Duration	Time Coverage	Occurrence
Patients vs. Controls	A	p =0.054, p _{holm} =0.270, d =-0.293	p =0.449, p _{holm} =0.898, d =-0.110	p =0.882, p _{holm} =0.898, d =0.023
	B	p=0.003, p_{holm}=0.018, d=-0.454	p =0.074, p _{holm} =0.296, d =-0.271	p =0.112, p _{holm} =0.336, d =-0.247
	C	p=1.315e-4, p_{holm}=0.001, d=0.590	p=1.452e-7, p_{holm}=1.742e-7, d=0.827	p=1.170e-4, p_{holm}=0.001, d=0.602
	D	p=3.010e-6, p_{holm}=3.311e-5, d=-0.732	p=3.445e-6, p_{holm}=3.445e-5, d=-0.725	p=1.620e-4, p_{holm}=0.001, d=-0.578
Siblings vs. Controls	A	p =0.055, p _{holm} =0.288, d =-0.371	p =0.097, p _{holm} =0.288, d =-0.320	p =0.250, p _{holm} =0.288, d =0.221
	B	p =0.049, p _{holm} =0.288, d =0.381	p =0.048, p _{holm} =0.288, d =0.382	p =0.069, p _{holm} =0.288, d =0.351
	C	p =0.022, p _{holm} =0.154, d =0.445	p=0.001, p_{holm}=0.010, d=0.631	p=0.006, p_{holm}=0.048, d=0.533
	D	p=3.380e-4, p_{holm}=0.004, d=-0.707	p=1.465e-4, p_{holm}=0.002, d=-0.751	p=0.004, p_{holm}=0.036, d=-0.566
Patients_32 vs. Siblings_32	A	p =0.235, p _{holm} =1.000, d =0.214	p =0.020, p _{holm} =0.200, d =0.434	p =0.014, p _{holm} =0.154, d =0.462
	B	p=0.003, p_{holm}=0.036, d=-0.576	p =0.069, p _{holm} =0.621, d =-0.333	p =0.200, p _{holm} =1.000, d =-0.231
	C	p =0.838, p _{holm} =1.000, d =-0.036	p =0.783, p _{holm} =1.000, d =0.049	p =0.588, p _{holm} =1.000, d =0.097
	D	p =0.270, p _{holm} =1.000, d =-0.199	p =0.491, p _{holm} =1.000, d =-0.123	p =0.772, p _{holm} =1.000, d =-0.052

Table S12 - List of studies identified during the literature search and information whether it was included or excluded from the meta-analysis.

N	Study ID	Excluded	Exclusion Reason	Population
1	Koenig et al., 1999	no		Schizophrenia
2	Lehmann et al., 2005	no		Schizophrenia
3	Kikuchi et al., 2007	no		Schizophrenia
4	Nishida et al., 2013	no		Schizophrenia
5	Andreou et al., 2014	no		FEP
6	Tomescu et al., 2015	yes	same participants as in da Cruz et al., current	Schizophrenia
7	Tomescu et al., 2014	no		22q11
8	Irisawa et al., 2006	yes	3 classes	Schizophrenia
9	Strelets et al., 2003	yes	not 10-20 system, low n of electrodes	Schizophrenia
10	Giordano et al., 2018	no		Schizophrenia
11	Murphy et al., 2019	no		FEP
12	Soni et al., 2019	yes	task-related	Schizophrenia
13	Soni et al., 2018	yes	5 classes	Schizophrenia
14	Sverak et al., 2018	yes	5 classes and TMS	Schizophrenia
15	Rieger et al., 2016	yes	meta-analysis, no original data	
16	Diaz Hernandez et al., 2016	yes	no schizophrenia patients + neurofeedback	Healthy
17	Khanna et al., 2015	yes	review, no original data	
18	Schlegel et al., 2012	yes	personality traits, skeptical versus believer	Healthy
19	Kindler et al., 2011	yes	no control group, hallucination	Schizophrenia
20	Mucci et al., 2005	yes	schizotypy	Healthy
21	Stevens et al., 1997	yes	task-related and only one microstate	Schizophrenia
22	Kleinlogel et al., 2007	yes	task-related	Schizophrenia
23	Katayama et al., 2007	yes	no schizophrenia patients, hypnosis	Healthy
24	Yoshimura et al., 2007	yes	heathy participants and drugs	Healthy
25	Kochi et al., 1996	yes	Evoked-related potentials	Schizophrenia
26	Begré et al., 2008	yes	task (working memory)	Schizophrenia
27	Stirk et al., 1995	yes	no schizophrenia patients	Depressive
28	Michel and Koenig, 2018	yes	review, no original data	
29	da Cruz et al., current	no		Schizophrenia

2. Results

2.3. Meta-analysis:

Our literature search identified 8 independent studies comparing the resting-state dynamics of the 4 *canonical* microstate classes in patients belonging to the schizophrenia spectrum against a control group (18,23,29–34). In addition to these 8 studies, we also included the current study in the meta-analysis. Forest plots of the mean effect sizes for each microstate parameter (mean duration, time coverage, and occurrence) and microstate class (A, B, C, and D) are shown in the Supplementary Figure S1 - Figure S12. Similar to Rieger and colleagues (15), we found consistently increased time coverage ($g=0.447$, $p=6.304e-5$, $p_{holm}=6.934e-4$) and occurrence ($g=0.688$, $p=2.430e-13$, $p_{holm}=2.916e-12$) of microstate class C in patients compared to controls, as well as decreased time coverage ($g=-0.506$, $p=0.003$, $p_{holm}=0.027$) and mean duration ($g=-0.540$, $p=7.170e-4$, $p_{holm}=0.007$) of microstate class D in patients compared to controls. We also found a decreased mean duration of microstate class B in patients compared to controls;

however, the effect was not significant after correction for multiple comparisons ($g=-0.353$, $p=0.017$, $p_{holm}=0.136$). No consistent group differences were found for microstate class A.

Figure S1 - Forest plot of studies considering the mean duration of microstate class A ($N=685$, $k=9$, $g=-0.232$, $p=0.140$, $p_{holm}=0.889$). Results show no consistent group differences between patients and controls. (Heterogeneity $I^2=68\%$, $p=0.002$).

Figure S2 - Forest plot of studies considering the time coverage of microstate class A ($N=647$, $k=8$, $g=0.004$, $p=0.980$, $p_{holm}=1.000$). Results show no consistent group differences between patients and controls. (Heterogeneity $I^2=65\%$, $p=0.006$).

Figure S3 - Forest plot of studies considering the occurrence of microstate class A (N=479, k=8, $g=0.318$, $p=0.127$, $p_{holm}=0.889$). Results show no consistent group differences between patients and controls. (Heterogeneity $I^2=74\%$, $p=0.0003$).

Figure S4 - Forest plot of studies considering the mean duration of microstate class B (N=685, k=9, $g=-0.353$, $p=0.017$, $p_{holm}=0.136$). We found that patients have shorter microstate class B mean durations than controls; however, the result was not significant after correction for multiple comparisons. (Heterogeneity $I^2=68\%$, $p=0.002$).

Figure S5 - Forest plot of studies considering the time coverage of microstate class B (N=647, k=8, $g=-0.048$, $p=0.754$, $p_{holm}=1.000$). Results show no consistent group differences between patients and controls. (Heterogeneity $I^2=62\%$, $p=0.010$).

Figure S6 - Forest plot of studies considering the occurrence of microstate class B (N=479, k=8, $g=0.148$, $p=0.378$, $p_{holm}=1.000$). Results show no consistent group differences between patients and controls. (Heterogeneity $I^2=66\%$, $p=0.005$).

Figure S7 - Forest plot of studies considering the duration of microstate class C (N=685, k=9, $g=0.078$, $p=0.611$, $p_{holm}=1.000$). Results show no consistent group differences between patients and controls. (Heterogeneity $I^2=69\%$, $p=0.001$).

Figure S8 - Forest plot of studies considering the time coverage of microstate class C (N=647, k=8, $g=0.447$, $p=6.304e-5$, $p_{holm}=6.934e-4$). We found that patients have significantly longer microstate class C time coverage than controls. (Heterogeneity $I^2=35\%$, $p=0.148$).

Figure S9 - Forest plot of studies considering the occurrence of microstate class C (N=479, k=8, $g=0.688$, $p=2.430e-13$, $p_{holm}=2.916e-12$). We found that microstate class C occurs significantly more in patients than controls. (Heterogeneity $I^2=8\%$, $p=0.367$).

Figure S10 - Forest plot of studies considering the mean duration of microstate class D (N=685, k=9, $g=-0.540$, $p=7.170e-4$, $p_{holm}=0.007$). We found that patients have significantly shorter microstate class B mean durations than controls. (Heterogeneity $I^2=73\%$, $p=0.0003$).

Figure S11 - Forest plot of studies considering the time coverage of microstate class D (N=647, k=8, $g=-0.506$, $p=0.003$, $p_{holm}=0.027$). We found that patients have significantly shorter microstate class D time coverage than controls. (Heterogeneity $I^2=68\%$, $p=0.002$).

Figure S12 - Forest plot of studies considering the occurrence of microstate class D (N=479, k=8, $g=-0.201$, $p=0.228$, $p_{holm}=1.000$). Results show no consistent group differences between patients and controls. (Heterogeneity $I^2=65\%$, $p=0.005$).

References:

4. Khanna A, Pascual-Leone A, Michel CM, Farzan F (2015): Microstates in resting-state EEG: Current status and future directions. *Neuroscience & Biobehavioral Reviews* 49: 105–113.
5. Michel CM, Koenig T (2018): EEG microstates as a tool for studying the temporal dynamics of whole-brain neuronal networks: A review. *NeuroImage* 180: 577–593.

6. Tomescu MI, Rihs TA, Roinishvili M, Karahanoglu FI, Schneider M, Menghetti S, *et al.* (2015): Schizophrenia patients and 22q11.2 deletion syndrome adolescents at risk express the same deviant patterns of resting state EEG microstates: A candidate endophenotype of schizophrenia. *Schizophrenia Research: Cognition* 2: 159–165.
15. Rieger K, Diaz Hernandez L, Baenninger A, Koenig T (2016): 15 Years of Microstate Research in Schizophrenia – Where Are We? A Meta-Analysis. *Front Psychiatry* 7. <https://doi.org/10.3389/fpsy.2016.00022>
18. Koenig T, Lehmann D, Merlo MCG, Kochi K, Hell D, Koukkou M (1999): A deviant EEG brain microstate in acute, neuroleptic-naïve schizophrenics at rest. *European Archives of Psychiatry and Clinical Neurosciences* 249: 205–211.
23. Tomescu MI, Rihs TA, Becker R, Britz J, Custo A, Grouiller F, *et al.* (2014): Deviant dynamics of EEG resting state pattern in 22q11.2 deletion syndrome adolescents: A vulnerability marker of schizophrenia? *Schizophrenia Research* 157: 175–181.
29. Murphy M, Stickgold R, Öngür D (2019): Electroencephalogram Microstate Abnormalities in Early-Course Psychosis. *Biological Psychiatry: Cognitive Neuroscience and Neuroimaging*. <https://doi.org/10.1016/j.bpsc.2019.07.006>
30. Giordano GM, Koenig T, Mucci A, Vignapiano A, Amodio A, Di Lorenzo G, *et al.* (2018): Neurophysiological correlates of Avolition-apathy in schizophrenia: A resting-EEG microstates study. *NeuroImage: Clinical* 20: 627–636.
31. Andreou C, Faber PL, Leicht G, Schoettle D, Polomac N, Hanganu-Opatz IL, *et al.* (2014): Resting-state connectivity in the prodromal phase of schizophrenia: Insights from EEG microstates. *Schizophrenia Research* 152: 513–520.
32. Nishida K, Morishima Y, Yoshimura M, Isotani T, Irisawa S, Jann K, *et al.* (2013): EEG microstates associated with salience and frontoparietal networks in frontotemporal dementia, schizophrenia and Alzheimer’s disease. *Clinical Neurophysiology* 124: 1106–1114.
33. Kikuchi M, Koenig T, Wada Y, Higashima M, Koshino Y, Strik W, Dierks T (2007): Native EEG and treatment effects in neuroleptic-naïve schizophrenic patients: Time and frequency domain approaches. *Schizophrenia Research* 97: 163–172.
34. Lehmann D, Faber PL, Galderisi S, Herrmann WM, Kinoshita T, Koukkou M, *et al.* (2005): EEG microstate duration and syntax in acute, medication-naïve, first-episode schizophrenia: a multi-center study. *Psychiatry Research: Neuroimaging* 138: 141–156.
52. Francis G (2017): Equivalent statistics and data interpretation. *Behav Res* 49: 1524–1538.

REVIEWERS' COMMENTS:

Reviewer #1 (Remarks to the Author):

The authors have successfully eliminated all of my concerns I had regarding the previous version of the paper, and I no further objections to publication.

Reviewer #2 (Remarks to the Author):

I found the manuscript substantially improved after the authors' careful revision. I especially liked the idea of using Bayesian ANOVAs to address sample size considerations.

I do have a final point that I believe should be addressed, regarding microstate truncation at the margins of the epochs: The authors believe that 'that does not pose a problem in our group comparisons because the amount of removed periods was similar across groups'. However, I would expect this claim to hold only if microstate frequency of occurrence was normally distributed and equal across groups, which is not the case. Could the authors back up their claim with some data (not necessarily to be included in the publication)?

REVIEWERS' COMMENTS:

Reviewer #1 (Remarks to the Author):

The authors have successfully eliminated all of my concerns I had regarding the previous version of the paper, and I no further objections to publication.

Response:

We thank the reviewer for the comments and previous suggestions.

Reviewer #2 (Remarks to the Author):

I found the manuscript substantially improved after the authors' careful revision. I especially liked the idea of using Bayesian ANOVAs to address sample size considerations.

I do have a final point that I believe should be addressed, regarding microstate truncation at the margins of the epochs: The authors believe that 'that does not pose a problem in our group comparisons because the amount of removed periods was similar across groups'. However, I would expect this claim to hold only if microstate frequency of occurrence was normally distributed and equal across groups, which is not the case. Could the authors back up their claim with some data (not necessarily to be included in the publication)?

Response:

Thank you for the comments and the question. To back-up the claim that potential truncated microstates do not pose a problem in our group comparisons because the amount of removed periods was similar across groups, we have re-processed the EEG data and, this time, we did not remove potential bad epochs. By not removing the potential bad epochs, we do not have the problem of microstate truncation when doing the back-fitting. Then, we re-did the microstates analysis on this new pre-processed dataset and compared the results with the results of the analysis done in the manuscript (where potential bad epochs were removed). The group summary statistics for all computed microstates parameters and classes as well as each pre-processing version (epochs removed and no epochs removed) are shown in Table A for patients with schizophrenia, their unaffected siblings, and healthy controls, and Table B for Patients_22 and FEP patients.

Finally, we computed a JZS Bayes ANOVA with default priors (1-3) with factors Group (either Patients, Siblings, and Controls, or Patients_22 and FEP), Microstate Class (A, B, C, and D), and Version (epochs removed and no epochs removed) for each of the computed microstates parameter (mean duration, time of coverage, and occurrence) to evaluate if the Version influenced the observed group effects in the manuscript.

In sum, for patients, siblings, and controls comparisons, group differences depend only on the microstate classes, independently of the removal or not of bad EEG epochs. Similarly, for FEP

vs. Patients_22, there is substantially more evidence for no group differences than for an effect of group, independently of the removal or not of bad EEG epochs.

First, for the case of Patients, Siblings, and Controls comparison:

- For mean duration, the results indicated that the interaction model Microstate Class + Group + Microstate Class \times Group was preferred to the models with Microstate Class + Group + Version + Microstate Class \times Group, Microstate Class + Group + Version + Microstate Class \times Group + Version \times Group, Microstate Class + Group + Version + Microstate Class \times Group + Version \times Group + Microstate Class \times Version + Microstate Class \times Version \times Group, and the Null model by Bayes factors of 12.636, 715.320, 8.131e+7, and 4.485e+207.
- For time of coverage, the results indicated that the interaction model Microstate Class + Group + Microstate Class \times Group was preferred to the models with Microstate Class + Group + Version + Microstate Class \times Group, Microstate Class + Group + Version + Microstate Class \times Group + Version \times Group, Microstate Class + Group + Version + Microstate Class \times Group + Version \times Group + Microstate Class \times Version + Microstate Class \times Version \times Group, and the Null model by Bayes factors of 19.161, 1282.550, 9.974e+7, and 8.330e+243.
- For occurrence, the results indicated that the interaction model Microstate Class + Group + Microstate Class \times Group was preferred to the models with Microstate Class + Group + Version + Microstate Class \times Group, Microstate Class + Group + Version + Microstate Class \times Group + Version \times Group, Microstate Class + Group + Version + Microstate Class \times Group + Version \times Group + Microstate Class \times Version + Microstate Class \times Version \times Group, and the Null model by Bayes factors of 12.250, 735.591, 1.777e+7, and 1.841e+137.

For the case of Patients_22 and FEP comparison:

- For mean duration, the results indicated that the model with only the Microstate Class effect was preferred to the models with Microstate Class + Group + Microstate Class \times Group, Microstate Class + Group + Version + Microstate Class \times Group, Microstate Class + Group + Version + Microstate Class \times Group + Version \times Group, Microstate Class + Group + Version + Microstate Class \times Group + Version \times Group + Microstate Class \times Version + Microstate Class \times Version \times Group, and the Null model by Bayes factors of 29.589, 263.926, 1430.369, 185499.062, and 1.977e+39.
- For time of coverage, the results indicated that the model with only the Microstate Class effect was preferred to the models with Microstate Class + Group + Microstate Class \times Group, Microstate Class + Group + Version + Microstate Class \times Group, Microstate Class + Group + Version + Microstate Class \times Group + Version \times Group, Microstate Class + Group + Version + Microstate Class \times Group + Version \times Group + Microstate Class \times Version + Microstate Class \times Version \times Group, and the Null model by Bayes factors of 53.385, 469.451, 2950.207, 607519.978, and 1.743e+57.

- For occurrence, that the model with only the Microstate Class effect was preferred to the models with Microstate Class + Group + Microstate Class \times Group, Microstate Class + Group + Version + Microstate Class \times Group, Microstate Class + Group + Version + Microstate Class \times Group + Version \times Group, Microstate Class + Group + Version + Microstate Class \times Group + Version \times Group + Microstate Class \times Version + Microstate Class \times Version \times Group, and the Null model by Bayes factors of 10.755, 77.614, 461.045, 172552.346, and 172552.346.

Table A - Group average statistics (\pm SD) of the Patients with Schizophrenia, their Unaffected Siblings and Healthy Controls all the computed microstates parameters, classes, and pre-processing versions (potential bad epochs removed or not).

Microstate Parameter	Microstate Class	Version	Group		
			Patients	Siblings	Controls
Mean Duration (ms)	A	Epochs removed	69.05 \pm 10.06	66.93 \pm 8.61	71.83 \pm 15.20
		No epochs removed	69.40 \pm 11.21	69.88 \pm 11.09	71.22 \pm 11.62
	B	Epochs removed	66.34 \pm 8.97	74.81 \pm 11.87	71.05 \pm 8.52
		No epochs removed	67.37 \pm 8.08	74.36 \pm 11.86	72.43 \pm 10.58
	C	Epochs removed	106.99 \pm 27.75	99.05 \pm 22.16	90.31 \pm 18.05
		No epochs removed	107.46 \pm 25.06	99.66 \pm 28.65	91.19 \pm 16.94
	D	Epochs removed	70.42 \pm 11.16	71.41 \pm 11.02	82.01 \pm 16.84
		No epochs removed	72.05 \pm 12.10	73.05 \pm 12.38	80.20 \pm 17.66
Time Coverage (%)	A	Epochs removed	18.21 \pm 8.98	15.75 \pm 6.79	18.85 \pm 11.03
		No epochs removed	17.98 \pm 9.65	17.21 \pm 9.44	18.77 \pm 9.04
	B	Epochs removed	16.15 \pm 7.71	22.48 \pm 9.99	19.27 \pm 7.35
		No epochs removed	15.84 \pm 6.96	21.97 \pm 9.72	20.20 \pm 8.29
	C	Epochs removed	46.38 \pm 12.98	41.68 \pm 12.70	34.00 \pm 11.87
		No epochs removed	46.07 \pm 13.74	40.16 \pm 16.10	34.77 \pm 10.88
	D	Epochs removed	19.27 \pm 8.38	20.10 \pm 7.73	27.89 \pm 11.61
		No epochs removed	20.11 \pm 9.14	20.67 \pm 9.47	26.26 \pm 12.97
Occurrence	A	Epochs removed	1.87 \pm 0.66	1.69 \pm 0.62	1.82 \pm 0.61
		No epochs removed	1.83 \pm 0.65	1.72 \pm 0.75	1.86 \pm 0.57
	B	Epochs removed	1.74 \pm 0.58	2.12 \pm 0.59	1.93 \pm 0.53
		No epochs removed	1.71 \pm 0.54	2.08 \pm 0.60	1.96 \pm 0.50
	C	Epochs removed	2.83 \pm 0.38	2.74 \pm 0.36	2.52 \pm 0.45
		No epochs removed	2.80 \pm 0.43	2.57 \pm 0.54	2.54 \pm 0.38
	D	Epochs removed	1.95 \pm 0.59	2.00 \pm 0.58	2.34 \pm 0.59
		No epochs removed	1.97 \pm 0.61	1.97 \pm 0.67	2.19 \pm 0.75

Table B - Group average statistics (\pm SD) of the FEP and Patients_22 all the computed microstates parameters, classes, and pre-processing versions (potential bad epochs removed or not).

Microstate Parameter	Microstate Class	Version	Group	
			FEP	Patients_22
Mean Duration (ms)	A	Epochs removed	69.97 \pm 18.62	66.74 \pm 9.86
		No epochs removed	67.20 \pm 10.44	68.70 \pm 9.63
	B	Epochs removed	78.07 \pm 39.85	67.71 \pm 10.00
		No epochs removed	72.85 \pm 9.22	68.61 \pm 7.97
	C	Epochs removed	100.80 \pm 22.48	103.85 \pm 13.08
		No epochs removed	103.63 \pm 20.58	99.85 \pm 12.87
	D	Epochs removed	68.50 \pm 11.55	68.90 \pm 8.91
		No epochs removed	68.86 \pm 11.89	73.80 \pm 12.97
Time Coverage (%)	A	Epochs removed	17.51 \pm 12.58	16.73 \pm 7.32
		No epochs removed	16.03 \pm 7.70	17.56 \pm 7.63
	B	Epochs removed	21.34 \pm 14.75	17.74 \pm 8.25
		No epochs removed	20.90 \pm 7.89	17.37 \pm 6.98
	C	Epochs removed	43.08 \pm 16.43	46.91 \pm 8.32
		No epochs removed	44.50 \pm 14.45	43.56 \pm 8.79
	D	Epochs removed	18.07 \pm 9.10	18.63 \pm 8.53
		No epochs removed	18.57 \pm 8.43	21.51 \pm 10.46
Occurrence	A	Epochs removed	1.73 \pm 0.61	1.78 \pm 0.61
		No epochs removed	1.71 \pm 0.63	1.84 \pm 0.57
	B	Epochs removed	1.92 \pm 0.53	1.83 \pm 0.54
		No epochs removed	2.03 \pm 0.59	1.82 \pm 0.47
	C	Epochs removed	2.68 \pm 0.62	2.94 \pm 0.23
		No epochs removed	2.71 \pm 0.39	2.89 \pm 0.25
	D	Epochs removed	1.88 \pm 0.59	1.91 \pm 0.63
		No epochs removed	1.91 \pm 0.47	2.03 \pm 0.69

References:

1. Wagenmakers, E.-J. A practical solution to the pervasive problems of p values. *Psychon. Bull. Rev.* **14**, 779–804 (2007).
2. Rouder, J. N., Morey, R. D., Speckman, P. L. & Province, J. M. Default Bayes factors for ANOVA designs. *J. Math. Psychol.* **56**, 356–374 (2012).

3. Ly, A., Verhagen, J. & Wagenmakers, E.-J. Harold Jeffreys's default Bayes factor hypothesis tests: Explanation, extension, and application in psychology. *J. Math. Psychol.* **72**, 19–32 (2016).